

# How to escape atypical regions in the symmetric binary perceptron: A journey through connected-solutions states

**Damien Barbier**

Information Learning & Physics laboratory,
École Polytechnique Fédérale de Lausanne, Lausanne, Switzerland

[damien.barbier@epfl.ch](mailto:damien.barbier@epfl.ch)

## Abstract

We study the binary symmetric perceptron model, and in particular its atypical solutions. While the solution-space of this problem is dominated by isolated configurations [1], it is also solvable for a certain range of constraint density $\alpha$ and threshold $\kappa$. We provide in this paper a statistical measure probing sequences of solutions, where two consecutive elements shares a strong overlap. After simplifications, we test its predictions by comparing it to Monte-Carlo simulations. We obtain good agreement and show that connected states with a Markovian correlation profile can fully decorrelate from their initialization only for $\kappa > \kappa_{\text{no−mem. state}}$ ($\kappa_{\text{no−mem. state}} \sim \sqrt{0.91 \log(N)}$ for $\alpha = 0.5$ and $N$ being the dimension of the problem). For $\kappa < \kappa_{\text{no−mem. state}}$, we show that decorrelated sequences still exist but have a non-trivial correlations profile. To study this regime we introduce an Ansatz for the correlations that we label as the nested Markov chain.

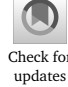
doi:[10.21468/SciPostPhys.18.3.115](https://doi.org/10.21468/SciPostPhys.18.3.115)

# 1  Introduction

## 1.1  Background and motivation

We consider the symmetric binary perceptron (SBP), introduced in [1], where we have a set $\underline{\xi} = \{\xi^{\mu}\}_{\mu \in [\![1,M]\!]}$ of $M$ i.i.d. standard Gaussian random vectors in $\mathbb{R}^N$, such that $M = \lfloor \alpha N \rfloor$ with $\alpha > 0$. This model consists in a constraint satisfaction problem for which binary vectors $\mathbf{x} \in \{-1, +1\}^N$ are solutions to the system of linear inequalities

$$\left| \xi^{\mu} \cdot \mathbf{x} \right| \leq \kappa \sqrt{N}, \quad \text{for all} \quad 1 \leq \mu \leq M, \tag{1}$$

with $\kappa > 0$. Defining this set of solutions by $S(\underline{\xi}, \kappa)$, it was proven by Aubin, Perkins and Zdeborová [1] that $S(\underline{\xi}, \kappa)$ is non-empty with high probability if and only if the margin threshold $\kappa$ verifies

$$\kappa > \kappa^{\alpha}_{\text{SAT}}, \quad \text{with} \quad \log(2) + \alpha \log\left( \int \mathcal{D}u \, \Theta(\kappa^{\alpha}_{\text{SAT}} - |u|) \right) = 0. \tag{2}$$

We will equation throughout this paper the notation $\mathcal{D}u$ to represent an integration with a scalar normal-distributed variable, and $\Theta(.)$ for the Heaviside function. This condition can also be reformulated as $S(\underline{\xi}, \kappa)$ being non-empty if and only if $\alpha$ verifies

$$\alpha < \alpha^{\kappa}_{\text{SAT}}, \quad \text{with} \quad \log(2) + \alpha^{\kappa}_{\text{SAT}} \log\left( \int \mathcal{D}u \, \Theta(\kappa - |u|) \right) = 0. \tag{3}$$

Our aim with this paper will be to analyse the geometrical arrangements of solutions for $\kappa > \kappa_{\text{SAT}}(\alpha)$ -or conversely $\alpha < \alpha^{\kappa}_{\text{SAT}}$-.

In their seminal paper [2], Mézard and Krauth showed with replica-based computation [3] that the set of solutions for the one-sided perceptron (where there is no absolute value in the constraints (1)) is dominated by *isolated* solutions. This means that the typical solutions to this problem are at large Hamming distance -linear in $N$- from any other solutions. It is an indirect consequence of their geometrical structure sometimes called "frozen replica symmetry breaking" [4–8]. From the mathematics point of view, the frozen replica symmetry breaking prediction was proven true for the SBP in works by Perkins and Xu [9] and Abbé, Li and Sly [10]. They showed more particularly that for $\kappa > \kappa_{\mathrm{SAT}}(\alpha)$, a solution drawn uniformly at random from $S(\underline{\xi}, \kappa)$ is *isolated* with high probability.

Having a constraint satisfaction problem with such a set of solutions has been usually associated with algorithmic hardness. Indeed, with the scope of using only local-moves algorithms, it is unlikely to find a routine capable of facing such extreme clustering. This phenomenology has been detailed and argued, for instance, by Zdeborová and Mézard [11], or Huang and Kabashima [7]. In some problems, this predicted algorithmic hardness was even confirmed empirically, [5, 11]. On the contrary, other constraint satisfaction problems are known to be solvable using certain efficient heuristics, while their typical solutions are predicted by statistical approaches to be isolated [6, 12–17]. A prime example of this is the above-defined binary perceptron, with symmetric or not threshold function. For this model, the solutions returned by efficient algorithms and their neighborhood have been the focus of several statistical mechanics papers [18–20]. One of the main intriguing observation of these studies is that solutions are arranged in dense regions. This could probably mean that *atypical, well connected* subset(s) of $S(\underline{\xi}, \kappa)$ are the regions found algorithmically. As mentioned earlier, these efficient algorithms also fail to return a solution if $\alpha$ is too large, which could be a hint for a *computational phase transition* in the binary perceptron.

Again in the symmetric binary perceptron, two recent mathematical works further elucidate the geometry of its solution landscape. First in [21], Abbé et al. show the existence of clusters comprising non-isolated solutions for $\kappa > \kappa_{\mathrm{SAT}}(\alpha)$ and $\alpha$ small enough. These clusters they found could possibly be probed with an algorithm as their diameter is linear in $N$. Secondly, in the small $\alpha$ regime, Gamarnik et al. [22] established an almost sharp result stating the following:

- the online algorithm of Bansal and Spencer [23] can find a solution for $\kappa \geq c_0 \sqrt{\alpha}$ with $c_0$ a positive constant.

- For $\kappa \leq c_1 \sqrt{-\alpha/\log(\alpha)}$, with $c_1$ a positive constant, $S(\underline{\xi}, \kappa)$ exhibits an *overlap gap property* ruling out local algorithms.

Being more precise, the first result is established in the case for which the data set $\underline{\xi}$ is Rademacher distributed instead of Gaussian. Nevertheless, the same result is expected in the Gaussian case.

With a different perspective, Baldassi et al. [24] suggested that this computational transition can be caused by the presence/disappearance of regions where atypical solutions have a monotonous *local entropy* as a function of the distance. More precisely, this local entropy measures the number of solutions that can be found at a given Hamming distance from a fixed configuration, in this case a targeted atypical solution. If such a conjecture is correct, then it must agree with the above-mentioned finding of Gamarnik et al. [22] in the regime of small $\alpha$.

In a previous paper [25], we elucidated why the replica method with a one-step replica symmetry breaking (1-RSB) Ansatz had so far not managed to find clusters of atypical solutions in the binary perceptron. As a reminder, this computation is a standard method for counting rare clusters in constraint satisfaction models, as long as they correspond to fixed points of a corresponding potential [26]. This analysis gave further elements for understanding that

solutions with a local maximum in the local entropy, put forward in [18–20], correspond actually to 1-RSB rare clusters. However, we will see in the very beginning of this paper that any atypical solutions -including these 1-RSB clustered solutions- are in fact isolated.

Keeping this phenomenology in mind, this paper focuses on introducing a formalism capable of probing non-isolated solutions and describing their configurational arrangement.

## 1.2 Summary of the results

**Solutions chain formalism:** We define a formalism that allows to study connected solutions of the SBP. It consists in building sequentially a chain of solutions $\{\mathbf{x}_j\}_{j\in[\![1,t]\!]}$, the starting configuration $\mathbf{x}_0$ being a solution we plant. Each "link" $\mathbf{x}_j$ is a solution with threshold $\kappa_j$ and built following the same pattern: having determined the chain from $\mathbf{x}_0$ to $\mathbf{x}_{j-1}$, we pick $\mathbf{x}_j$ by sampling a local entropy measure biased around $\mathbf{x}_{j-1}$. In practice the local entropy probes a subset of $S(\underline{\xi},\kappa_j)$ for which we have the additional constraint $\mathbf{x}_j \cdot \mathbf{x}_{j-1}/N = m_{j,j-1}$. This additional constraint is the reason why we call this approach a "chain formalism" as the set of solutions $\{\mathbf{x}_j\}_{j\in[\![1,t]\!]}$ verifies for any consecutive "links" $\mathbf{x}_j \cdot \mathbf{x}_{j-1}/N = m_{j,j-1}$.

In the general case, evaluating the local entropy via replica computation is involving as any replica Ansatz can be considered a priori. However, when taking the limit $m_{j,j-1} \to 1$ it can be shown that an *annealed* evaluation of the entropy becomes exact (the interested reader can find a detailed explanation about this result in App. B.2). This simplification is yet not sufficient for our computation to be numerically tractable. Indeed, it also requires to evaluate the entire correlation function $m_{j,j'} = \mathbf{x}_j \cdot \mathbf{x}_{j'}/N$ for all $j$ and $j'$. This task, close from dynamical mean-field theory computation [27–29], is know to be difficult.

**The no-memory Ansatz:** In a first attempt to determine chains of solutions in the SBP, we start by focusing on so-called no-memory chains. It consists in supposing that a "link" solution $\mathbf{x}_j$ correlates non-trivially only with its neighbors $\mathbf{x}_{j-1}$ and $\mathbf{x}_{j+1}$. A direct consequence of this simplification is the solutions chain becoming Markovian. It allows to study analytically the chain and the evolution of its local entropy as the number of "links" is increased. For fixed $\alpha$ and fixed overlap ($\mathbf{x}_j \cdot \mathbf{x}_{j-1}/N = m$ for all j) we identify a critical threshold $\kappa^{\alpha}_{\text{no−mem. state}} \sim \sqrt{-\log(1-m)}$ above which no-memory chains can be of infinite size and consequently be delocalized, i.e. $\lim_{t\to+\infty} \mathbf{x}_t \cdot \mathbf{x}_0/N = 0$.

We compare these prediction with Monte-Carlo simulations and observe good agreements when we set that two consecutive "link" solutions differ only by a spin flip, i.e. $m = 1 - 2/N$. More particularly, we show that the distribution of margins follow a profile that is not the one of typical solutions of this problem. For $\kappa < \kappa^{\alpha}_{\text{no−mem. state}}$, the no-memory chain describes the behavior of the Monte-Carlo dynamics as long as the total number of spin flip trials in the dynamics scales linearly with the size of the system. For longer time-scales, simulations attest to the presence of delocalized connected states which are not Markovian and thus do not follow the no-memory chain predictions.

**The nested Markov chain Ansatz:** To describe non-Markovian chains, we introduce what we call a nested Markov chain Ansatz. It consists in a hierarchical interactions profile between the elements $\{\mathbf{x}_j\}_{j\in[\![1,t]\!]}$ of the chain -schematized in Fig. 10- that can be integrated recursively. With it we show that delocalized chains exist for a wider range of parameters $\{\alpha,\kappa\}$ than predicted with the no-memory Ansatz.

## 1.3 Organization of the paper

The rest of the paper is organized as follows:

- Section II focuses on showing that any planted configurations in the symmetric binary perceptron is isolated. It serves as a warm-up and justification for the introduction of a formalism guaranteeing connectivity between solutions.

- Section III is dedicated to the introduction of a "solutions chain" formalism. Broadly speaking, it consists in studying a sequence of solutions $\{\mathbf{x}_j\}_{t\in[\![1,t]\!]}$ for which two consecutive elements have a fixed overlap $\mathbf{x}_{j+1} \cdot \mathbf{x}_j/N = m_{j+1,j}$.

- Section IV focuses on simplifying the chain formalism by introducing an Ansatz for the correlation function $m_{j,j'} = \mathbf{x}_j \cdot \mathbf{x}_{j'}/N$ -with $j, j' \in [\![1,t]\!]^2$-. We will call it the no-memory Ansatz as it describes a Markov-like chain.

- Section V is devoted to put in parallel predictions from this no-memory Ansatz and Monte-Carlo simulations.

- Finally, in section VI we propose to go beyond the no-memory *Ansantz* with our chain formalism. More practically, we introduce a routine that allows us to study chains of solutions with non-trivial correlations $\{m_{j,j'}\}_{j,j'\in[\![1,t]\!]^2}$.

## 2  A first remark: All planted solutions are isolated

We start our analysis of the symmetric binary perceptron with a small warm-up. In this section, we will focus on the planted version of this model and highlight how any planted solution is isolated. Starting with definitions, the planted symmetric binary perceptron is built as follows: suppose that we fix a configuration $\mathbf{x}_0$ on the hypercube, we draw a biased data set $\{\xi^\mu\}_{\mu\in[\![1,M]\!]}$ following the rule

$$\xi^\mu = \frac{w^\mu \mathbf{x}_0}{\sqrt{N}} + \xi^{\top\mu}, \tag{4}$$

where $\{\xi^{\top\mu}\}_{\mu\in[\![1,M]\!]}$ is a set of i.i.d. random Gaussian variables orthogonal to $\mathbf{x}_0$ -i.e. $\xi^{\top\mu} \sim \mathcal{N}(0, \mathbf{I} - \frac{1}{N}\mathbf{x}_0\mathbf{x}_0^\top)$-. The set of margins $\{w^\mu = \xi^\mu \cdot \mathbf{x}_0 \sqrt{N}\}_{\mu\in[\![1,M]\!]}$ is drawn from an arbitrary distribution $P^\kappa[.]$. which verifies the constraint

$$P^\kappa[w] = 0, \quad \text{for} \quad |w| > \kappa. \tag{5}$$

Throughout this article we will keep the convention according to which the index $\kappa$ indicates that a margin distribution is null outside the interval $[-\kappa, \kappa]$. With this setting $\mathbf{x}_0$ is a solution of the planted perceptron with threshold $\kappa$, as it verifies $\xi^\mu \cdot \mathbf{x}_0 = \sqrt{N}w^\mu$ and $|w^\mu| < \kappa$.

In a previous paper focusing on this model [25] it has been showed that the local entropy of solutions around the planted solution $\mathbf{x}_0$ is [25]

$$\phi^\kappa_{\text{planted}}[\mathbf{x}_0, m] = \frac{1}{N}\mathbb{E}_\xi \left\{ \log \left[ \sum_{\substack{\mathbf{x}\in\Sigma^N \\ \text{s.t. } \frac{\mathbf{x}\cdot\mathbf{x}_0}{N}=m}} e^{\sum_{\mu=1}^M \log[\Theta(\kappa-|\xi^\mu\cdot\mathbf{x}|)]} \right] \right\} \tag{6}$$

$$= \operatorname*{opt}_{q,\hat{q},\hat{m}} \left\{ \phi^{\kappa,*}_{\text{planted}}[\mathbf{x}_0, q, \hat{q}, m, \hat{m}] \right\},$$

with

$$\phi_{\text{planted}}^{\kappa,*}[\mathbf{x}_0, q, \hat{q}, m, \hat{m}] = -\frac{1-q}{2}\hat{q} - m\hat{m} + \int \mathcal{D}z\, \phi_{\text{in}}[z, \hat{q}, \hat{m}] \tag{7}$$

$$+ \alpha \int \mathcal{D}z\, dw P^{\kappa}[w]\, \phi_{\text{out}}^{\kappa}[w, z, q, m],$$

$$\phi_{\text{out}}^{\kappa}[w, z, q, m] = \log\left[H(\kappa, mw + \sqrt{q - m^2}z, 1-q)\right], \tag{8}$$

$$\phi_{\text{in}}[z, \hat{q}, \hat{m}] = \log\left[\sum_{x=\pm 1} e^{(\hat{m} + \sqrt{\hat{q}}z)x}\right] = \log\left[2\cosh(\hat{m} + \sqrt{\hat{q}}z)\right], \tag{9}$$

$$H(x, y, z) = \int_{\frac{-y-x}{\sqrt{z}}}^{\frac{-y+x}{\sqrt{z}}} \mathcal{D}u = \frac{1}{2}\text{erf}\left[\frac{x-y}{\sqrt{2z}}\right] + \frac{1}{2}\text{erf}\left[\frac{x+y}{\sqrt{2z}}\right], \tag{10}$$

and where again $\mathcal{D}z$, $\mathcal{D}u$ represent an integration with a scalar normal-distributed variable. The magnetization $m$ is the overlap between the equilibrated system and the planted configuration, i.e. $\mathbb{E}_{\xi}[\mathbf{x}\cdot\mathbf{x}_0] = Nm$. The self-overlap $q$ corresponds to the overlap between two distinct typical configurations $a$ and $b$ of the equilibrium measure, i.e. $\mathbb{E}_{\xi}[\mathbf{x}^{\mathbf{a}} \cdot \mathbf{x}^{\mathbf{b}}] = Nq$. Finally $\hat{q}$ and $\hat{m}$ are external fields that fix these overlaps. For more details on the derivation of this local entropy we redirect the interested readers to previous statistical mechanics works [1,25].

The number of solutions that lie extremely close to the planted solution can be obtained by fixing $m \approx 1$ and optimizing over $\{q, \hat{q}, \hat{m}\}$ in the local entropy. In this case we identify a saddle-point that corresponds to the annealed replica Ansatz, i.e. $\{q = m^2, \hat{q} \ll \hat{m}, \hat{m} = \text{artanh}[m]\}$. More details about the demonstration can be found in App. A. In this setting, the total number of such solutions is therefore given by the annealed local entropy

$$\phi_{\text{planted}}^{\kappa,\text{annealed}}[\mathbf{x}_0, m] = -\frac{1-m}{2}\log\left(\frac{1-m}{2}\right) - \frac{1+m}{2}\log\left(\frac{1+m}{2}\right) \tag{11}$$

$$+ \alpha \int dw P^{\kappa}[w]\log\left[H(\kappa, mw, 1-m^2)\right].$$

In App. A we also show that at first order in $1 - m$ this local entropy boils down to

$$\phi_{\text{planted}}^{\kappa,\text{annealed}}[\mathbf{x}_0, m]\bigg|_{m\approx 1} \approx -\frac{1-m}{2}\log\left(\frac{1-m}{2}\right) \tag{12}$$

$$+ 2\alpha\sqrt{1-m}\int_0^{+\infty} dB\,\{P^{\kappa}[\kappa] + P^{\kappa}[-\kappa]\}\log\left[\frac{1+\text{erf}[B]}{2}\right].$$

It can be easily check numerically that there is a range of magnetization such that the local entropy remains strictly negative, as long as $P^{\kappa}[\kappa] + P^{\kappa}[-\kappa] \neq 0$. More precisely this simplification allows to check that

$$\exists m_0 \in [0, 1[ \quad \text{s.t.} \quad \phi_{\text{planted}}^{\kappa,\text{annealed}}[\mathbf{x}_0, m] < 0 \quad \forall m \in ]m_0, 1[. \tag{13}$$

This means that all the solutions that we can plant in the symmetric binary perceptron are isolated. Starting from the configuration $\mathbf{x}_0$, we have to flip an extensive number of spins in order to reach any other solution of the problem, this is often referred as an overlap gap. So as to beat this curse we will focus in the following on building connected solutions.

## 3 Probing connected solutions around a planted solution

In this section we will describe how we can find connected solutions around a planted configuration. A first tentative for finding such solutions has been carried out in [25]. In this paper

the authors avoided the overlap gap we described in the previous section by planting a solution with threshold $\kappa'$ and probing solutions with higher threshold $\kappa \geq \kappa'$. Such a release in the constraints of the problem allows for an exponential number of solutions to appear around the planted configuration. However, the distance between these new typical solutions and the planted one is extensive, i.e $Nm = \mathbb{E}_\xi[\mathbf{x} \cdot \mathbf{x}_0] = \mathcal{O}(N)$. This means that there is no control over whether or not there is a solution path connecting the planted configuration to these new typical solutions (at threshold $\kappa$ and overlap $m$). In other words, we do not know if there is a sequence of single spin flips which makes it possible to join $\mathbf{x}_0$ to $\mathbf{x}$ while remaining a solution of the perceptron with threshold $\kappa$.

In the following, we will therefore follow the same procedure of planting a solution with threshold $\kappa'$ and probing solutions with threshold $\kappa \geq \kappa'$. But we will build an equilibrium measure that imposes solutions to be connected. More practically speaking, we will focus on the equilibrium potential

$$V_t\left[\mathbf{x}_0, \{m_{j+1,j}\}_{j\in[\![0,t-1]\!]}, \{\kappa_j\}_{j\in[\![1,t]\!]}\right] \tag{14}$$

$$= \frac{1}{N}\mathbb{E}_\xi\left\{\sum_{\mathbf{x}_1} P^{\kappa_1}\left[\mathbf{x}_1\bigg|\frac{\mathbf{x}_0 \cdot \mathbf{x}_1}{N} = m_{1,0}\right] \times \cdots \times \sum_{\mathbf{x}_{t-1}} P^{\kappa_{t-1}}\left[\mathbf{x}_{t-1}\bigg|\frac{\mathbf{x}_{t-2} \cdot \mathbf{x}_{t-1}}{N} = m_{t-1,t-2}\right]\right.$$

$$\left.\times \log\left[\sum_{\substack{\mathbf{x}_t \in \Sigma^N \\ \text{s.t. } \frac{\mathbf{x}_t \cdot \mathbf{x}_{t-1}}{N} = m_{t,t-1}}} e^{\sum_{\mu=1}^M \log\left[\Theta(\kappa^t - |\xi^\mu \cdot \mathbf{x}_t|)\right]}\right]\right\},$$

with

$$P^{\kappa_j}\left[\mathbf{x}_j\bigg|\frac{\mathbf{x}_{j-1} \cdot \mathbf{x}_j}{N} = m_{j,j-1}\right] = \frac{\delta\left(\frac{\mathbf{x}_{j-1} \cdot \mathbf{x}_j}{N} - m_{j,j-1}\right)\prod_{\mu=1}^M \Theta\left(\kappa_j - |\xi^\mu \cdot \mathbf{x}_j|\right)}{Z^{\kappa_j}[\mathbf{x}_{j-1}, m_{j,j-1}]}, \tag{15}$$

and $Z^{\kappa_j}[\mathbf{x}_{j-1}, m_{j,j-1}]$ being the renormalization factor of the distribution. As in [25], $\mathbf{x}_0$ is a fixed planted solution that act like a quench variable. But in the setting above, we also add a solutions chain (from "link" $j = 1$ to $j = t-1$) that will bias the ensemble of solutions we probe. We call it a chain as two consecutive "link" configurations $\mathbf{x}_j$ and $\mathbf{x}_{j+1}$ are constrained to have an overlap $m_{j+1,j}$. We also impose that each "link" $\mathbf{x}_j$ is a solution of the constraint satisfaction problem with a given threshold $\kappa_j$. Thus, $V_t[.,.,.]$ is a local entropy which counts the number of solutions $\mathbf{x}_t$ that can form the next "link" in the chain (given that it has the correct threshold $\kappa_t$ and overlap $\mathbb{E}_\xi[\mathbf{x}_t \cdot \mathbf{x}_{t-1}/N] = m_{t,t-1}$). If the potential is positive, it means that $\mathbf{x}_t$ can be chosen among an exponential number of binary configurations. Therefore, we will consider that it is possible to join $\mathbf{x}_0$ to $\mathbf{x} = \mathbf{x}_t$ by exploring sequentially the solutions chain $\{\mathbf{x}_j\}_{j\in[\![0,t]\!]}$ if the chain potential $V_j[.,.,.]$ is positive for $j \in [\![1,t]\!]$. In fact, this construction shares similarities with previous works on quasi-equilibrium formalism, in particular for spin glasses models [30,31]. And more broadly speaking, sequences of conditional probabilities is a central feature of the dynamical mean-field theory framework [27–29].

After some computation steps it can be shown that the potential reads (the detailed computation can be found in App. B)

$$V_t\left[\mathbf{x}_0, \{m_{j+1,j}\}_{j\in[\![0,t-1]\!]}, \{\kappa_j\}_{j\in[\![1,t]\!]}\right] \tag{16}$$

$$= \operatorname*{opt}_{\substack{\{\hat{m}_{t,j'}\}_{0\leq j'\leq t-1} \\ \{m_{t,j'}\}_{0\leq j'\leq t-2}}}\left[\operatorname*{opt^*}_{\substack{\{\hat{m}_{j,j'}\}_{0\leq j'<j\leq t-1} \\ \{m_{j,j'}\}_{0\leq j'<j-1\leq t-2}}}\left\{V_t^*\left[\mathbf{x}_0, \{m_{j,j'}\}_{0\leq j'<j\leq t}, \{\hat{m}_{j,j'}\}_{0\leq j'<j\leq t}, \{\kappa_j\}_{j\in[\![1,t]\!]}\right]\right\}\right],$$

with

$$V_t^* \left[ \mathbf{x}_0, \{m_{j,j'}\}_{0 \leq j' < j \leq t}, \{\hat{m}_{j,j'}\}_{0 \leq j' < j \leq t}, \{\kappa_j\}_{j \in [\![1,t]\!]} \right] = -\sum_{j' < t} \hat{m}_{t,j'} m_{t,j'} \tag{17}$$

$$+ \sum_{x_0, \dots, x_{t-1} = \pm 1} \frac{1}{2} \prod_{j=1}^{t-1} \frac{e^{\sum_{0 \leq j' < j} \hat{m}_{j,j'} x_j x_{j'}}}{2 \cosh\left(\sum_{0 \leq j' < j} \hat{m}_{j,j'} x_{j'}\right)} \log\left[ 2 \cosh\left(\sum_{0 \leq j' < t} \hat{m}_{t,j'} x_{j'}\right) \right]$$

$$+ \alpha \int \prod_{j=0}^{t-1} dw_j \, P^{\kappa_0}[w_0] \prod_{j=1}^{t-1} \frac{e^{-\frac{\sum_{0 \leq j' \leq j} \Sigma_{j,j'}(\mathbf{m}) w_j w_{j'}}{2}} \Theta(\kappa^j - |w_j|)}{\int dw_j^* e^{-\frac{\sum_{0 \leq j' \leq j} \Sigma_{j,j'}(\mathbf{m}) w_j^* w_{j'}}{2}} \Theta(\kappa^j - |w_j^*|)}$$

$$\times \log\left[ \int dw_t \, e^{-\frac{\sum_{0 \leq j' \leq t} \Sigma_{t,j'}(\mathbf{m}) w_t w_{j'}}{2}} \Theta(\kappa^t - |w_t|) \right].$$

The matrix $\Sigma(\mathbf{m})$ simply fixes the covariances for the interactions $w_j^\mu = \xi^\mu \cdot \mathbf{x}_j$ as

$$\Sigma^{-1}(\mathbf{m})_{j,j'} = \mathbb{E}_\xi \left[ \frac{(\xi^\mu \cdot \mathbf{x}_j)(\xi^\mu \cdot \mathbf{x}_{j'})}{N} \right] = \mathbb{E}_\xi \left[ \frac{\mathbf{x}_j \cdot \mathbf{x}_{j'}}{N} \right] = m_{j,j'}, \quad \text{and} \quad m_{j,j} = 1. \tag{18}$$

We specified two types of optimization, respectively labeled "opt*" and "opt", in Eq. (16). The first one corresponds to a "time-ordered" optimization. By this we mean that the fields $\hat{m}_{j,j'}$ and overlaps $m_{j,j'}$ which do not involve the "link" at time $t$ are set by optimizing (in increasing order) the potentials $V_{t'=1}^*[.,.,.,.]$ to $V_{t'=t-1}^*[.,.,.,.]$. The second optimization corresponds simply to the usual maximization of the potential $V_t^*[.,.,.,.]$ over the variables $\hat{m}_{t,j'}$ and $m_{t,j'}$. To put it more concretely, we set $\hat{m}_{1,0}$ by optimizing $V_{t=1}^*[.,.,.,.]$. Then, we fix $\hat{m}_{2,0}$, $\hat{m}_{2,1}$ and $m_{2,0}$ by optimizing $V_{t=2}^*[.,.,.,.]$ (and so on and so forth). This time-ordered optimization is a direct consequence of the chain construction. Indeed, each conditional probabilities $P^{\kappa_{t-1}}\left[\mathbf{x}_t \Big| \frac{\mathbf{x}_t \cdot \mathbf{x}_{t-1}}{N} = m_{t,t-1}\right]$ depends solely on its past "links" $\mathbf{x}_{j(<t)}$. Thus, to characterize the configurations $\mathbf{x}_t$ dominating this measure, we should only optimize over the parameters $\{\hat{m}_{t,j'}\}_{0 \leq j' \leq t-1}$ and $\{m_{t,j'}\}_{0 \leq j' \leq t-2}$. A global optimization (not time-ordered) of $V_t^*[.]$ could yield a different saddle-point, it would describe a dynamics where time-ordering can be violated.

This optimization scheme actually becomes more and more difficult as we increase the number of "links" in the chain. Indeed, if we look at the $t^{\text{th}}$ "link" in the chain, we have to optimize the potential over $2(t-1)$ variables. It is thus crucial to simplify this procedure in order to study large chains of connected solutions. In the following section we will propose a first simple Ansatz for fields $\{\hat{m}_{j,j'}\}_{0 \leq j' < j \leq t}$ and overlaps $\{m_{j,j'}\}_{0 \leq j' < j \leq t}$. This Ansatz will drastically reduce the number of variables involved in the optimization scheme. A second leg of this paper will then be dedicated at refining this simple Ansatz. In the rest of this paper, we will always refer to the term involving a sum over $x$'s binary variables - respectively an integral over $w$'s continuous variables- in Eq. (17) as the entropic term -respectively the energetic term-.

## 4   A first simplification the potential: The no-memory Ansatz

### 4.1   Detailed simplifications

As mentioned just above, optimizing the potential $V_t^*[.,.,.,.]$ over the whole set of free variables $\{\hat{m}_{j,j'}\}$ and $\{m_{j,j'}\}$ is a difficult task. Therefore in this part, we will suppose that any given "link" -indexed $j$- only gets coupled with its nearest-neighboring "links", i.e. the ones labelled $j' = j \pm 1$. In other words, we will suppose that the number of solutions $\mathbf{x}_{j+1}$ probed

at the step $j+1$ only depends on the "link" configuration $\mathbf{x}_j$, and not on the whole set of configurations $\{\mathbf{x}_{j'}\}_{0\le j'\le j}$. Regarding the fields $\{\hat{m}_{j,j'}\}$, this no-memory Ansatz implies that we set

$$\hat{m}_{j,j'} = 0, \quad \text{for} \quad |j-j'| \ne 1. \tag{19}$$

Keeping only the nearest-neighbor interactions, the entropic term in the potential behaves like an effective 1D Ising spin chain, i.e.

$$\sum_{x_0,\dots,x_{t-1}=\pm 1} \frac{1}{2} \prod_{j=1}^{t-1} \frac{e^{\sum_{0\le j'<j}\hat{m}_{j,j'}x_j x_{j'}}}{2\cosh\left(\sum_{0\le j'<j}\hat{m}_{j,j'}x_{j'}\right)} \log\left[2\cosh\left(\sum_{0\le j'<t}\hat{m}_{t,j'}x_{j'}\right)\right] \tag{20}$$

$$= \sum_{x_0,\dots,x_{t-1}=\pm 1} \frac{1}{2} \prod_{j=1}^{t-1} \frac{e^{\hat{m}_{j,j-1}x_j x_{j-1}}}{2\cosh\left(\hat{m}_{j,j-1}\right)} \log\left[2\cosh\left(\hat{m}_{t,t-1}x_{t-1}\right)\right].$$

As in the 1D Ising spin chain, this implies that the overlaps have to verify

$$m_{j,j'} = \prod_{l=j'}^{j-1} m_{l+1,l}. \tag{21}$$

This last result allows for important simplifications in the correlation matrix $\Sigma(\mathbf{m})$. In fact, we have now that

$$\Sigma_{j,j'}^{-1}(\mathbf{m}) = \mathbb{E}_\xi\left[\frac{\mathbf{x}_j \cdot \mathbf{x}_{j'}}{N}\right] = \prod_{l=j'}^{j-1} m_{l+1,l}. \tag{22}$$

Keeping this correlation structure, we can perform the standard change of variable (see Appendix A in [1] for more details)

$$w_j = m_{j,j-1} w_{j-1} + \sqrt{1-m_{j,j-1}^2}\, u_j, \tag{23}$$

with $u_l \sim \mathcal{N}(0,1)$ for $l \in [\![1,j]\!]$. Implementing Eqs. (20,23) in our potential we can derive the simplified expression

$$V_t^*\left[\mathbf{x}_0, \{m_{j,j-1}\}_{1\le j\le t}, \hat{m}_{t,t-1}, \{\kappa_j\}_{j\in[\![1,t]\!]}\right] = -\hat{m}_{t,t-1} m_{t,t-1} + \log\left[2\cosh\left(\hat{m}_{t,t-1}\right)\right] \tag{24}$$

$$+ \alpha \int dw_{t-1} P^{\kappa_{t-1}}[w_{t-1}] \log\left[H\left(\kappa^t, m_{t,t-1}w_{t-1}, 1-m_{t,t-1}^2\right)\right],$$

with the update rule

$$P^{\kappa_j}[w_j] = \int dw_{j-1} \frac{e^{-\frac{(w_j - m_{j,j-1}w_{j-1})^2}{2\left(1-m_{j,j-1}^2\right)}} \Theta\left(\kappa_j - |w_j|\right)}{\sqrt{2\pi(1-m_{j,j-1}^2)}\, H\left(\kappa_j, m_{j,j-1}w_{j-1}, 1-m_{j,j-1}^2\right)} P^{\kappa_{j-1}}\left[w_{j-1}\right], \tag{25}$$

and the function $H(.,.,.)$ is given in Eq. (10). The detailed computation steps to obtain this potential can be found in App. C.

## 4.2 Solving the optimization and choosing the magnetizations

Now that we have simplified the potential $V_t^*[.,.,.,.]$, the double optimization required to derive $V_t[.,.,.,.]$ -see Eq. (16)- is trivial. Indeed, the no-memory Ansatz presented just above fixes all fields to zero except $\{\hat{m}_{j+1,j}\}_{j\in[\![0,t-1]\!]}$. We recall that almost all the fields, $\{\hat{m}_{j+1,j}\}_{j\in[\![0,t-2]\!]}$, appear in the "time-ordered" optimization labelled opt*. Again, this means that we set $\hat{m}_{1,0}$

by optimizing $V^*_{t=1}[.,.,.,.]$, then $\hat{m}_{2,1}$ by optimizing $V^*_{t=2}[.,.,.,.]$ and so on and so forth. We finally optimize the potential $V^*_t[.,.,.,.]$ over the last free variable, namely $\hat{m}_{t,t-1}$. Each of these optimizations is identical and simply boils down to verifying

$$\partial_{\hat{m}_{j+1,j}} \left\{ -\hat{m}_{j+1,j} m_{j+1,j} + \log\left[2\cosh\left(\hat{m}_{j+1,j}\right)\right] \right\} = 0, \quad \forall j \in [\![0, t-1]\!], \qquad (26)$$

$$\implies \hat{m}_{j+1,j} = \operatorname{artanh}\left[m_{j+1,j}\right].$$

In the following, we will consider for simplification that the distance between two consecutive "link" solutions $\mathbf{x}_j$ and $\mathbf{x}_{j+1}$ is constant along the chain, i.e. $m_{j+1,j} = m$. If we plug the saddle-point from Eq. (26) in the expression of $V^*_t[.,.,.,.]$ we finally obtain the expression for the potential $V_t[.,.,.,.]$

$$V_t\left[\mathbf{x}_0, m, \{\kappa_j\}_{j\in[\![1,t]\!]}\right] = -\frac{1-m}{2}\log\left(\frac{1-m}{2}\right) - \frac{1+m}{2}\log\left(\frac{1+m}{2}\right) \qquad (27)$$

$$+ \alpha \int dw_{t-1} P^{\kappa_{t-1}}[w_{t-1}] \log\left[H\left(\kappa^t, mw_{t-1}, 1-m^2\right)\right],$$

where we recall that the update rule for the distribution of interactions $P^{\kappa_j}[.]$ is

$$P^{\kappa_j}[w_j] = \int dw_{j-1} \frac{e^{-\frac{(w_j - mw_{j-1})^2}{2(1-m^2)}} \Theta\left(\kappa_j - |w_j|\right)}{\sqrt{2\pi(1-m^2)} H\left(\kappa_j, mw_{j-1}, 1-m^2\right)} P^{\kappa_{j-1}}\left[w_{j-1}\right]. \qquad (28)$$

In general one could be tempted to fine tune $m$ in order to optimize $V_t[.,.,.,.]$ along the chain. For example, by determining the overlap $m^*$ achieving the best minimal value in the chain, i.e.

$$m^* = \underset{m}{\operatorname{argmax}} \left\{ \min_t V_t\left[\mathbf{x}_0, m, \{\kappa_j\}_{j\in[\![1,t]\!]}\right] \right\}. \qquad (29)$$

However, our goal is different in this paper: we want to probe connected states. Therefore, we shall not optimize over $m$ but rather send it to 1. In the remainder of this section, we will discuss to which extend we can take the overlap $m$ close to one.

In the standard setting, the potential $V_t\left[\mathbf{x}_0, m, \{\kappa_j\}_{j\in[\![1,t]\!]}\right]$ is computed by taking the number of dimensions $N$ going to infinity first and keeping all parameters $\alpha$, $\kappa$'s and $m$ constants. This means for example that the distance between two consecutive solutions in the chain - which is $1-m$- cannot dependent on the system size. In practice, the computation can be generalized and $1-m$ can be a function of $N$. This will be particularly handy when comparing our chain approach to Monte-Carlo simulations. By definition Monte-Carlo dynamics follows paths of solutions by performing a sequence of single spin-flip (i.e. $m = 1-2/N$). To correctly describe a regime for which $N(1-m) = o(N)$, we need to control that the saddle-point approximation -used in the computation of $V_t[.,.,.,.]$- remains correct. In other words, when we take $N(1-m) = o(N)$ the finite-size corrections to the saddle-point evaluation have to remain negligible. A common result from the study of the canonical ensemble - see the Chapter 7.2 in [32]- is that these corrections are of order $\log(N)/N$. Thus, a simple criteria we can identify for the overlap is that

$$V_j\left[\mathbf{x_0}, m, \{\kappa_{j'}\}_{j'\in[\![1,j]\!]}\right] \gg \frac{\log(N)}{N}. \qquad (30)$$

For the sake of simplicity, if we only consider the entropic term of the potential we obtain

$$-\frac{1-m}{2}\log\left(\frac{1-m}{2}\right) - \frac{1+m}{2}\log\left(\frac{1+m}{2}\right) \gg \frac{\log(N)}{N}$$

$$\implies -\frac{1-m}{2}\log\left(\frac{1-m}{2}\right) \gg \frac{\log(N)}{N}$$

$$\implies 1-m \gg \frac{2}{N},$$

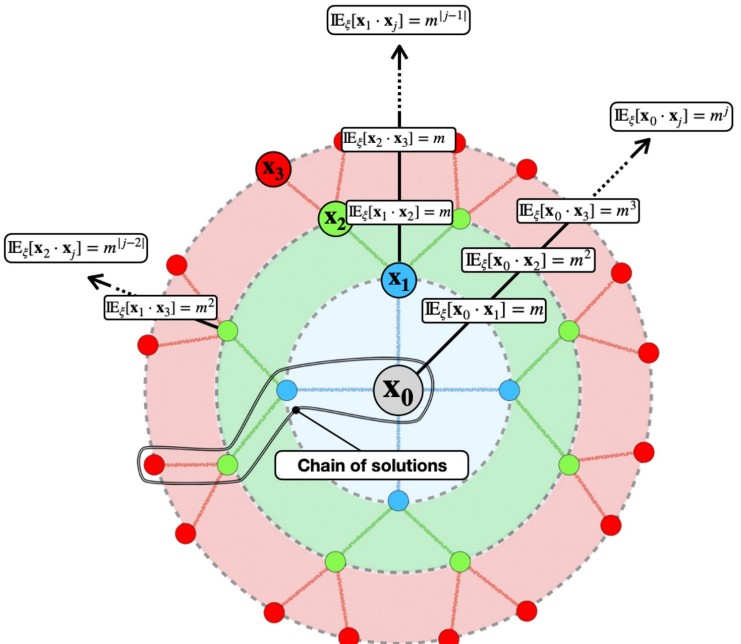

Figure 1: Schematic representation for the geometrical arrangement of the connected solutions we probe in the symmetric binary perceptron. At the center we find the initial planted configuration $\mathbf{x}_0$. Directly away from it we have the solutions $\mathbf{x}_1$, their typical number being $e^{NV_1[\mathbf{x}_0,m,\kappa_1]}$. Then, having fixed the solutions $\mathbf{x}_0$ and $\mathbf{x}_1$ we are able to probe $e^{NV_2[\mathbf{x}_0,m,\kappa_1,\kappa_2]}$ solutions $\mathbf{x}_2$. A chain of connected solutions corresponds to a set of solutions $\{\mathbf{x}_j\}_{j\in[\![1,t]\!]}$ where we have fixed the overlap between to consecutive configurations -$\mathbf{x}_{j+1} \cdot \mathbf{x}_j/N = m$-.

where $1-m = 2/N$ is exactly the distance between $\mathbf{x}_j$ and $\mathbf{x}_{j-1}$ when only one spin has been flipped. Taking into an account this criteria, we will consider the limiting case $1-m = 2f(N)/N$ where the function f(.) can be any function verifying $\lim_{N\to+\infty} f(N) = +\infty$ and $f(N) > 1$. Thus, we will probe solutions $\{\mathbf{x}_j\}_{0<j<t}$ which can be visited by sequentially flipping $f(N)$ spins, while correctly counting their number with the saddle-point evaluation of $V_t[.,.,.]$. In particular, we can set $f(N) \ll N$, which means that we flip a large but sub-extensive number of spins.

## 4.3 Some properties of the simplified potential

In this section we will detail the more straightforward properties implied by our construction of connected solutions. First, in Fig. 1 we represented a schematic view of the connected solutions arrangement. At the center we have the planted configuration $\mathbf{x}_0$. Then, we find $e^{NV_1[\mathbf{x}_0,m,\kappa_1]}$ solutions $\mathbf{x}_1$ to form the first "link" in our chain -at a distance corresponding to $\mathbb{E}_\xi[\mathbf{x}_0 \cdot \mathbf{x}_1] = m$-. Subsequently, we select a given solution $\mathbf{x}_1$ and find $e^{NV_2[\mathbf{x}_0,m,\kappa_1,\kappa_2]}$ solutions $\mathbf{x}_2$ for the second "link" in the chain. Again, the distance between $\mathbf{x}_1$ and $\mathbf{x}_2$ is such that $\mathbb{E}_\xi[\mathbf{x}_1 \cdot \mathbf{x}_2] = m$. The no-memory Ansatz is such that $\mathbb{E}_\xi[\mathbf{x}_0 \cdot \mathbf{x}_2] = m^2$, see Eq. (21). We can generalize this construction saying that we select a chain of connected configurations $\{\mathbf{x}_j\}_{1\leq j\leq t-1}$ and find $e^{NV_t[\mathbf{x}_0,m,\{\kappa_j\}_{j\in[\![1,t]\!]}]}$ solution $\mathbf{x}_t$ with $\mathbb{E}_\xi[\mathbf{x}_{t-1} \cdot \mathbf{x}_t] = m$. Again, Eq. (21) shows that

$$\mathbb{E}_\xi[\mathbf{x}_j \cdot \mathbf{x}_t] = m^{|t-j|}. \tag{31}$$

Another remark we can make is that $V^t[.,.,.]$ has exactly the same form as the potential $\phi_{\text{planted}}^{\kappa,\text{annealed}}[.,.]$, see Eq. (11). We recall that this potential counts the number of solutions with threshold $\kappa$ around a planted signal also with threshold $\kappa$. In Sec. 2, we showed with it that any planted signal is isolated as it verifies

$$\exists m_0 \in [0,1[ \quad \text{s.t.} \quad \phi_{\text{planted}}^{\kappa,\text{annealed}}[\mathbf{x}_0, m'] < 0, \quad \forall m' \in ]m_0, 1[. \tag{32}$$

This means that there is a finite distance range for which no solution can be found around the planted signal, it is often called an overlap-gap [33, 34]. If we transcribe this for our connected solutions setting, this shows that any "link" configuration $\mathbf{x}_{t-1}$ with threshold $\kappa_{t-1}$ is an isolated solution. In other words, if we set $\kappa_t \le \kappa_{t-1}$ we have

$$\exists m_0 \in [0,1[ \quad \text{s.t.} \quad V_t\Big[\mathbf{x}_0, m', \{\kappa_j\}_{j \in [\![1,t]\!]}\Big]\Big|_{\kappa_t \le \kappa_{t-1}} < 0, \quad \forall m' \in ]m_0, 1[. \tag{33}$$

In the case where $\kappa_t > \kappa_{t-1}$, solutions appear around $\mathbf{x}_{t-1}$ as the problem becomes less constrained. However, the overlap-gap is still present if the increase in $\kappa_t$ remains small. Thus, we observe numerically

$$\exists \{m_0, m_1\} \in [0,1[^2 \quad \text{s.t.} \quad V_t\Big[\mathbf{x}_0, m', \{\kappa_j\}_{j \in [\![1,t]\!]}\Big]\Big|_{\kappa_t > \kappa_{t-1}} < 0, \quad \forall m' \in ]m_0, m_1[. \tag{34}$$

If $\kappa_t$ is increased above a critical value $\kappa_c(t)$, the overlap-gap disappears. In this case, the value $\kappa_c(t)$ depends on the whole past of the chain as it is a function of the distribution of interactions $P^{\kappa_{t-1}}[.]$. This last regime will not be tackled in this paper. Indeed, setting $\kappa_t > \kappa_c(t)$ for all times $t$ yields to a fast divergence of $\kappa_t$ and we reach a regime for which the problem becomes trivial. In Fig. 2 we have summarized the behavior of $V_t\Big[\mathbf{x}_0, m', \{\kappa_j\}_{j \in [\![1,t]\!]}\Big]$ depending on the value of $\kappa_t$.

Regarding the chain formalism, the presence of this overlap-gap between $m_0$ and $m_1$ will dictate whether or not our chain of solutions stops or continues. As presented in Fig. 3, if the solutions $\mathbf{x}_t$ with $\mathbb{E}_\xi[\mathbf{x}_t \cdot \mathbf{x}_{t-1}] = m$ are in this overlap-gap ($V_t[\mathbf{x}_0, m, \{\kappa_j\}_{j \in [\![1,t]\!]}] < 0$) their typical number will be $\lim_{N \to +\infty} \exp(N V_t[\mathbf{x}_0, m, \{\kappa_j\}_{j \in [\![1,t]\!]}]) = 0$. In this case, $\mathbf{x}_{t-1}$ has no solution with threshold $\kappa_t$ at the distance we fixed and thus the chain of solution stops. On the contrary, if $V_t[\mathbf{x}_0, m, \{\kappa_j\}_{j \in [\![1,t]\!]}] > 0$ we find an exponential number of solutions $\mathbf{x}_t$ and the chain can continue. As we detailed in Fig. 3, different scenarios are possible depending on whether $\kappa_t \le \kappa_{t-1}$ or $\kappa_t > \kappa_{t-1}$. With the former the overlap-gap is between $m' = 1$ and $m' = m_0$, so the condition for the chain to continue is $m \le m_0$. In the following, we will see this condition when studying quenches, i.e $\kappa_1 > \kappa_0$ and $\kappa_{j(\neq 0)} = \kappa_1$. The latter situation is a bit more complex as the gap is between $m' = m_1$ and $m' = m_0$. In this case the chain can continue only if $m \notin ]m_0, m_1[$.

Obviously this construction raises the question of the feasibility of crossing such overlap gaps. In recent work [22, 33, 34], this has been considered to be algorithmically impossible as going from $\mathbf{x}_{t-1}$ to $\mathbf{x}_t$ means being able to flip an extensive number of spins in one go. However, with our connected solutions setting we have $m = 1 - 2f(N)/N$ with f(.) any function verifying $\lim_{N \to +\infty} f(N) = +\infty$ and $f(N) > 1$. This means that the number of spins we have to flip between $\mathbf{x}_{t-1}$ to $\mathbf{x}_t$ is actually $f(N)$, and it can be arbitrary small. Thus, if an overlap gap appears for a range of spin flips smaller than $f(N)$ we will consider that it is algorithmically possible to cross it.

When comparing with the original planted model of Sec. 2, it is now clear how the connected solutions formalism enables us to avoid the curse of observing only isolated solutions in this model. With the chain formalism a minimum distance $1 - m$ is set and divides overlap gaps into two category: either they are discarded when they appear below this minimal distance, or they are taken into account when they appear above it.

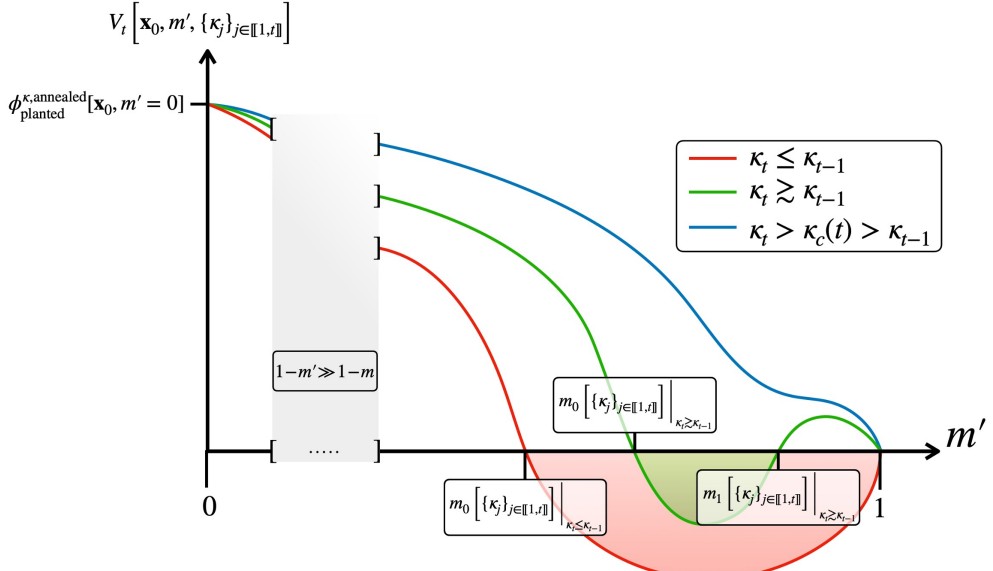

Figure 2: Sketch of the potential $V_t[\mathbf{x}_0, m', \{\kappa_j\}_{j\in[\![1,t]\!]}]$ as a function of $m'$. In this representation, we focus on a range of overlaps $m'$ close to the chain overlap $m$ (i.e. $1-m' \sim 1-m$). Each color corresponds to a different regime for $\kappa_t$. In red, we have $\kappa_t \leq \kappa_{t-1}$. In this case, the solution $\mathbf{x}_t$ is isolated for $m' \in [m_0, 1[$ as the potential is negative. In green, $\kappa_t$ has been slightly increased. Solutions have been created around $\mathbf{x}_{t-1}$ for a small range of overlaps ($m' \in [m_1, 1[$), but an overlap-gap remains (for $m' \in [m_1, m_0[$). The last regime regime, in blue, corresponds to a large increase of $\kappa_t$ -above a critical value $\kappa_c(t)$- for which the overlap-gap disappear. To have this regime at all time in the chain a large and constant increase in $\kappa_t$ is required.

## 5 The quenched procedure

In this section we will study the case of quench dynamics with the aim to compare the predictions made by the no-memory chain with Monte-Carlo simulations. Our quench dynamics setting corresponds to $\kappa_{j(\neq 0)} = \kappa(> \kappa_0)$ with an initial planted configuration $\mathbf{x}_0$ verifying

$$\kappa_0 = \kappa_{\text{SAT}}^\alpha \approx 0.31, \quad \text{with} \quad \alpha = 0.5, \tag{35}$$

$$P^{\kappa_0}[w] = \frac{e^{-w^2/2}}{\sqrt{2\pi}} \Theta(\kappa_0 - |w|). \tag{36}$$

With this distribution of margins $w$ the planted signal corresponds a typical solution with threshold $\kappa_{\text{SAT}}$ [1,25]. Choosing $\kappa_0 = \kappa_{\text{SAT}}$ is simply a convention and changing its value will have no incidence on our conclusions. We want also to note that all cases presented here will be for $\alpha = 0.5$.

### 5.1 The setting of the Monte-Carlo procedure and matching with the theoretical modelization

For the Monte-Carlo simulations, we will set $\mathbf{x}_0 = \{+1\}^N$, then draw the margins with the distribution from Eq. (36) and finally set the random data set $\{\xi^\mu\}_{\mu\in[\![1,M]\!]}$ following Eq. (4). The system is initialized in the planted configuration -$\mathbf{x}_{t=0} = \mathbf{x}_0$- and the dynamics goes as follows:



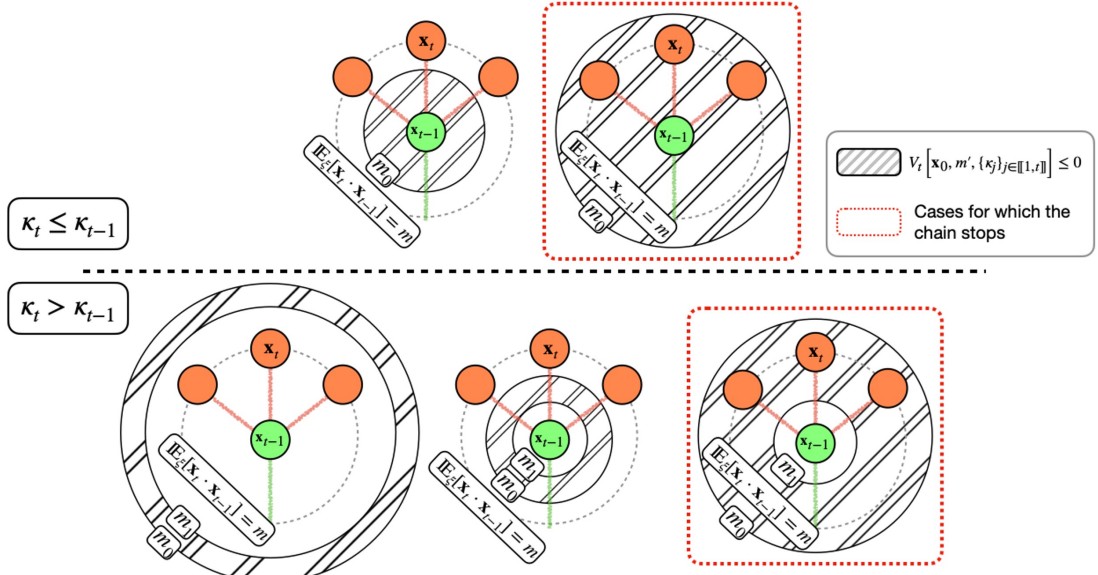

Figure 3: Representation of the different scenarios for the overlap gap around connected solutions. We have separated the cases $\kappa_t \leq \kappa_{t-1}$ (top) and $\kappa_t > \kappa_{t-1}$ (bottom). For both we have highlighted with a dotted red box the cases for which the chain of solutions stops, i.e when $V_t[\mathbf{x}_0, m, \{\kappa_j\}_{j \in [\![1,t]\!]}] < 0$.

- at each time-step $t$ we pick randomly an entry of $\mathbf{x}_{t-1}$ and change its sign, we label the resulting vector as $\mathbf{x}_t^*$.

- if $\mathbf{x}_t^*$ verify $|\xi^\mu \cdot \mathbf{x}_t^*|/\sqrt{N} < \kappa$ for all vectors $\xi^\mu$ in the data set, then we set $\mathbf{x}_t = \mathbf{x}_t^*$. If this condition is not verified then we take $\mathbf{x}_t = \mathbf{x}_{t-1}$.

So as to match Monte-Carlo simulations with our theoretical predictions, two essential remarks must be made. First, both approaches are dependent on the dimension $N$ of the problem. For the simulations this remark is trivial. But for the chain computation, we highlighted that the chain computation is valid until $1 - m \gg 2/N$. In the following we will crudely take the limiting case $1 - m = 2/N$ and check a posteriori that this distance allows a good match between the simulations and our theoretical predictions. With this, two neighbouring solutions in the chain $\mathbf{x}_j$ and $\mathbf{x}_{j+1}$ will be one spin flip away from each other (i.e. $f(N) = 1$).

Secondly, we also need to juxtapose correctly the time-scales between the simulations and the chain computation in order to compare the evolution of overlaps throughout the dynamics. The Monte-Carlo dynamics has a natural time-scale which is the one of a spin flip trial, i.e. being at time $t = 10$ means we have tried to flip consecutively 10 spins. In the following we will always call this the "natural" time-scale. In the case of the theoretical predictions, going from a "link" configuration $\mathbf{x}_{j-1}$ to $\mathbf{x}_j$ means that $f(N) = 1$ spins have been flipped. Therefore, we will define a "rescaled" time-scale corresponding to the number of accepted spin flips, i.e. $t_{\text{rescaled}} = 10$ means that $10 f(N)$ spins have been flipped. In particular, flipping two times the same entry during the dynamics still counts as two spin flips. To go from the "natural" to the "rescaled" time-scales, we introduce the matching function $g : t_{\text{rescaled}} \to t$. This function simply attributes the number of spin flip trials $t$ that are required to actually perform $t_{\text{rescaled}} f(N)$ spin flips, it will depend on each run of Monte-Carlo dynamics. In this scaling, Monte-Carlo simulations should approximately follow the overlaps rule from Eq. (31)

with

$$\frac{\mathbf{x}_{g(t_{\text{rescaled}})} \cdot \mathbf{x}_{g(t'_{\text{rescaled}})}}{N} \approx m^{|t_{\text{rescaled}} - t'_{\text{rescaled}}|} \tag{37}$$

$$\approx \left(1 - \frac{2f(N)}{N}\right)^{|t_{\text{rescaled}} - t'_{\text{rescaled}}|} .$$

This is an approximation as it supposes that the Monte-Carlo dynamics goes forward in a chain of solutions, i.e. it always go from "link" $\mathbf{x}_j$ to "link" $\mathbf{x}_{j+1}$. In reality we can observe backward moves, meaning the dynamics can go from "link" $\mathbf{x}_j$ to "link" $\mathbf{x}_{j-1}$. However, this remains negligible if the number of possible forward moves outweighs the number of backward ones. As a last note on the rescaling, we observe that the dynamics typically slows down. Thus, this induces the matching function $g : t_{\text{rescaled}} \rightarrow t$ to be convex, i.e.

$$g(t^2_{\text{rescaled}} + \Delta t_{\text{rescaled}}) - g(t^2_{\text{rescaled}}) > g(t^1_{\text{rescaled}} + \Delta t_{\text{rescaled}}) - g(t^1_{\text{rescaled}}), \tag{38}$$

for

$$t^2_{\text{rescaled}} > t^1_{\text{rescaled}}. \tag{39}$$

This inequality simply transcribes the fact that the later we are in a Monte-Carlo run, the longer it takes for the dynamics to accept $\Delta t_{\text{rescaled}} f(N)$.

## 5.2 The update of the chain for a quench procedure

For the chain computation the update scheme is in fact quite simple. It goes as follows:

- First, we compute $V_{t=1}[\mathbf{x}_0, m, \kappa_0, \kappa_{t=1} = \kappa]$ with Eq. (27) and the margin distribution $P^{\kappa_0}[.]$ of the planted signal.

- Then, if $V_{t=1}[\mathbf{x}_0, m, \kappa_0, \kappa_{t=1} = \kappa] > 0$, we derive the new distribution of interaction $P^{\kappa_{t=1}}[.]$ using Eq. (28).

- We repeat this process by computing $V_{t=2}[\mathbf{x}_0, m, \kappa_0, \kappa_1 = \kappa, \kappa_{t=2} = \kappa]$ and, if the potential is positive, we derive $P^{\kappa_{t=2}}[.]$.

- We keep this update for any step $t > 2$ until we obtain $V_t[\mathbf{x}_0, m, \kappa_0, \{\kappa_j = \kappa\}_{1 \leq j \leq t}] < 0$. Then, the chain stops.

One crucial remark has to be added. When performing a quench, the update rule [28] for the interaction distribution $P^{\kappa_j}[.]$ simplifies as the margin threshold remains constant: $\kappa_{j(>0)} = \kappa$. In this case we have

$$P^{\kappa_j}[w_j] = \int dw_{j-1} \frac{e^{-\frac{(w_j - mw_{j-1})^2}{2(1-m^2)}} \Theta(\kappa - |w_j|)}{\sqrt{2\pi(1-m^2)} H(\kappa, mw_{j-1}, 1-m^2)} P^{\kappa_{j-1}}[w_{j-1}], \quad \text{with } \forall j \neq 0, \quad \kappa_j = \kappa, \tag{40}$$

which corresponds to a Markov-Chain distribution update. Following results on Markov-chain processes [35], there exists only one stable distribution of interactions to this update. In other words, we have only one function $P[.]$ verifying

$$P[w'] = \int dw \frac{e^{-\frac{(w' - mw)^2}{2(1-m^2)}} \Theta(\kappa - |w'|)}{\sqrt{2\pi(1-m^2)} H(\kappa, mw, 1-m^2)} P[w]. \tag{41}$$

We will label this distribution $P^{\kappa}_{\text{no-mem. state}}[.]$.

A complementary point of view is to say that the linear operator $\mathbf{T}_{\kappa_j}$ corresponding to this update, $P^{\kappa_j} = \mathbf{T}_{\kappa_j} P^{\kappa_{j-1}}$, verifies the Perron-Frobenius theorem [36]. Thus, $\mathbf{T}_{\kappa_j}$ has a non-degenerate top eigenvector $P^\kappa_{\text{no-mem. state}}[.]$ with only positive entries and eigenvalue one. All other eigenvectors have an eigenvalue strictly smaller than one. As we update the distribution of interactions along the chain, the projection on each eigenvector -except for $P^\kappa_{\text{no-mem. state}}[.]$- will decay exponentially fast with a rate corresponding to their respective eigenvalue. Therefore, with an infinitely long chain, the distribution of interactions will inevitably collapse onto $P^\kappa_{\text{no-mem. state}}[.]$. In fact, this distribution has a simple expression, it is

$$P^\kappa_{\text{no-mem. state}}[w] = \frac{e^{\frac{-w^2}{2}} H(\kappa, mw, 1-m^2)}{\mathcal{N}}, \tag{42}$$

with $\mathcal{N}$ ensuring its normalization.

In order to fully decorrelate from any planted signal we need this top eigenvector to correspond to non-isolated solutions in the perceptron, i.e.

$$V_{t=+\infty}\left[\mathbf{x}_0, m, \kappa_0, \{\kappa_j = \kappa\}_{j \in [\![1,t]\!]}\right] > 0, \tag{43}$$

with

$$V_{t=+\infty}\left[\mathbf{x}_0, m, \kappa_0, \{\kappa_j = \kappa\}_{j \in [\![1,t]\!]}\right] = -\frac{1-m}{2}\log\left(\frac{1-m}{2}\right) - \frac{1+m}{2}\log\left(\frac{1+m}{2}\right) \tag{44}$$
$$+ \alpha \int dw_{t-1} P^\kappa_{\text{no-mem. state}}[w_{t-1}] \log\left[H\left(\kappa, mw_{t-1}, 1-m^2\right)\right].$$

If $\kappa$ is large enough ($\alpha$ being fixed) this condition can be verified. In this case, our formalism predicts that any dynamics initialized in a configuration $\mathbf{x}_0$ with $P^\kappa_{\text{no-mem. state}}[.]$ as its distribution of margins $\{w^\mu = \xi^\mu \cdot \mathbf{x}_0/\sqrt{N}\}$ will asymptotically decorrelate from its initialization. Thus, we will always refer to this ensemble of solution as the "delocalized no-memory states".

In the following, the lower margin threshold required to verify this condition (with $\alpha$ being fixed) will be referred as $\kappa^\alpha_{\text{no-mem. state}}$ -and respectively $\alpha^\kappa_{\text{no-mem. state}}$ (with $\kappa$ being fixed)-.

## 5.3 Remaining correlated with the planted signal ($\kappa < \kappa^\alpha_{\text{no-mem. state}}$)

In this section we will focus on the case where we do not release enough the threshold $\kappa$, i.e. $\kappa < \kappa^\alpha_{\text{no-mem. state}}$. It is a situation for which the system remains correlated with its initialization.

In Fig. 4, we display overlap curves for the quench dynamics described above with several system sizes, $N = \{3 \times 10^3, 7 \times 10^3, 1 \times 10^4, 2 \times 10^4, 4 \times 10^4\}$. We have set the margin threshold to $\kappa = 0.75$ and averaged the dynamics for each system size over 10 realizations of the disorder. Starting with the right panel, we simply plotted the overlap of the system at time t (natural time-scale) with its initial configuration. We can observe that this quantity appears to converge to a plateau in the long-time limit, and the height of this plateau depends on $N$. In particular, the bigger the system is the closer this plateau gets to 1. In other words, this means that the dynamical system remains closer and closer to its initialization as the number of dimensions increases.

In fact, the landscape of solutions around a planted configuration have already been studied in [25]. In particular, authors analysed the Franz-Parisi potential around a fixed solution of the problem. This allowed them to count the number of typical solutions around a fixed configuration in the $N \to +\infty$ limit. One of the main results is that a Monte-Carlo initialized in $\mathbf{x}_0$ should be able to decorrelate up to an extensive Hamming distance from its initialization in the long-time limit (given that the number of dimension goes to infinity first), i.e.

$\lim_{t\to+\infty}\mathbf{x}_t\cdot\mathbf{x}_0/N = m'$ and $m' < 1$. However, this prediction is not consistent with our Monte-Carlo simulations or our chain computation. This discrepancy comes from the fact that the Franz-Parisi potential counts solutions whether or not they are isolated, i.e. disregarding any dynamical accessibility. In most models this consideration is superfluous. However, as our simulations show, it is a core problem in the symmetric binary perceptron.

Coming back on Fig. 4, the left panel displays the same simulations but this time with the rescaled time-scale. We recall that this time-scale corresponds to the number of spin flips that have been accepted. With this framing we can over-impose the predictions from the chain computation. In particular, we added the decorrelation profile predicted by the no-memory chain (dashed black line) and highlighted the point for which its update stops (colored spot) -when the local entropy becomes negative-. We want to emphasize that the stopping point in the no-memory chain is $N$-dependent since we have set the overlap $m = 1-2/N$. Thus, both the Monte-Carlo dynamics and our theoretical predictions are dependent on the size of the system.

With the rescaled time-scale we can highlight that the Monte-Carlo simulations undergo two dynamical regimes. A first one that follows the chain prediction: the system takes $\mathcal{O}(N)$ natural time steps to decorrelate as a no-memory system. In this regime we observe that it accepts roughly from $0.1N$ to $0.3N$ spin flips, depending on the size of the system. Then, while the chain computation predicts that no more spin flips should be accepted, the Monte-Carlo enters a second regime where it continues to find possible spin flips. This second regime however cannot be obtained if the dynamics is ran for an usual $\mathcal{O}(N)$ total number of natural time steps. Indeed, we can observe that while all simulations were run for a maximum of $1500N$ natural time steps (as shown in the right panel), the left panel shows that the second regime (in which more spin flips than expected are accepted) shrinks as $N$ increases.

Showing further agreements between the simulations and the no-memory chain predictions, we plot in Fig. 5 quenches with an adaptive value for $\kappa$. In this case $\kappa$ is fixed such that the chain formalism predicts for each size $N$ a stop at exactly $\mathbf{x}_t\cdot\mathbf{x}_0/N = 0.9$. Again, we have averaged the dynamics over 10 realizations of the disorder. On the right-hand panel, as the size of the system is increased we see that the correlation profile gets closer and closer to a sudden drop at $\mathbf{x}_t\cdot\mathbf{x}_0/N = 0.9$ followed by a plateau. The left-panel displays even more clearly this behavior, where the fraction of spin being flipped shrinks to the predictions of the no-memory chain. Again, we have a fast regime -where the system decorrelates according to our theoretical predictions- and then a second slow regime -where the system further decorrelates but requires more than $\mathcal{O}(N)$ natural time steps to achieve it-.

As a last note, we observe in the left panel of Fig. 5 that increasing the system size lowers the correlation function more and more. This can be interpreted straightforwardly. Each time the Monte-Carlo dynamics performs a spin-flip, there is a finite probability that it performs a backward move. In the chain formalism, this means that instead of going from $\mathbf{x}_t$ to $\mathbf{x}_{t+1}$ we go from $\mathbf{x}_t$ to $\mathbf{x}_{t-1}$. As the system size increases, this move becomes more and more negligible and the correlation function collapses on the prediction given by a no-memory chain (for which only forward moves are accepted).

In a nutshell, after releasing the margin to $\kappa$ the Monte-Carlo dynamics explore connected states corresponding to a chain of solutions with no memory with its past. This first regime stops after $\mathcal{O}(N)$ natural time steps, as no solutions with such a decorrelation profile can be found anymore. However, the dynamics can further decorrelates but it now takes $\omega(N)$ natural time steps -i.e. $t \gg N$- for finding a correct set of spins to flip. For now we will leave aside this second regime as we lack the correct theoretical frame to analyse it. We will go back to it in Sec 6.

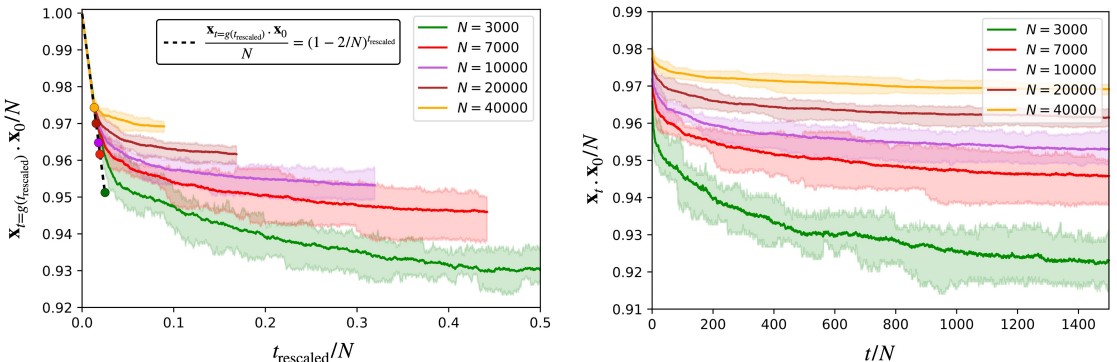

Figure 4: We plot the overlap of the system with its initial configuration $\mathbf{x}_0$ as it decorrelates via a quench Monte-Carlo dynamics. The quench is performed for $\alpha = 0.5$, $\kappa = 0.75$ and several sizes of the system $N$. For each size the correlation curve is averaged over 10 realizations of both the dynamics and disorder. It is also attached to a matching color shade which highlight the maximum and minimum overlap values attained over these 10 realizations. On the right, the correlation curves are plotted with the natural time-scale. This means that every time the Monte-Carlo algorithm performs a spin-flip -accepted or not- time is incremented by one. On the left, the same correlation curves are plotted with the rescaled time-scale. In this case, time is incremented by one only when a spin-flip is accepted. In this left panel we added the correlation curve predicted by the no-memory chain. Each colored dot represented the point where the chain stops when setting $m = 1 - 2/N$.

## 5.4 Escaping in delocalized no-memory states ($\kappa > \kappa_{\text{no-mem. state}}^{\alpha}$)

In this section we focus on the case where the quench is done for $\kappa > \kappa_{\text{no-mem. state}}^{\alpha}$. This means that we have released the threshold enough for the system to escape from the planted configuration with $\mathcal{O}(N)$ natural time steps. To show this, we plot in Fig. 6 the evolution of the correlation function $C(t, t')_{t'>t} = \mathbf{x}_t \cdot \mathbf{x}_{t'}/N$ for a quench in this regime. On the right-hand side the decorrelation regime appears clearly as we have $\lim_{|t-t'|/N \to +\infty} C(t, t') = 0$ with the natural time-scale. We can also notice a slowdown in this dynamics as the correlation functions decay more and more slowly as time $t$ increases. It is important to emphasize that once the correlation functions are plotted with the rescaled time-axis (left panel) this aging effect disappear. With this rescaling all correlation functions collapse on the theoretical predictions from the no-memory chain. This means that the system decorrelates through the connected no-memory configurations, the slowdown in the dynamics being simply due to fewer of these configurations being available as time flows. Indeed, if the number of available solutions decreases, more spin-flip trials will be required for a Monte-Carlo routine to find one of these solutions. This behavior also appears clearly with our theoretical framework as we observe that $V_{t=1}[\mathbf{x}_0, m, \{\kappa_j\}_{j\in[\![1,t]\!]}] > V_{t=2}[\mathbf{x}_0, m, \{\kappa_j\}_{j\in[\![1,t]\!]}] > \cdots > V_t[\mathbf{x}_0, m, \{\kappa_j\}_{j\in[\![1,t]\!]}]$. In this case, the decrease in the potential straightforwardly quantifies the diminution in number of available no-memory solutions.

In Fig. 7 we display the distribution of interactions $w^{\mu} = \xi^{\mu} \cdot \mathbf{x}/\sqrt{N}$ for a single quenched dynamics. It is obtained after averaging over 9000 configurations $\mathbf{x}$ explored by the Monte-Carlo algorithm. On both panel we have added the truncated Gaussian envelop -which corresponds to the distribution of interactions for the typical states verifying $\mathbf{x}_0 \cdot \mathbf{x}/N = 0$- and the distribution $P_{\text{no-mem. state}}^{\kappa}[.]$ -which corresponds to the delocalized connected states with no memory-. The right panel displays the same distribution as the left panel but zoomed around one of its edge, in this case at $w^{\mu} = \kappa = 1.9$. First, we can note that the discrepancy be-

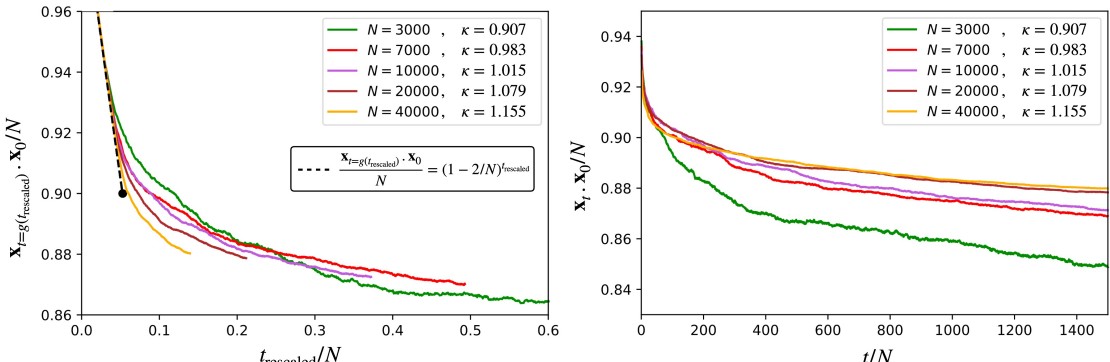

Figure 5: We plot the overlap of the system with its initial configuration $\mathbf{x}_0$ as it decorrelates via a quench Monte-Carlo dynamics. The quench is performed for $\alpha = 0.5$ and several sizes of the system $N$. For each size the correlation curve is averaged over 10 realizations of both the dynamics and disorder. We set $\kappa$ such that the decorrelation is predicted to stop at exactly $\mathbf{x}_t \cdot \mathbf{x}_0/N = 0.9$ by the no-memory chain computation. We recall that this prediction depends on the size of the system as we have $m = 1-2/N$. On the right, the correlation curves are plotted with the natural time-scale. On the left, the same correlation curves are plotted with the rescaled time-scale. In this left panel we added in dashed black the correlation curve predicted by the no-memory chain.

tween the truncated Gaussian envelop and $P^\kappa_{\text{no-mem. state}}[.]$ appears only around the edges, i.e. $|w^\mu| \approx \kappa$. More specifically, this discrepancy depends on the overlap $m$ set for the no-memory chain computation as it is observed for $|w^\mu| - \kappa \sim \sqrt{1-m^2}$. This means in particular that the difference between the two distributions vanishes when we set $m \to 0$. However, this does not mean that this edge effect is negligible in our formalism. In fact, when computing the potential $V_{t=+\infty}[\mathbf{x}_0, m, \kappa_0, \{\kappa_j = \kappa\}_{j \in [\![1,t]\!]}]$ from Eq. (44) the function appearing in the integral takes non-negligible values only within this range $|w^\mu| - \kappa \sim \sqrt{1-m^2}$. This means in other words that this edge effect is crucial for properly counting and describing the connected states with no memory. Comparing these two distribution with our simulation, we observe that the solutions resulting from the Monte-Carlo procedure follow the distribution predicted by the no-memory chain. More strikingly, it follows the distribution for which we have set $m = 1-2/N$. This feature is a good a posteriori check for setting $m = 1-2/N$. Indeed, we mentioned just above that the distribution decay in $P^\kappa_{\text{no-mem. state}}[.]$ around the edges $|w^\mu| \approx \kappa$ depends on the value of $m$. If we would have set $1-m = 2f(N)/N$, with $f(N) \gg 1$, this decay would be greater and would not match the Monte-Carlo simulations.

Finally, in Fig. 8 we plot the phase diagram for the presence of delocalized connected states (with no memory) with fixed $\alpha = 0.5$. We recall that $\kappa^\alpha_{\text{no-mem. state}}$ corresponds to the margin for which $V_{t=+\infty}[\mathbf{x}_0, m, \kappa_0, \{\kappa_j = \kappa_1\}_{j \in [\![1,t]\!]}] = 0$ (with $m = 1-2/N$). If $\kappa > \kappa^\alpha_{\text{no-mem. state}}$, the potential $V_{t=+\infty}[.,.,.,.]$ is positive and the steady-state to which any no-memory chain converges is not isolated and thus can delocalize forever. For $\kappa < \kappa^\alpha_{\text{no-mem. state}}$ this steady state is isolated, which means that all no-memory chain will stop after a certain number of iterations. Going back on the phase diagram itself, it shows that while the system size $N$ diverges (which implies $m \to 1$) the critical margin $\kappa^\alpha_{\text{no-mem. state}}$ diverges as $\sqrt{0.91 \ln N}$. It is important to note that the region for which delocalized no-memory states can be observed remains non-vanishing for $N \to +\infty$. To understand this we can note that margin threshold $\kappa_{\text{trivial}}$ of a random point $\mathbf{x}$ on the hypercube verifies the scaling law $\kappa_{\text{trivial}} \sim \sqrt{2 \ln N}$ [37]. Thus, we have that $\kappa^\alpha_{\text{no-mem. state}}$ remains distinct from $\kappa_{\text{trivial}}$ (even for large system sizes).

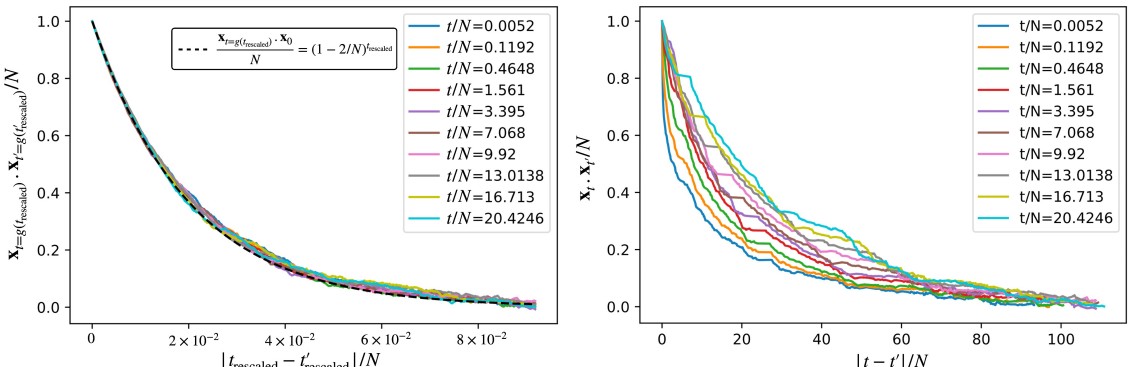

Figure 6: We plot the two-time correlation function of a quench Monte-Carlo dynamics. The system is tuned to $\alpha = 0.5$, $\kappa = 1.9$, $N = 5000$. Both panels display the same correlation curve but with different time-axis. On the right, the time-axis is the natural time-scale -for which time is incremented by one for every spin-flip trial-. On the left, the time-axis corresponds to the rescaled one -for which time is incremented only when a spin-flip is accepted-. In the left panel we also added in dashed black the decorrelation profile corresponding to the no-memory chain.

Finally, we should also note that tuning $\alpha$ does not change qualitatively the phase diagram. The case we display ($\alpha = 0.5$) is generic.

## 6 Going beyond the no-memory Ansatz

As mentioned in Sec. 5.3, when we set $\kappa < \kappa^\alpha_{\text{no}-\text{mem. state}}$ the Monte-Carlo procedure is capable of decorrelating more than what is predicted by the no-memory Ansatz. We have also shown that this second regime of decorrelation requires to run the quench for $\omega(N)$ natural time steps -i.e $t \gg N$-. Put differently, this means that there is a set of subdominant connected solutions that allow escape further from the planted configuration than the no-memory states.

In fact, for a certain range of system size $N$ and tuning parameters $\{\alpha, \kappa\}$, we can observe that the algorithm ends up totally decorrelating from the planting signal, if again it is executed for a sufficiently long time period. As an example, if we set $N = 2000$, $\alpha = 0.5$ and $\kappa = 0.91$ ($\kappa^\alpha_{\text{no}-\text{mem. state}} \approx 1.13$) we can observe $\mathbf{x}_t \cdot \mathbf{x}_0/N \approx 0$ for $t/N \geq 4 \times 10^4$. In Fig. 9 we plot the distribution of interactions $w^\mu = \xi^\mu \cdot \mathbf{x}/\sqrt{N}$ for a single quenched dynamics. It is obtained after averaging over 30000 configurations $\mathbf{x}$ explored by the Monte-Carlo algorithm after full decorrelation. As in Fig.7, we added on both panel the truncated Gaussian envelop -which corresponds to the distribution of interactions for the typical states verifying $\mathbf{x}_0 \cdot \mathbf{x}/N = 0$- and the distribution $P^\kappa_{\text{no}-\text{mem. state}}[.]$ -which corresponds to the delocalized connected states with no memory-. The more striking output from this is the decay in the density of interactions around the edges $|w^\mu| \approx \kappa$. Indeed, we observe that this decay is more pronounced than the no-memory state prediction.

At this point it is important to recall one key feature which follows from the no-memory chain computation. For the no-memory delocalized state, we have briefly mentioned that the decay at the edges of its density of interactions is directly linked to the parameter $m$ we have chosen. Indeed, this decay appears within a range $|\omega^\mu| - \kappa \sim \sqrt{1 - m^2}$. We can try to consider that this equivalence -between the edge-decay in the distribution of interactions and the typical correlation length within a chain of connected solution- is general. More particularly, if our Monte-Carlo algorithm probes connected states with a greater distribution decay than the

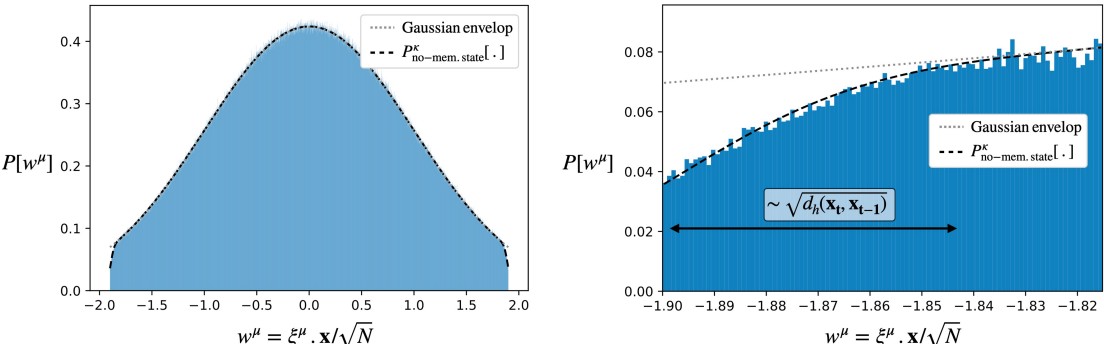

Figure 7: We plot the distribution of interactions $w^\mu = \xi^\mu \cdot \mathbf{x}/\sqrt{N}$ obtained with a Monte-Carlo quench. The dynamics is performed for $\alpha = 0.5$, $\kappa = 1.9$ and $N = 5000$. We derived the histogram of interactions after sampling $N_{\text{sampling}} = 9000$ different configurations throughout the dynamics. In dotted grey we have added the truncated (between $-\kappa$ and $\kappa$) Gaussian distribution. It corresponds to the distribution of interactions for the typical solutions in the binary perceptron. The distribution of interactions predicted by the no-memory chain -with $m = 1 - 2/N$- is plotted dashed black. While the left panel displays the distribution in its entirety, the right one zooms in on one edge of the distribution -$|w^\mu| \approx \kappa$-.

no-memory steady state it is because the typical correlation length between the solutions it probes is also greater. With our formalism, increasing the typical correlation length between connected states means than we have to give up the no-memory Ansatz and start analysing chains with a correlation profile involving memory of previous links, i.e. $m_{|t-t'|} > m^{|t-t'|}$. Consequently, the task we propose to do in the rest of this paper is to implement memory in our chain computation -see Eq. (16,17)-.

We recall that the potential $V_t[\mathbf{x}_0, \{m_{j+1,j}\}_{j\in[\![0,t-1]\!]}, \{\kappa_j\}_{j\in[\![1,t]\!]}]$ -as it is written in Eq. (16)- cannot be computed. Indeed, it requires to perform multidimensional sums and integrals and to optimize over a large number of parameters. Therefore, so as to have a tractable computation we will have to simplify the correlations variables $m_{t,t'}$ and $\hat{m}_{t,t'}$. Keeping the case of a quench where $\kappa_{t(\neq 0)} = \kappa$, we will propose in the following a correlation pattern that allows us to fall back on an effective no-memory chain. More particularly, we will study as schematized in Fig. 10 the case of nested Markov memory chains. Starting with a configuration $\mathbf{x}_{t_0}$, the system will perform $t_1$ steps decorrelating as a no-memory chain (also known as a Markov chain) with $m_{j+1,j} = m_1$. The step $\Delta_t^2 = t_1 + 1$ then recorrelates with $\mathbf{x}_{t_0}$ by verifying the overlaps

$$\mathbb{E}_\xi[\mathbf{x}_{t_0+\Delta_t^2} \cdot \mathbf{x}_{t_0+t_1}] = m_1, \quad \text{and} \quad \mathbb{E}_\xi[\mathbf{x}_{t_0+\Delta_t^2} \cdot \mathbf{x}_{t_0}] = m_{\Delta_t^2}, \tag{45}$$

with the condition on the fields

$$\hat{m}_{t_0+\Delta_t^2,t'(\neq t_0+t_1,t_0)} = 0. \tag{46}$$

This overlaps and fields structure allows for integrating the intermediate $t_1$ steps in both the entropic and the energetic terms of the potential. From this follows an effective step that goes directly from $\mathbf{x}_{t_0}$ to $\mathbf{x}_{t_0+\Delta_t^2}$. Thus, we have defined a new effective no-memory chain where a "link" $\mathbf{x}_{t_0+j\Delta_t^2}$ only correlates with its effective nearest neighbors $\mathbf{x}_{t_0+(j-1)\Delta_t^2}$ and $\mathbf{x}_{t_0+(j+1)\Delta_t^2}$. Once this first building block is defined it can be iterated easily. Indeed, we can continue and consider that the system passes $t_2$ effective steps following our new Markov chain. The step $\Delta_t^3 = (t_2 + 1)\Delta_t^2$ then recorrelates with $\mathbf{x}_{t_0}$ by setting the overlaps

$$\mathbb{E}_\xi[\mathbf{x}_{t_0+\Delta_t^3} \cdot \mathbf{x}_{t_0+t_2\Delta_t^2+t_1}] = m_1, \quad \mathbb{E}_\xi[\mathbf{x}_{t_0+\Delta_t^3} \cdot \mathbf{x}_{t_0+t_2\Delta_t^2}] = m_{\Delta_t^2}, \quad \mathbb{E}_\xi[\mathbf{x}_{t_0+\Delta_t^3} \cdot \mathbf{x}_{t_0}] = m_{\Delta_t^3}, \tag{47}$$

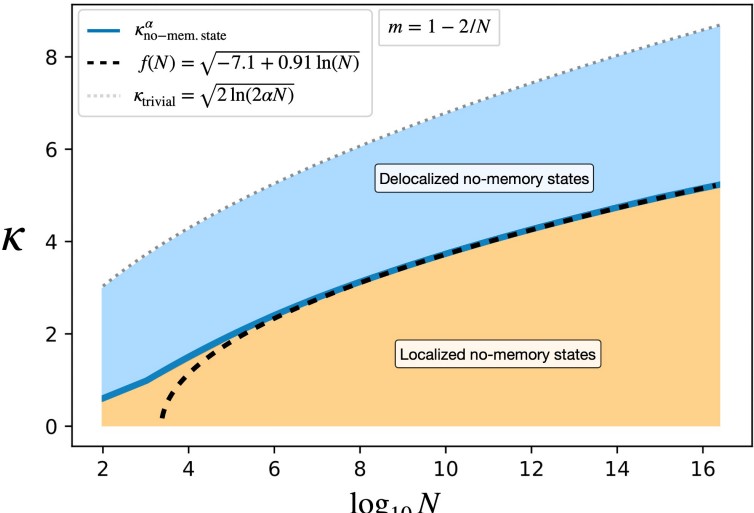

Figure 8: Phase diagram for the localization/delocalization transition of the steady-state predicted by the no-memory chain ($\alpha = 0.5$). We recall that this state is described by the distribution of interactions $P^\kappa_{\text{no-mem. state}}[.]$ and is delocalized when its potential $V_{t=+\infty}\left[\mathbf{x}_0, m, \kappa_0, \{\kappa_j = \kappa\}_{j \in [\![1,t]\!]}\right]$ is positive. Again, to compute the potential and obtain the transition line, we equated the system size and the no-memory chain overlap by setting $m = 1 - 2/N$. The value $\kappa_{\text{trivial}}$ corresponds to the margin $\max_\mu\left[\xi^\mu \cdot \mathbf{x}/\sqrt{N}\right]$ obtained for a random configuration $\mathbf{x}$ on the hypercube of dimension $N$. The derivation of this threshold can be found in [37].

with the condition on the fields

$$\hat{m}_{t_0 + \Delta_t^3, t'(\neq t_0 + t_2\Delta_t^2 + t_1, t_0 + t_2\Delta_t^2, t_0)} = 0. \tag{48}$$

Consequently, this builds a new effective Markov chain where a "link" $\mathbf{x}_{t_0 + j\Delta_t^3}$ now only couples with its neighbors $\mathbf{x}_{t_0 + (j-1)\Delta_t^3}$ and $\mathbf{x}_{t_0 + (j+1)\Delta_t^3}$. It is then straightforward to iterate this routine an arbitrary number of times. The detailed computation for building sequently the nested Markovian chain can be found in App. D. In the rest of this section we will refer at each iteration of the computation as a level of the nested Markov chain. Level one corresponds to the original no-memory chain, level two to the Markov chain with $\Delta_t^2$ steps increment, and so on and so forth.

With this correlation pattern and a total of $k_{tot}$ levels of nested Markov chains, we end up with a new effective no-memory chain where the overlap between two "links" is now $m_{\Delta_t^{k_{tot}}}(< m_1)$. In this case, the update rule for the distribution of interactions becomes

$$P^{t_0 + \Delta_t^{k_{tot}}}\left[w_{t_0 + \Delta_t^{k_{tot}}}\right] = \int dw_{t_0} T_{\kappa, k_{tot}}\left(w_{t_0 + \Delta_t^{k_{tot}}}, w_{t_0}, \{V_{k'}\}_{k' \in [\![1, k_{tot}]\!]}\right) P^{t_0}[w_{t_0}], \tag{49}$$

where the expression for $T_{\kappa, k_{tot}}$ can be found in App. D. We will label the stable distribution of this new Markov process as $P^\kappa_{k_{tot}-\text{mem. state}}[.]$. Although this state is stable for the $k_{tot}$th level of our Markovian diffusion, it is not stable regarding lower levels. In other words we have

$$P^\kappa_{k_{tot}-\text{mem. state}}[w_{t + \Delta_t^k}] \neq \int dw_t T_{\kappa, k}(w_{t + \Delta_t^k}, w_t, \{V_{k'}\}_{k' \in [\![1, k]\!]}) P^\kappa_{k_{tot}-\text{mem. state}}[w_t], \tag{50}$$

for $k \neq k_{tot}$. This will induce the chain potential $V_t^*[.,.,.,.]$ to fluctuate during the diffusion and more particularly to be $\Delta_t^{k_{tot}}$-periodic. To examine if the new stable state can delocalize

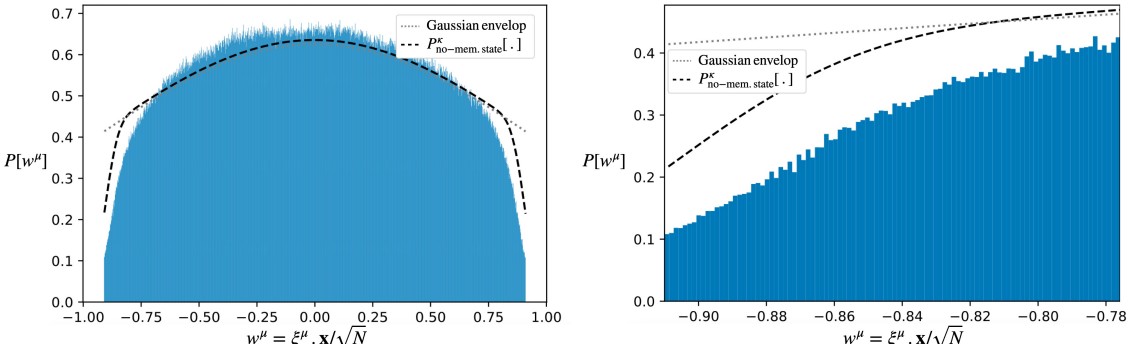

Figure 9: We plot the distribution of interactions $w^\mu = \xi^\mu \cdot \mathbf{x}/\sqrt{N}$ obtained with a Monte-Carlo quench. The dynamics is performed for $\alpha = 0.5$, $\kappa = 0.91$ and $N = 2000$. We derived the histogram of interactions after sampling $N_{\text{sampling}} = 30000$ configurations throughout the dynamics after full decorrelation, i.e. after obtaining $\mathbf{x}_t \cdot \mathbf{x}_0/N \approx 0$. In dotted grey we have added the truncated (between $-\kappa$ and $\kappa$) Gaussian distribution. It corresponds to the distribution of interactions for the typical solutions in the binary perceptron. The distribution of interactions predicted by the no-memory chain -with $m = 1 - 2/N$- is plotted dashed black. While the left panel displays the distribution in its entirety, the right one zooms in on one edge of the distribution -$|w^\mu| \approx \kappa$-.

we should in theory compute the chain potential on $\Delta_t^{k_{tot}}$ consecutive "links". In practice, we observe numerically that it is always the last step in the nested Markovian process that has the minimal potential. More concretely, if we take the example represented in Fig. 10 where we have $k_{tot} = 4$ levels of nested Markov chains starting at $t = t_0$ and finishing at $t = t_0 + \Delta_t^4$, the step where we are most limited in number of solutions to extend the chain is the one where we go from $\mathbf{x}_{t_0+\Delta_t^{k_{tot}}-1}$ to $\mathbf{x}_{t_0+\Delta_t^{k_{tot}}}$. Consequently, to determine whether this state is delocalized or not, we will simply compute the chain potential for the step $t_0 + \Delta_t^{k_{tot}}$ only. If this quantity is positive -respectively negative- then the state delocalizes -respectively remains localized-. The calculation of this potential will not be derived in this section, but the detailed steps to obtain it can be found in App. D.

Having $\kappa$ fixed, the delocalized phase ($\alpha < \alpha_{k_{tot}-\text{mem.state}}^\kappa$) corresponds to the existence of a steady-state induced by $k_{tot}$ nested Markov chains with a positive potential $V^*_{t_0+\Delta_t^{k_{tot}}}[.,.,.,.]$. While in the localized phase ($\alpha < \alpha_{k_{tot}-\text{mem.state}}^\kappa$), no steady-state has a positive potential. In App. D, the interested reader can also find the detailed computation on how the critical point $\alpha_{k_{tot}-\text{mem.state}}^\kappa$ is obtained. In short, we solve the saddle-points equations for the fields $\hat{m}$'s given by the chain potential and determine numerically the set of magnetizations and time-intervals $\left\{ m_{\Delta_t^j}, \Delta_t^j \right\}_{j \in [[2, k_{tot}]]}$ that yield a positive potential with the highest possible value of $\alpha$. In particular, while our construction implies that $t_j \in \mathbb{N}$ (and consequently $\Delta_t^j \in \mathbb{N}$) for all $j \in [[2, k_{tot}]]$, we determine the critical point after extending the computation to $t_j \in \mathbb{R}$.

In the following we will present results where we have injected only a few level of nested Markov chains with $m_1 = 0.98$. The aim is to show that our construction reproduces qualitatively the compression of the interactions distribution observed in Fig. 9 and yield connected states for $\alpha > \alpha_{\text{no-mem.state}}^\kappa$. To obtain quantitative comparisons, in particular with known algorithmic thresholds, we would have to perform this procedure with $m_1$ closer to one and with a greater number of nested Markov chains. To start with, we display in Table 1 the value of the critical point $\alpha_{k_{tot}-\text{mem.state}}^\kappa$ for $k_{\text{tot}} \in [[1, 5]]$, $\kappa = 0.9$, $m_1 = 0.98$. We obtain that increasing the total number of nested Markov chains extends the range of the parameter $\alpha$ for which a

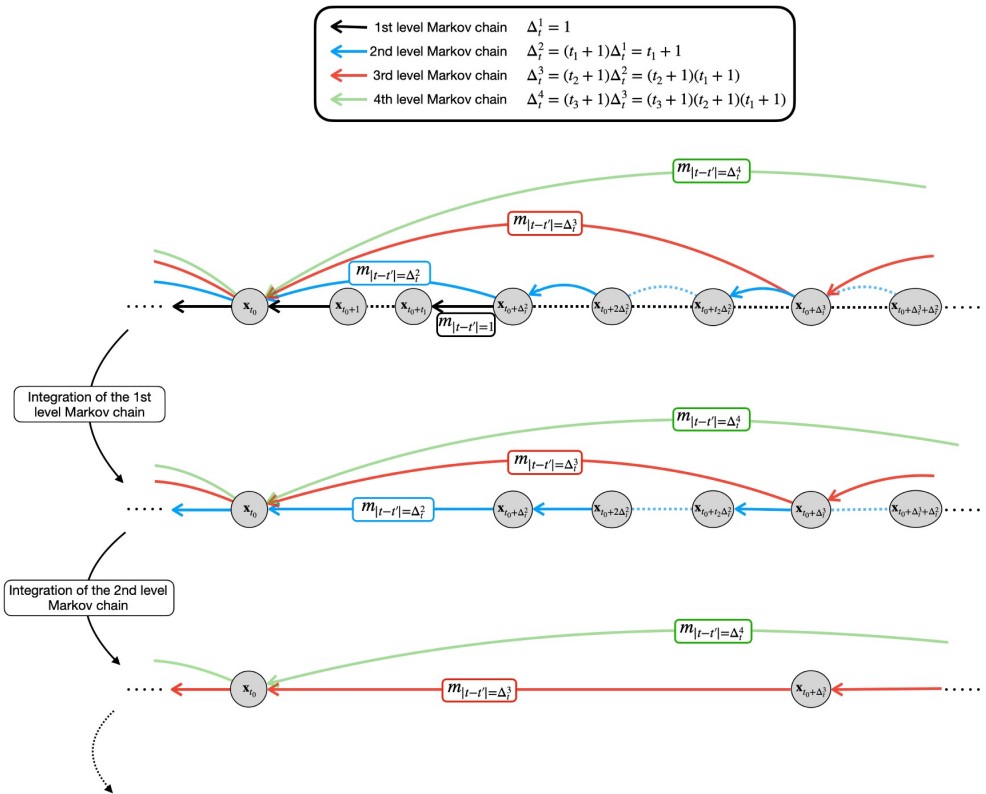

Figure 10: Schematic representation for the iterative integration of the nested Markov chain. The case we display corresponds to $k_{tot} = 4$, i.e. a total of 4 level of nested Markovian chains. We highlight with it that the iterative integration of the chain starts with the nearest-neighbor interaction (black), then continues with shortest distance interaction (blue) and so on an so forth until we obtain an effective no-memory chain with $m = m_{\Delta_t^4}$.

delocalized steady-state can be observed. We also see that each increment of $k_{tot}$ results in a smaller and smaller increase in the critical point. Moreover in Fig. 11, we plot the decorrelation profiles obtained for each optimized $k_{tot}$ nested Markov chain. In this case, increasing $k_{tot}$ allows to find delocalizable chains with greater correlation functions. By putting these two results in parallel, we directly see that we must increase the correlations between the "links" of the chain in order to build a chain of connected solutions with an increasingly high $\alpha$. For the interested reader, we display in Table 2 the order parameters $m_{\Delta_t^k}$ and $\Delta_t^k$ obtained for the optimized chains with again $k_{\text{tot}} \in [\![1, 5]\!]$, $\kappa = 0.9$, $m_1 = 0.98$.

We already mentioned the link between increasing correlations and the possibility to delocalize for $\alpha > \alpha_{\text{no}-\text{mem.state}}^{\kappa}$. Indeed, when analysing the distribution of interactions after a quench with $\kappa < \kappa_{\text{no}-\text{mem.state}}$ (Fig. 9), we highlighted that the decay at the edges $|w^\mu| \approx \kappa$ could be a clue for the increase of correlations. In Fig. 12 we plot the distribution of interactions $-P_{k_{tot}-\text{mem.state}}^{\kappa}$- for each optimized $k_{tot}$ nested Markov chain. As predicted, increasing the number of nested levels -and consequently the correlations- creates an increasingly pronounced decay at the edges of the distributions. Thus, implementing memory between the "links" in the chain allows us to mimic the contraction in the margin distribution that we observed with the Monte-Carlo dynamics.

As a final note, we can also observe a significant discrepancy appearing already between the truncated Gaussian envelop and the no-memory state distribution. This is due to our choice

Table 1: Table displaying the critical point $\alpha^{\kappa}_{k_{tot}-\text{mem. state}}$ as a function of $k_{\text{tot}}$, the total number of nested Markov chains in the memory pattern. It is obtained for $\kappa = 0.9$ and $m_1 = 0.98$ (with $\Delta^1_t = 1$).

| | $k_{tot} = 1$ (no-memory) | $k_{tot} = 2$ | $k_{tot} = 3$ | $k_{tot} = 4$ | $k_{tot} = 5$ |
|---|---|---|---|---|---|
| $\alpha^{\kappa}_{k_{tot}-\text{mem. state}}$ | 0.889 | 0.961 | 0.993 | 1.009 | 1.015 |

Table 2: Table displaying the order parameters $m_{\Delta^k_t}$ and $\Delta^k_t$ for optimized nested Markov chains with varying values of $k_{tot}$. We recall that $k_{tot}$ corresponds to the total number of levels we introduced in the memory Ansatz. For the interested reader, the optimization scheme is detailed in App. D and is independent of $\alpha$. The case we present is obtained fixing $\kappa = 0.9$ and $m_1 = 0.98$ (with $\Delta^1_t = 1$).

| | $k_{tot} = 1$ | $k_{tot} = 2$ | $k_{tot} = 3$ | $k_{tot} = 4$ | $k_{tot} = 5$ |
|---|---|---|---|---|---|
| $m_1/\Delta^1_t$ | 0.98/1 | 0.98/1 | 0.98/1 | 0.98/1 | 0.98/1 |
| $m_{\Delta^2_t}/\Delta^2_t$ | | 0.9740/2.00 | 0.9745/2.10 | 0.9747/1.64 | 0.9749/1.69 |
| $m_{\Delta^3_t}/\Delta^3_t$ | | | 0.9529/4.20 | 0.9636/3.38 | 0.9623/2.07 |
| $m_{\Delta^3_t}/\Delta^3_t$ | | | | 0.9295/6.76 | 0.9560/5.53 |
| $m_{\Delta^3_t}/\Delta^3_t$ | | | | | 0.8903/11.04 |

for $m_1$ which is far from being close to one. If this overlap was set closer to one, the two distributions would be more similar. However, we would need more nested levels to observe strong memory effects in the steady-state of the effective Markov chain.

## 7 Conclusion and discussion

In this paper we developed a formalism allowing us to probe connected solutions in the symmetric binary perceptron. Our starting point was to justify the need of such a formalism. Thus, we showed that any planted solution in this model is cursed to be isolated and that a more refined strategy is reacquired to unveil an algorithmic transition. We then detailed our approach, which in few words consists in building a chain of solutions $\{\mathbf{x}_j\}_{j\in[\![1,t]\!]}$ for which the overlap between two consecutive configurations is fixed close to one.

Under its general form the solutions chain construction is intractable both analytically and numerically. Indeed, it involves optimizing a potential over multiples variables while evaluating the multidimensional integrals it contains. Therefore, as a first attempt to describe connected solutions with our formalism, we introduced a no-memory Ansatz that allows analytically studying its resulting states. This Ansatz is so named because the chain of solutions becomes Markovian-like. With it we determined a critical threshold value $\kappa^{\alpha}_{\text{no-mem.state}} \sim \sqrt{\log(N)}$ ($\alpha$ being fixed) above which such chains are fully delocalized.

Then, focusing on quenches in $\kappa$, we compared our no-memory chain predictions with Monte-Carlo simulations. As long as we probed a regime where time scales linearly with $N$, we obtained good agreement between the two settings. In particular when no-memory states are delocalized ($\kappa > \kappa^{\alpha}_{\text{no-mem.state}}$), we could observe that the distributions of interactions $\{w^{\mu} = \xi^{\mu} \cdot \mathbf{x}\}_{\mu\in[\![1,M]\!]}$ are matching. This highlighted the presence of an edge decay in the distributions, which can be linked directly to the correlation length in the chain computation.

Moreover for $\kappa < \kappa^{\alpha}_{\text{no-mem.state}}$, we mentioned that Monte-Carlo simulations decorrelate further than predicted -conditioned that time scales more than linearly with $N$-. For this

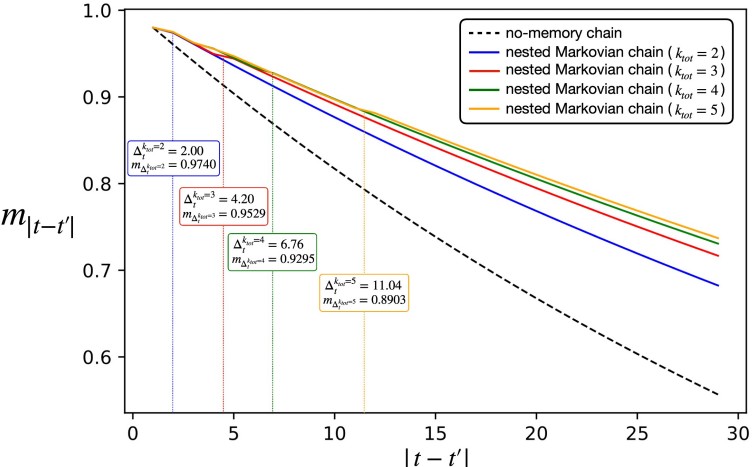

Figure 11: We plot the correlation function for each of the optimized nested Markov chains, $k_{tot} \in [[1,5]]$, $\kappa = 0.9$ and $m_1 = 0.98$ (with $\Delta_t^1 = 1$). We recall that the optimization yields, for fixed $k_{tot}$, the chain that can delocalize with the largest possible value for $\alpha$. Each stair-shape increments in the correlation curves corresponds to a new level of nested Markov chain being felt by the system. In particular, we have highlighted the very last one corresponding to the level $k_{tot}$ for each of the optimized decorrelation profiles.

regime, we showed that fully delocalized states have a distribution of interactions with a more pronounced edge decay than the one of no-memory states. Generalizing the parallel between the distribution decay and the correlation length in the solutions chain formalism, we proposed to study connected solutions with an increased correlation function. Therefore, we introduced a nested Markovian chain Ansatz that implements memory in the chain while allowing us to fall back on effective no-memory states. This approach showed that increasing the correlation length indeed led to a more pronounced edge-decay in the distribution of interactions. More interestingly, it yielded chains that delocalize for $\kappa < \kappa_{no-mem.state}^{\alpha}$.

For future work, we hope to further investigate the nested Markov chain iterative construction, and in particular to push for increasing the number of total levels $k_{tot}$. One straightforward direction would be to consider that we operate in a regime for which $m_{\Delta_t^j} \approx 1$ -with $j \in [[1, k_{tot}]]$-. In this regime we can expand all Markov-chain generator around identity, i.e.

$$T_{\kappa,k}\left(x, y, \{V_k'\}_{k' \in [[1,k]]}\right) \approx \delta(x-y) + \tilde{T}_{\kappa,k}\left(x, y, \{V_k'\}_{k' \in [[1,k]]}\right) + o\left(\tilde{T}_{\kappa,k}\left(x, y, \{V_k'\}_{k' \in [[1,k]]}\right)\right). \quad (51)$$

Implementing this linearization we should be able to simplify our iteration routine for the Markov generators and to push it to higher values of $k_{tot}$.

# Acknowledgments

We thank Florent Krzakala, Lenka Zdeborová, Riccardo Zecchina, Luca Saglietti, Enrico Malatesta, Clarissa Lauditi, Jérôme Garnier, Rémi Monasson and Guilhem Sermerjian for enlightening discussions on this problem.

**Funding information** We acknowledge funding from the Swiss National Science Foundation grants OperaGOST (grant number 200390).

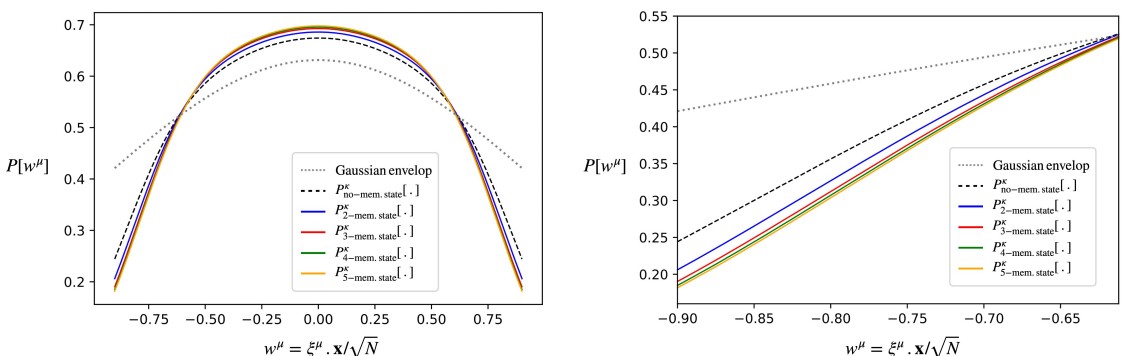

Figure 12: We plot the distributions of interactions for each optimized nested Markov chain. We recall that this optimization consists in determining the the chain's parameters (overlaps $m$, time intervals $\Delta_t$ and fields $\hat{m}$ ) that yields a positive potential $V^*_{t_0+\Delta_t^{k_{tot}}}[.,.,.,.,.]$ with maximum value for $\alpha$. In the present case we have set $k_{tot} \in [[1,5]]$, $\kappa = 0.9$ and $\underline{m}_1 = 0.98$ (with $\Delta_t^1 = 1$). In dotted grey we added the truncated Gaussian distribution, it corresponds to the typical distribution of interactions for solutions in the perceptron -with threshold $\kappa$-.

# A  Proving that the planted free energy $\phi^\kappa_{\text{planted}}[.]$ is approximately annealed

In this section we will prove that a saddle-point for the planted free energy, defined in Eq. (6), verifies $\hat{q} \ll \hat{m}$ and $q \approx m^2$ when fixing $m \approx 1$. Starting with the saddle-point equations, the partial derivatives of the free energy with respect to $q$ and $m$ are

$$\partial_q \phi^{\kappa,*}_{\text{planted}}[\mathbf{x}_0,q,\hat{q},m,\hat{m}] = 0 = \frac{\hat{q}}{2} + \alpha \int \mathcal{D}z\, dw P^\kappa[w]\left[\frac{\partial_q H\left(\kappa,mw+\sqrt{q-m^2}z,1-q\right)}{H\left(\kappa,mw+\sqrt{q-m^2}z,1-q\right)}\right], \quad \text{(A.1)}$$

$$\partial_m \phi^{\kappa,*}_{\text{planted}}[\mathbf{x}_0,q,\hat{q},m,\hat{m}] = -\hat{m} + \alpha \int \mathcal{D}z\, dw P^\kappa[w]\left[\frac{\partial_m H\left(\kappa,mw+\sqrt{q-m^2}z,1-q\right)}{H\left(\kappa,mw+\sqrt{q-m^2}z,1-q\right)}\right]. \quad \text{(A.2)}$$

We recall that the first equation shall be set to zero while the second equation should not. Indeed, in this case we optimize over $q$ whereas $m$ is fixed to a given value. The partial derivative with respect to $q$ can be rewritten as

$$\partial_q \phi^{\kappa,*}_{\text{planted}}[\mathbf{x}_0,q,\hat{q},m,\hat{m}] = \frac{\hat{q}}{2}$$

$$+ \alpha \int \mathcal{D}z\, dw P^\kappa[w]\left[\frac{\int \mathcal{D}u\left(\frac{z}{2\sqrt{q-m^2}} - \frac{u}{2\sqrt{1-q}}\right)\Theta'(\kappa-|mw+\sqrt{q-m^2}z+\sqrt{1-q}u|)}{H\left(\kappa,mw+\sqrt{q-m^2}z,1-q\right)}\right], \quad \text{(A.3)}$$

with

$$\Theta'(\kappa-|x|) = -\frac{x}{|x|}\delta(\kappa-|x|). \quad \text{(A.4)}$$

Now if we assume $q - m^2 \ll 1 - q$ we see that

$$\partial_q \phi_{\text{planted}}^{\kappa,*}[\mathbf{x}_0, q, \hat{q}, m, \hat{m}] \tag{A.5}$$

$$= \frac{\hat{q}}{2} + \alpha \int \mathcal{D}z \, dw P^\kappa[w] \left[ \frac{\int \mathcal{D}u \left( \frac{z}{2\sqrt{q-m^2}} - \frac{u}{2\sqrt{1-q}} \right) \Theta'(\kappa - |mw + \sqrt{q-m^2}z + \sqrt{1-q}u|)}{H(\kappa, mw, 1-q)} \right]$$

$$- \alpha \int \mathcal{D}z \, dw P^\kappa[w] \left[ \frac{\int \mathcal{D}u \left( \frac{z^2}{2} \partial_{\underline{y}} H\left( \underline{x} = \kappa, \underline{y} = mw, \underline{z} = 1-q \right) \right) \Theta'(\kappa - |mw + \sqrt{1-q}u|)}{H^2(\kappa, mw, 1-q)} \right]$$

$$+ \mathcal{O}\left( \sqrt{q-m^2} \right),$$

$$= \frac{\hat{q}}{2} + \alpha \int dw P^\kappa[w] \left[ \frac{\int \mathcal{D}u \mathcal{D}z \left( \frac{z}{2\sqrt{q-m^2}} - \frac{u}{2\sqrt{1-q}} \right) \Theta'(\kappa - |mw + \sqrt{q-m^2}z + \sqrt{1-q}u|)}{H(\kappa, mw, 1-q)} \right]$$

$$- \frac{\alpha}{2} \int dw \frac{P^\kappa[w]}{H(\kappa, mw, 1-q)} + \mathcal{O}\left( \sqrt{q-m^2} \right),$$

and it it easy to show with an integration by part that

$$\frac{\int \mathcal{D}u \mathcal{D}z \left( \frac{z}{2\sqrt{q-m^2}} - \frac{u}{2\sqrt{1-q}} \right) \Theta'(\kappa - |mw + \sqrt{q-m^2}z + \sqrt{1-q}u|)}{H(\kappa, mw, 1-q)} = 0. \tag{A.6}$$

This finally yields

$$\hat{q} \approx \alpha \int dw \frac{P^\kappa[w]}{H(\kappa, mw, 1-q)} + \mathcal{O}\left( \sqrt{q-m^2} \right). \tag{A.7}$$

Now if we focus on the partial derivatives with respect to $\hat{q}$ and $\hat{m}$ we obtain

$$q = \int \mathcal{D}z \tanh^2\left[ \hat{m} + \sqrt{\hat{q}}z \right] \underset{\hat{q} \ll \hat{m}}{=} \tanh^2[\hat{m}], \tag{A.8}$$

$$m = \int \mathcal{D}z \tanh\left[ \hat{m} + \sqrt{\hat{q}}z \right] \underset{\hat{q} \ll \hat{m}}{=} \tanh[\hat{m}], \tag{A.9}$$

or, rewritten in another fashion,

$$q \underset{\hat{q} \ll \hat{m}}{=} m^2, \tag{A.10}$$

$$\hat{m} \underset{\hat{q} \ll \hat{m}}{=} \operatorname{artanh}[m]. \tag{A.11}$$

In the case where $m \approx 1$ it follows from Eq. (A.11) that $\hat{m}$ diverges while Eq. (A.7) implies that $\hat{q}$ remains finite. Therefore, we have closed our saddle-point equations self-consistently for $m \approx 1$ and proved that it verifies the annealed Ansatz ($q = m^2$ and $\hat{q} \ll \hat{m}$).

When plugging the annealed Ansatz in the free energy we obtain the simplification

$$\phi_{\text{planted}}^{\kappa,\text{annealed}}[\mathbf{x}_0, m] = -\frac{1-m}{2} \log\left( \frac{1-m}{2} \right) - \frac{1+m}{2} \log\left( \frac{1+m}{2} \right) \tag{A.12}$$

$$+ \alpha \int dw P^\kappa[w] \log\left[ H(\kappa, mw, 1-m^2) \right].$$

This equation can be even further simplified when adding $m \approx 1$, indeed we can derive

$$\phi_{\text{planted}}^{\kappa,\text{annealed}}[\mathbf{x}_0, m]\Big|_{m \approx 1} \tag{A.13}$$

$$\approx -\frac{1-m}{2} \log\left(\frac{1-m}{2}\right) + \alpha \int_{-\kappa}^{\kappa} dw P^{\kappa}[w] \log\left[H(\kappa, mw, 1-m^2)\right]$$

$$\approx -\frac{1-m}{2} \log\left(\frac{1-m}{2}\right) + \alpha \int_{0}^{\kappa} dw P^{\kappa}[w] \log\left[\frac{1 + \text{erf}\left[\frac{\kappa - mw}{\sqrt{2(1-m^2)}}\right]}{2}\right]$$

$$+ \alpha \int_{-\kappa}^{0} dw P^{\kappa}[w] \log\left[\frac{1 + \text{erf}\left[\frac{\kappa + mw}{\sqrt{2(1-m^2)}}\right]}{2}\right]$$

$$\approx -\frac{1-m}{2} \log\left(\frac{1-m}{2}\right) + \frac{\alpha \sqrt{2(1-m^2)}}{m} \int_{\frac{\kappa(1-m)}{\sqrt{2(1-m^2)}}}^{\frac{\kappa}{\sqrt{2(1-m^2)}}} dB\, P^{\kappa}\left[\frac{\kappa - B\sqrt{2(1-m^2)}}{m}\right] \log\left[\frac{1 + \text{erf}[B]}{2}\right]$$

$$+ \frac{\alpha \sqrt{2(1-m^2)}}{m} \int_{\frac{-\kappa(1-m)}{\sqrt{2(1-m^2)}}}^{\frac{\kappa}{\sqrt{2(1-m^2)}}} dB\, P^{\kappa}\left[\frac{-\kappa + B\sqrt{2(1-m^2)}}{m}\right] \log\left[\frac{1 + \text{erf}[B]}{2}\right]$$

$$\approx -\frac{1-m}{2} \log\left(\frac{1-m}{2}\right) + 2\alpha\sqrt{1-m} \int_{0}^{+\infty} dB\, \{P^{\kappa}[\kappa] + P^{\kappa}[-\kappa]\} \log\left[\frac{1 + \text{erf}[B]}{2}\right].$$

The implications following this simplification of the local free energy can be found in Sec. 2.

# B  General Ansatz for the chained equilibrium

In this section we present the detailed computation for

$$V_t\left[\mathbf{x}_0, \{m_{j,j+1}\}_{j \in [\![0,t-1]\!]}, \{\kappa_j\}_{j \in [\![1,t]\!]}\right] \tag{B.1}$$

$$= \frac{1}{N} \mathbb{E}_{\xi}\left\{ \sum_{\mathbf{x}_1} P^{\kappa_1}\left[\mathbf{x}_1 \Big| \frac{\mathbf{x}_0 \cdot \mathbf{x}_1}{N} = m_{1,0}\right] \times \cdots \times \sum_{\mathbf{x}_{t-1}} P^{\kappa_{t-1}}\left[\mathbf{x}_{t-1} \Big| \frac{\mathbf{x}_{t-2} \cdot \mathbf{x}_{t-1}}{N} = m_{t-1,t-2}\right]\right.$$

$$\left. \times \log\left[\sum_{\substack{\mathbf{x}_t \in \Sigma^N \\ \text{s.t. } \frac{\mathbf{x}_t \cdot \mathbf{x}_{t-1}}{N} = m_{t,t-1}}} e^{\sum_{\mu=1}^{M} \log\left[\Theta(\kappa^t - |\xi^\mu \cdot \mathbf{x}_t|)\right]}\right]\right\}.$$

For simplicity in the following we will drop the variables of this potential and simply write $V_t$. This potential can be rewritten as

$$V_t = \frac{1}{N} \mathbb{E}_{\xi}\left\{ \sum_{\mathbf{x}_1} \int d\hat{m}_{1,0} \frac{e^{\sum_\mu \log[\Theta(\kappa_1 - |\xi^\mu \cdot \mathbf{x}_1|)] + \hat{m}_{1,0}(\mathbf{x}_1 \cdot \mathbf{x}_0 - N m_{1,0})}}{Z^{\kappa_1}[\mathbf{x}_0, m_{1,0}]}\right. \tag{B.2}$$

$$\times \cdots$$

$$\times \sum_{\mathbf{x}_{t-1}} \int d\hat{m}_{t-1,t-2} \frac{e^{\sum_\mu \log[\Theta(\kappa_{t-1} - |\xi^\mu \cdot \mathbf{x}_{t-1}|)] + \hat{m}_{t-1,t-2}(\mathbf{x}_{t-1} \cdot \mathbf{x}_{t-2} - N m_{t-1,t-2})}}{Z^{\kappa_{t-1}}[\mathbf{x}_{t-2}, m_{t-1,t-2}]}$$

$$\left. \times \log\left[\sum_{\mathbf{x}_t} \int d\hat{m}_{t,t-1} e^{\sum_\mu \log[\Theta(\kappa_t - |\xi^\mu \cdot \mathbf{x}_t|)] + \hat{m}_{t,t-1}(\mathbf{x}_t \cdot \mathbf{x}_{t-1} - N m_{t,t-1})}\right]\right\}.$$

## B.1   General computation

If we introduce replica for each "link" configuration in the chain we have

$$
V_t = \lim_{\{n_j \to 0\}_{j \in [\![1,t]\!]}} \frac{1}{n_t} - \frac{1}{Nn_t} \mathbb{E}_\xi \left\{ \prod_{a_1=1}^{n_1} \sum_{\mathbf{x}_{1,a_1}} \int d\hat{m}_{1,0,a_1} e^{\sum_\mu \log\left[\Theta(\kappa_1 - |\xi^\mu \cdot \mathbf{x}_{1,a_1}|)\right] + \hat{m}_{1,0,a_1}(\mathbf{x}_{1,a_1} \cdot \mathbf{x}_0 - Nm_{1,0})} \right.
$$

$$
\times \cdots \tag{B.3}
$$

$$
\times \prod_{a_{t-1}=1}^{n_{t-1}} \sum_{\mathbf{x}_{t-1,a_{t-1}}} \int d\hat{m}_{t-1,t-2,a_{t-1}} e^{\sum_\mu \log\left[\Theta(\kappa_{t-1} - |\xi^\mu \cdot \mathbf{x_{t-1,a_{t-1}}}|)\right] + \hat{m}_{t-1,t-2,a_{t-1}}(\mathbf{x}_{t-1,a_{t-1}} \cdot \mathbf{x}_{t-2,1} - Nm_{t-1,t-2})}
$$

$$
\left. \times \prod_{a_t=1}^{n_t} \sum_{\mathbf{x}_{t,a_t}} \int d\hat{m}_{t,t-1,a_t} e^{\sum_\mu \log\left[\Theta(\kappa_t - |\xi^\mu \cdot \mathbf{x}_{t,a_t}|)\right] + \hat{m}_{t,t-1,a_t}(\mathbf{x}_{t,a_t} \cdot \mathbf{x}_{t-1,1} - Nm_{t,t-1})} \right\}
$$

$$
= \lim_{\{n_j \to 0\}_{j \in [\![1,t]\!]}} \frac{1}{n_t} - \frac{1}{Nn_t} \mathbb{E}_\xi \left\{ \prod_{a_1=1}^{n_1} \sum_{\mathbf{x}_{1,a_1}} \int d\hat{m}_{1,0,a_1} \prod_{\mu=1}^{M} \left( dw_{1,a_1}^\mu d\hat{w}_{1,a_1}^\mu \right) \right.
$$

$$
\times e^{\sum_\mu \log\left[\Theta(\kappa_1 - |w_{1,a_1}^\mu|)\right] + \hat{w}_{1,a_1}^\mu(\xi^\mu \cdot \mathbf{x}_{1,a_1} - w_{1,a_1}^\mu) + \hat{m}_{1,0,a_1}(\mathbf{x}_{1,a_1} \cdot \mathbf{x}_0 - Nm_{1,0})}
$$

$$
\times \cdots
$$

$$
\times \prod_{a_{t-1}=1}^{n_{t-1}} \sum_{\mathbf{x}_{t-1,a_{t-1}}} \int d\hat{m}_{t-1,t-2,a_{t-1}} \prod_{\mu=1}^{M} \left( dw_{t-1,a_{t-1}}^\mu d\hat{w}_{t-1,a_{t-1}}^\mu \right)
$$

$$
\times e^{\sum_\mu \log\left[\Theta(\kappa_{t-1} - |w_{t-1,a_{t-1}}^\mu|)\right] + \hat{w}_{t,a_t}^\mu(\xi^\mu \cdot \mathbf{x}_{t-1,a_{t-1}} - w_{1,a_1}^\mu) + \hat{m}_{t,t-1,a_t}(\mathbf{x}_{t-1,a_{t-1}} \cdot \mathbf{x}_{t-2,1} - Nm_{t,t-1})}
$$

$$
\times \prod_{a_t=1}^{n_t} \sum_{\mathbf{x}_{t,a_t}} \int d\hat{m}_{t,t-1,a_t} \prod_{\mu=1}^{M} \left( dw_{t,a_t}^\mu d\hat{w}_{t,a_t}^\mu \right)
$$

$$
\left. \times e^{\sum_\mu \log\left[\Theta(\kappa_t - |w_{t,a_t}^\mu|)\right] + \hat{w}_{t,a_t}^\mu(\xi^\mu \cdot \mathbf{x}_{t,a_t} - w_{t,a_t}^\mu) + \hat{m}_{t,t-1,a_t}(\mathbf{x}_{t,a_t} \cdot \mathbf{x}_{t-1,1} - Nm_{t,t-1})} \right\}.
$$

The integration over the disorder $\{\xi^\mu\}_{\mu \in [\![1,M]\!]}$ and after over the variables $\{\hat{w}_{j,a_j}^\mu\}_{\mu \in [\![1,M]\!]}$ is trivial as it simply involves Gaussian integrals. It yields

$$
V_t = \lim_{\{n_j \to 0\}_{j \in [\![1,t]\!]}} \frac{1}{n_t} - \frac{1}{Nn_t} \left[ \int dw_0 P^{\kappa_0}[w_0] \prod_{j=1}^{t} \prod_{a_j=1}^{n_j} \left( \int dw_{j,a_j} e^{\log\left[\Theta(\kappa_j - |w_{j,a_j}|)\right]} \right) e^{\frac{-\mathbf{w}^\top \Sigma(\mathbf{x})\mathbf{w}}{2}} \right]^{M}
$$

$$
\times \left[ \sum_{\mathbf{x}_0 \in \Sigma^N} \frac{1}{2^N} \prod_{j=1}^{t} \prod_{a_j=1}^{n_j} \left( \sum_{\mathbf{x}_{j,a_j} \in \Sigma^N} \int d\hat{m}_{j,j-1,a_j} e^{\hat{m}_{j,a_j}(\mathbf{x}_{j,a_j} \cdot \mathbf{x}_{j-1,1} - Nm_{j,j-1})} \right) \right],
$$

$$
\tag{B.4}
$$

with

$$
[\Sigma(\mathbf{x})^{-1}]_{j,a_j,j',a_{j'}} = \mathbf{x}_{j,a_j} \cdot \mathbf{x}_{j',a_{j'}}, \quad \text{and} \quad \mathbf{w}_{j,a_j} = w_{j,a_j}. \tag{B.5}
$$

Finally, we introduce the overlap and field matrices, respectively $\mathbf{m}_{j,a_j,j',a_{j'}} = \mathbb{E}_\xi[\mathbf{x}_{j,a_j} \cdot \mathbf{x}_{j',a_{j'}}]$ and $\hat{\mathbf{m}}_{j,a_j,j',a_{j'}} = \hat{m}_{j,a_j,j',a_{j'}}$. The potential then reads

$$V_t = \lim_{\{n_j \to 0\}_{j \in [\![1,t]\!]}} \frac{1}{n_t} \tag{B.6}$$

$$-\frac{1}{Nn_t} \operatorname*{opt}_{\mathbf{m}_{j' \neq j-1, a_{j'} \neq 1}} \left\{ \left[ \int dw_0 P^{\kappa_0}[w_0] \prod_{j=1}^{t} \prod_{a_j=1}^{n_j} \left( \int dw_{j,a_j} e^{\log\left[\Theta(\kappa_j - |w_{j,a_j}|)\right]} \right) e^{\frac{-\mathbf{w}^\top \Sigma(\mathbf{m})\mathbf{w}}{2}} \right]^M \right.$$

$$\left. \times \left[ \sum_{x_0 = \pm 1} \frac{1}{2} \prod_{j=1}^{t} \prod_{a_j=1}^{n_j} \left( \sum_{x_{j,a_j} = \pm 1} \int \prod_{j'=1}^{j-1} \prod_{a_{j'}=1}^{n_{j'}} d\hat{m}_{j,j',a_j,a_{j'}} e^{\hat{\mathbf{m}}_{j,a_j,j',a_{j'}}(x_{j,a_j}x_{j',a_{j'}} - N\mathbf{m}_{j,j-1})} \right) \right]^N \right\},$$

and $\Sigma(\mathbf{m})^{-1} = \mathbf{m}$.

## B.2 Proving that the replica Ansatz for $V_t[.]$ is annealed

In App. A we showed that $V_{t=1}[\mathbf{x}_0, m_{1,0}, \kappa_1] = \phi^{\kappa=\kappa_1}_{\text{planted}}[\mathbf{x}_0, m = m_{1,0}]$ can be determined with an annealed replica Ansatz when $m_{1,0} \approx 1$. Indeed, the planted and chain settings are equivalent when considering the first "link" in the chain. In the following, we will continue the reasoning by proving that the second "link", namely $\mathbf{x}_2$, also verifies an annealed geometry for $m_{2,1} \approx 1$.

A simple fashion to write a physically correct replica Ansatz is to focus first on the interaction set $\mathbf{w}$. We see from the construction of the potential $V_t$, and in particular with the term $e^{\frac{-\mathbf{w}^\top \Sigma(\mathbf{m})\mathbf{w}}{2}}$, that this set can be decomposed over a basis of independent random Gaussian processes such that we match the covariances given by $\Sigma(\mathbf{m})^{-1}$. More practically, we can write for the first "link" in the chain

$$w_0 \sim P^{\kappa_0}[.], \tag{B.7}$$

$$w_1^{a_1} = m_{1,0} w_0^1 + \sqrt{1 - m_{1,0}^2}\, u_1^{a_1}, \quad \text{and} \quad u_1^{a_1} \sim \mathcal{N}(0,1). \tag{B.8}$$

If we then take a replica symmetric Ansatz for the second "link" we obtain

$$w_2^{a_2} = m_{2,0} w_0^{a_0} + \frac{m_{2,1} - m_{2,0}m_{1,0}}{\sqrt{1 - m_{1,0}^2}} u_1^1 + \sqrt{q - m_{2,0}^2 - \frac{(m_{2,1} - m_{2,0}m_{1,0})^2}{1 - m_{1,0}^2}}\, z_2 + \sqrt{1 - q}\, u_2^{a_2}, \tag{B.9}$$

and $\{z_2, u_2^{a_2}\} \sim \mathcal{N}(0,1)$. We can now simplify $V_{t=2}$ to manipulate a more handy formula,

$$V_{t=2} = \operatorname*{opt}_{q, m_{2,0}, \hat{q}, \hat{m}_{2,1}, \hat{m}_{2,0}} \left\{ -\frac{1-q}{2} \hat{q} - \hat{m}_{2,1} m_{2,1} - \hat{m}_{2,0} m_{2,0} \tag{B.10} \right.$$

$$+ \sum_{x_0 = \pm 1} \frac{1}{2} \sum_{x_1 = \pm 1} \frac{e^{\hat{m}_{1,0} x_0 x_1}}{2\cosh\left[\hat{m}_{1,0} x_0\right]} \log\left\{ 2\cosh\left[\hat{m}_{2,0} x_0 + \hat{m}_{2,1} x_1\right] \right\}$$

$$+ \alpha \int dw_0 P^{\kappa_0}[w_0] \frac{\int \mathcal{D}u_1 \Theta(\kappa^1 - |m_{1,0}w_0 + \sqrt{1 - m_{1,0}^2}\, u_1|)}{\int \mathcal{D}u_1^* \Theta(\kappa^1 - |m_{1,0}w_0 + \sqrt{1 - m_{1,0}^2}\, u_1^*|)}$$

$$\left. \times \int \mathcal{D}z_2 \log\left[ H\left(\kappa^2, \underline{w}, 1 - q\right) \right] \right\},$$

$$\underline{w} = m_{2,0} w_0 + \frac{m_{2,1} - m_{2,0}m_{1,0}}{\sqrt{1 - m_{1,0}^2}} u_1 + \sqrt{q - m_{2,0}^2 - \frac{(m_{2,1} - m_{2,0}m_{1,0})^2}{1 - m_{1,0}^2}}\, z_2, \tag{B.11}$$

with $\hat{m}_{1,0} = \text{arctanh}\left[m_{1,0}\right]$. As mentioned in Sec. 3, the optimization being "time-ordered" we took for $\hat{m}_{1,0}$ the value obtained with the optimization of $V_{t=1}[\mathbf{x_0}, m_{1,0}, \kappa_1]$. Again the demonstration for this saddle-point can be found in App. A. To keep light notation throughout the computation we will rewrite $V_{t=2}$ as

$$V_{t=2} = \underset{q, m_{2,0}, \hat{q}, \hat{m}_{2,1}, \hat{m}_{2,0}}{\text{opt}} \left\{ V^*[q, m_{2,1}, m_{2,0}, \hat{q}, \hat{m}_{2,1}, \hat{m}_{2,0}] \right\}, \tag{B.12}$$

with

$$V^*_{t=2}[q, m_{2,1}, m_{2,0}, \hat{q}, \hat{m}_{2,1}, \hat{m}_{2,0}] = -\frac{1-q}{2}\hat{q} - \hat{m}_{2,1} m_{2,1} - \hat{m}_{2,0} m_{2,0} \tag{B.13}$$

$$+ \sum_{x_0 = \pm 1} \frac{1}{2} \sum_{x_1 = \pm 1} \frac{e^{\hat{m}_{1,0} x_0 x_1}}{2\cosh\left[\hat{m}_{1,0} x_0\right]} \log\left\{2\cosh\left[\hat{m}_{2,0} x_0 + \hat{m}_{2,1} x_1\right]\right\}$$

$$+ \alpha \int dw_0 \mathcal{D}u_1 \mathcal{D}z_2 P^{\kappa_0, \kappa_1}[u_1, w_0] \log\left[H\left(\kappa^2, \underline{w}, 1-q\right)\right],$$

and

$$\underline{w} = m_{2,0} w_0 + \underline{m}_{2,1} u_1 + \sqrt{q - m_{2,0}^2 - \underline{m}_{2,1}^2}\, z_2, \tag{B.14}$$

$$P^{\kappa_0, \kappa_1}[u_1, w_0] = P^{\kappa_0}[w_0] \frac{\Theta(\kappa^1 - |m_{1,0} w_0 + \sqrt{1 - m_{1,0}^2}\, u_1|)}{\int \mathcal{D}u_1^* \Theta(\kappa^1 - |m_{1,0} w_0 + \sqrt{1 - m_{1,0}^2}\, u_1^*|)}, \tag{B.15}$$

$$\underline{m}_{2,1} = \frac{m_{2,1} - m_{2,0} m_{1,0}}{\sqrt{1 - m_{1,0}^2}}. \tag{B.16}$$

Then, we can repeat the computation carried out in App. A. More precisely we have

$$\partial_q V^*_{t=2} = \frac{\hat{q}}{2} + \alpha \int dw_0 \mathcal{D}u_1 \mathcal{D}z_2 P^{\kappa_0, \kappa_1}[u_1, w_0] \tag{B.17}$$

$$\times \left[ \frac{\int \mathcal{D}u_2 \left( \frac{z_2}{2\sqrt{q - m_{2,0}^2 - \underline{m}_{2,1}^2}} - \frac{u_2}{2\sqrt{1-q}} \right) \Theta'(\kappa - |\underline{w}|)}{H\left(\kappa, \underline{w}, 1-q\right)} \right].$$

Now if we assume $q - m_{2,0}^2 - \underline{m}_{2,1}^2 \ll 1 - q$ we see that

$$\partial_q V^*_{t=2} = \frac{\hat{q}}{2} + \alpha \int dw_0 \mathcal{D}u_1 \mathcal{D}z_2 P^{\kappa_0, \kappa_1}[u_1, w_0] \tag{B.18}$$

$$\times \left[ \frac{\int \mathcal{D}u_2 \left( \frac{z_2}{2\sqrt{q - m_{2,0}^2 - \underline{m}_{2,1}^2}} - \frac{u_2}{2\sqrt{1-q}} \right) \Theta'(\kappa - |w + \sqrt{1-q}\, u_2|)}{H\left(\kappa, w, 1-q\right)} \right]$$

$$- \alpha \int dw_0 \mathcal{D}u_1 \mathcal{D}z_2 P^{\kappa_0, \kappa_1}[u_1, w_0]$$

$$\times \left[ \frac{\int \mathcal{D}u_2 \left( \frac{z_2^2}{2} \partial_{\underline{y}} H\left(\underline{x} = \kappa, \underline{y} = w, \underline{z} = 1-q\right) \right) \Theta'(\kappa - |w + \sqrt{1-q}\, u_2|)}{H^2\left(\kappa, w, 1-q\right)} \right]$$

$$+ \mathcal{O}\left( \sqrt{q - m_{2,0}^2 + \underline{m}_{2,1}^2} \right),$$

with

$$w = m_{2,0}w_0 + \underline{m}_{2,1}u_1 \,. \tag{B.19}$$

As in App. A, it can be shown with an integration by part that the second term in the r.h.s. of the previous equation is null. It thus follows that the saddle-point verifies

$$\hat{q} = \alpha \int dw_0 \mathcal{D}u_1 \frac{P^{\kappa_0,\kappa_1}[u_1, w_0]}{H\left(\kappa, m_{2,0}w_0 + \underline{m}_{2,1}u_1, 1-q\right)} \,. \tag{B.20}$$

Now if we focus on the partial derivatives with respect to $\hat{q}$, $\hat{m}_{2,1}$ and $\hat{m}_{2,0}$ we obtain

$$q = \sum_{X=\pm 1} \frac{e^{\hat{m}_{1,0}X}}{2\cosh\left(\hat{m}_{1,0}\right)} \int \mathcal{D}z \tanh^2\left[\hat{m}_{2,0}X + \hat{m}_{2,1} + \sqrt{\hat{q}}z\right] \tag{B.21}$$

$$\underset{\hat{q}\ll\hat{m}}{=} \sum_{X=\pm 1} \frac{e^{\hat{m}_{1,0}X}}{2\cosh\left(\hat{m}_{1,0}\right)} \tanh^2\left[\hat{m}_{2,0}X + \hat{m}_{2,1}\right],$$

$$m_{2,1} = \sum_{X=\pm 1} \frac{e^{\hat{m}_{1,0}X}}{2\cosh\left(\hat{m}_{1,0}\right)} \int \mathcal{D}z \tanh\left[\hat{m}_{2,0}X + \hat{m}_{2,1} + \sqrt{\hat{q}}z\right] \tag{B.22}$$

$$\underset{\hat{q}\ll\hat{m}}{=} \sum_{X=\pm 1} \frac{e^{\hat{m}_{1,0}X}}{2\cosh\left(\hat{m}_{1,0}\right)} \tanh\left[\hat{m}_{2,0}X + \hat{m}_{2,1}\right],$$

$$m_{2,0} = \sum_{X=\pm 1} \frac{X e^{\hat{m}_{1,0}X}}{2\cosh\left(\hat{m}_{1,0}\right)} \int \mathcal{D}z \tanh\left[\hat{m}_{2,0}X + \hat{m}_{2,1} + \sqrt{\hat{q}}z\right] \tag{B.23}$$

$$\underset{\hat{q}\ll\hat{m}}{=} \sum_{X=\pm 1} \frac{X e^{\hat{m}_{1,0}X}}{2\cosh\left(\hat{m}_{1,0}\right)} \tanh\left[\hat{m}_{2,0}X + \hat{m}_{2,1}\right],$$

or, rewritten in another fashion,

$$q \underset{\hat{q}\ll\hat{m}}{=} m_{2,0}^2 + \underline{m}_{2,1}^2 \,, \tag{B.24}$$

$$m_{2,1} \underset{\hat{q}\ll\hat{m}}{=} \sum_{X=\pm 1} \frac{e^{\hat{m}_{1,0}X}}{2\cosh\left(\hat{m}_{1,0}\right)} \tanh\left[\hat{m}_{2,0}X + \hat{m}_{2,1}\right], \tag{B.25}$$

$$m_{2,0} \underset{\hat{q}\ll\hat{m}}{=} \sum_{X=\pm 1} \frac{X e^{\hat{m}_{1,0}X}}{2\cosh\left(\hat{m}_{1,0}\right)} \tanh\left[\hat{m}_{2,0}X + \hat{m}_{2,1}\right]. \tag{B.26}$$

In the case where $m_{2,1} \approx 1$, it follows from Eq. (B.25) that $\hat{m}_{21}$ diverges, while Eq. (B.20) implies that $\hat{q}$ remains finite. We have once again closed the equation self-consistently and showed that for $m_{21} \approx 1$ the correct replica Ansatz is the annealed one.

In a nutshell, we showed in App. A that the first "link" in the chain verifies an annealed replica Ansatz (for $m_{1,0} \approx 1$). In this appendix we continued and proved that the second "link" also follows an annealed replica Ansatz (for $m_{2,1} \approx 1$). To show that this applies for all "links" in the chain, we simply have to repeat recursively the exact same demonstration written just above. Finally, if we apply an annealed replica Ansatz for each "link" in the chain we obtain we

$$V_t\left[\mathbf{x}_0, \left\{m_{j+1,j}\right\}_{j\in[\![0,t-1]\!]}, \left\{\kappa_j\right\}_{j\in[\![1,t]\!]}\right] \tag{B.27}$$

$$= \underset{\substack{\{\hat{m}_{t,j'}\}_{0\leq j'\leq t-1} \\ \{m_{t,j'}\}_{0\leq j'\leq t-2}}}{\mathrm{opt}} \left[ \underset{\substack{\{\hat{m}_{j,j'}\}_{0\leq j'<j\leq t-1} \\ \{m_{j,j'}\}_{0\leq j'<j-1\leq t-2}}}{\mathrm{opt}^*} \left\{ V_t^*\left[\mathbf{x}_0, \left\{m_{j,j'}\right\}_{0\leq j'<j\leq t}, \left\{\hat{m}_{j,j'}\right\}_{0\leq j'<j\leq t}, \left\{\kappa_j\right\}_{j\in[\![1,t]\!]}\right]\right\}\right],$$

with

$$V_t^*\left[\mathbf{x}_0, \{m_{j,j'}\}_{0\le j'<j\le t}, \{\hat{m}_{j,j'}\}_{0\le j'<j\le t}, \{\kappa_j\}_{j\in[\![1,t]\!]}\right] = -\sum_{j'<t}\hat{m}_{t,j'}m_{t,j'} \tag{B.28}$$

$$+ \sum_{x_0,\dots,x_{t-1}=\pm 1}\frac{1}{2}\prod_{j=1}^{t-1}\frac{e^{\sum_{0\le j'<j}\hat{m}_{j,j'}x_jx_{j'}}}{2\cosh\left(\sum_{0\le j'<j}\hat{m}_{j,j'}x_{j'}\right)}\log\left[2\cosh\left(\sum_{0\le j'<t}\hat{m}_{t,j'}x_{j'}\right)\right]$$

$$+ \alpha\int\prod_{j=0}^{t-1}dw_j\,P^{\kappa_0}[w_0]\prod_{j=1}^{t-1}\frac{e^{-\frac{\sum_{0\le j'\le j}\Sigma_{j,j'}(\mathbf{m})w_jw_{j'}}{2}}\Theta(\kappa^j-|w_j|)}{\int dw_j^*e^{-\frac{\sum_{0\le j'\le j}\Sigma_{j,j'}(\mathbf{m})w_j^*w_{j'}}{2}}\Theta(\kappa^j-|w_j^*|)}$$

$$\times\log\left[\int dw_t\,e^{-\frac{\sum_{0\le j'\le t}\Sigma_{t,j'}(\mathbf{m})w_tw_{j'}}{2}}\Theta(\kappa^t-|w_t|)\right].$$

## C  Simplifying the no-memory Ansatz potential

In this section we present how to simplify the potential

$$V_t^*\left[\mathbf{x}_0, \{m_{j,j'}\}_{0\le j'<j\le t}, \{\hat{m}_{j,j'}\}_{0\le j'<j\le t}, \{\kappa_j\}_{j\in[\![1,t]\!]}\right] = -\sum_{j'<t}\hat{m}_{t,j'}m_{t,j'} \tag{C.1}$$

$$+ \sum_{x_0,\dots,x_{t-1}=\pm 1}\frac{1}{2}\prod_{j=1}^{t-1}\frac{e^{\sum_{0\le j'<j}\hat{m}_{j,j'}x_jx_{j'}}}{2\cosh\left(\sum_{0\le j'<j}\hat{m}_{j,j'}x_{j'}\right)}\log\left[2\cosh\left(\sum_{0\le j'<t}\hat{m}_{t,j'}x_{j'}\right)\right]$$

$$+ \alpha\int\prod_{j=0}^{t-1}dw_j\,P^{\kappa_0}[w_0]\prod_{j=1}^{t-1}\frac{e^{-\frac{\sum_{0\le j'\le j}\Sigma_{j,j'}(\mathbf{m})w_jw_{j'}}{2}}\Theta(\kappa^j-|w_j|)}{\int dw_j^*e^{-\frac{\sum_{0\le j'\le j}\Sigma_{j,j'}(\mathbf{m})w_j^*w_{j'}}{2}}\Theta(\kappa^j-|w_j^*|)}$$

$$\times\log\left[\int dw_t\,e^{-\frac{\sum_{0\le j'\le t}\Sigma_{t,j'}(\mathbf{m})w_tw_{j'}}{2}}\Theta(\kappa^t-|w_t|)\right].$$

We can in fact operate the standard change of variable

$$w_j = m_{j,j-1}w_{j-1} + \sqrt{1-m_{j,j-1}^2}\,u_j \tag{C.2}$$

$$= m_{j,j-1}m_{j-1,j-2}w_{j-2} + m_{j,j-1}\sqrt{1-m_{j-1,j-2}^2}\,u_{j-1} + \sqrt{1-m_{j,j-1}^2}\,u_j$$

$$= \left(\prod_{l=j'}^{j-1}m_{l+1,l}\right)w_{j'} + \sum_{l=j'+1}^{j-1}\left(\prod_{n=l}^{j-1}m_{n+1,n}\right)\sqrt{1-m_{l,l-1}^2}\,u_l + \sqrt{1-m_{j,j-1}^2}\,u_j$$

$$= \left(\prod_{l=0}^{j-1}m_{l+1,l}\right)w_0 + \sum_{l=1}^{j-1}\left(\prod_{n=l}^{j-1}m_{n+1,n}\right)\sqrt{1-m_{l,l-1}^2}\,u_l + \sqrt{1-m_{j,j-1}^2}\,u_j,$$

and obtain

$$V_t^*\left[\mathbf{x}_0, \{m_{j,j-1}\}_{1\le j\le t}, \hat{m}_{t,t-1}, \{\kappa_j\}_{j\in[\![1,t]\!]}\right] = -\hat{m}_{t,t-1}m_{t,t-1} + \log\left[2\cosh(\hat{m}_{t,t-1})\right] \tag{C.3}$$

$$+ \alpha\int dw_0\,P^{\kappa_0}[w_0]\prod_{j=1}^{t-1}\left[\int\mathcal{D}u_j\frac{\Theta\left(\kappa^j-\left|m_{j,j-1}w_{j-1}+\sqrt{1-m_{j,j-1}^2}\,u_j\right|\right)}{H\left(\kappa^j, m_{j,j-1}w_{j-1}, 1-m_{j,j-1}^2\right)}\right]$$

$$\times\log\left[H\left(\kappa^t, m_{t,t-1}w_{t-1}, 1-m_{t,t-1}^2\right)\right].$$

In fact, the energetic term can be estimated recursively. If we perform back the change of variable

$$w_j = m_{j,j-1} w_{j-1} + \sqrt{1 - m_{j,j-1}^2}\, u_j\,,\tag{C.4}$$

we can rewrite the potential as

$$V_t^*\Big[\mathbf{x}_0, \{m_{j,j-1}\}_{1\le j\le t}, \hat{m}_{t,t-1}, \{\kappa_j\}_{j\in[\![1,t]\!]}\Big] = -\hat{m}_{t,t-1} m_{t,t-1} + \log\Big[2\cosh(\hat{m}_{t,t-1})\Big]\tag{C.5}$$

$$+ \alpha \int dw_0\, P^{\kappa_0}[w_0] \prod_{j=1}^{t-1}\left[\int dw_j \frac{e^{-\frac{(w_j - m_{j,j-1} w_{j-1})^2}{2(1-m_{j,j-1}^2)}}\Theta(\kappa_j - |w_j|)}{\sqrt{2\pi(1-m_{j,j-1}^2)}\, H(\kappa_j, m_{j,j-1} w_{j-1}, 1-m_{j,j-1})}\right]$$

$$\times \log\Big[H\big(\kappa^t, m_{t,t-1} w_{t-1}, 1-m_{t,t-1}^2\big)\Big]\,.$$

The potential can now be integrated recursively, starting with the integral over $w_0$ (then $w_1$ and so on). Thus, the potential becomes

$$V_t^*\Big[\mathbf{x}_0, \{m_{j,j-1}\}_{1\le j\le t}, \hat{m}_{t,t-1}, \{\kappa_j\}_{j\in[\![1,t]\!]}\Big] = -\hat{m}_{t,t-1} m_{t,t-1} + \log\Big[2\cosh(\hat{m}_{t,t-1})\Big]\tag{C.6}$$

$$+ \alpha \int dw_k\, P^{\kappa_k}[w_k] \prod_{j=k+1}^{t-1}\left[\int dw_j \frac{e^{-\frac{(w_j - m_{j,j-1} w_{j-1})^2}{2(1-m_{j,j-1}^2)}}\Theta(\kappa_j - |w_j|)}{\sqrt{2\pi(1-m_{j,j-1}^2)}\, H(\kappa_j, m_{j,j-1} w_{j-1}, 1-m_{j,j-1})}\right]$$

$$\times \log\Big[H\big(\kappa^t, m_{t,t-1} w_{t-1}, 1-m_{t,t-1}^2\big)\Big]\,,$$

with the update rule

$$P^{\kappa_{j+1}}[w_{j+1}] = \int dw_j\, P^{\kappa_j}\big[w_j\big] \frac{e^{-\frac{w_{j+1} - m_{j+1,j} w_j}{2(1-m_{j+1,j}^2)}}\Theta(\kappa_{j+1} - |w_{j+1}|)}{\sqrt{2\pi(1-m_{j+1,j}^2)}\, H\big(\kappa_{j+1}, m_{j+1,j} w_j, 1-m_{j+1,j}^2\big)}\,.\tag{C.7}$$

Applying this recursion until the end we obtain

$$V_t^*\Big[\mathbf{x}_0, \{m_{j,j-1}\}_{1\le j\le t}, \hat{m}_{t,t-1}, \{\kappa_j\}_{j\in[\![1,t]\!]}\Big] = -\hat{m}_{t,t-1} m_{t,jt-1} + \log\Big[2\cosh(\hat{m}_{t,t-1})\Big]\tag{C.8}$$

$$+ \alpha \int dw_{t-1}\, P^{\kappa_{t-1}}[w_{t-1}] \log\Big[H\big(\kappa^t, m_{t,t-1} w_{t-1}, 1-m_{t,t-1}^2\big)\Big]\,.$$

# D  Computing the chain potential for the nested Markovian process

## D.1  Building the nested Markovian process

In this section we detail how to build iteratively the nested Markovian process. Starting with the level 2 Markov chain, we recall that the overlap structure verifies

$$\mathbb{E}_\xi[\mathbf{x}_{t_0+\Delta_t^2} \cdot \mathbf{x}_{t_0+t_1}] = m_1\,, \quad\text{and}\quad \mathbb{E}_\xi[\mathbf{x}_{t_0+\Delta_t^2} \cdot \mathbf{x}_{t_0}] = m_{\Delta_t^2}\,,\tag{D.1}$$

with the condition on the fields

$$\hat{m}_{t_0+\Delta_t^2, t'(\neq t_0+t_1, t_0)} = 0\,.\tag{D.2}$$

We will now detail how the entropic and energetic term simplify. The former is the simplest to compute, we have straightforwardly

$$
\sum_{x_{t_0},\ldots,x_{t_0+\Delta_t^2}=\pm 1} \frac{1}{2} \left[ \prod_{j=t_0+1}^{t_0+t_1} \frac{e^{\hat{m}_{j,j-1}x_j x_{j-1}}}{2\cosh(\hat{m}_{j,j-1})} \right] \frac{e^{\hat{m}_{t_0+\Delta_t^2,t_0+t_1}x_{t_0+\Delta_t^2}x_{t_0+t_1}+\hat{m}_{t_0+\Delta_t^2,0}x_{t_0+\Delta_t^2}x_{t_0}}}{2\cosh(\hat{m}_{t_0+\Delta_t^2,t_0+t_1}x_{t_0+t_1}+\hat{m}_{t_0+\Delta_t^2,0}x_{t_0})} \tag{D.3}
$$

$$
= \sum_{x_{t_0},x_{t_0+\Delta_t^2}=\pm 1} \frac{e^{\tilde{m}_{\Delta_t^2}x_{t_0}x_{t_0+\Delta_t^2}}}{2\cosh(\tilde{m}_{\Delta_t^2})},
$$

with $\tilde{m}_{\Delta_t^2} = \operatorname{artanh}(m_{\Delta_t^2})$. For the energetic term, we first have to perform the change of variables in the integrals -preserving the correct correlations-

$$
w_j = m_1 w_{j-1} + \sqrt{1-m_1^2}\, u_j, \quad \text{for} \quad j \in [\![1,\ldots,t_1]\!], \tag{D.4}
$$

$$
w_{t_1+1} = m_1 w_{t_0} + \frac{m_{\Delta_t^2} - m_1^{t_1+1}}{1-m_1^{2t_1}}(w_0 - m_1^{t_1}w_{t_1}) + \sqrt{1-m_1^2 - \frac{(m_{\Delta_t^2} - m_1^{t_1+1})^2}{1-m_1^{2t_1}}}\, u_{t_1+1}, \tag{D.5}
$$

with again $u_l \sim \mathcal{N}(0,1)$ for $l \in [\![1, t_1+1]\!]$. We will use later the general notation

$$
\underline{m}_1 = m_1, \quad \underline{m}_{k\neq 1} = \frac{m_{\Delta_t^k} - m_{\Delta_t^{k-1}}^{t_{k-1}+1}}{1-m_{\Delta_t^{k-1}}^{2t_{k-1}}}, \quad V_1 = 1-m_1^2, \quad V_{k\neq 1} = V_{k-1} + \frac{\left(m_{\Delta_t^k} - m_{\Delta_t^{k-1}}^{t_{k-1}+1}\right)^2}{1-m_{k-1}^{2t_{k-1}}}. \tag{D.6}
$$

Injecting this change of variable in the energetic term we obtain

$$
\int \prod_{j=t_0}^{t_0+\Delta_t^2} dw_j\, e^{-\frac{w\Sigma(\mathbf{m})w}{2}} \left[ \prod_{j=t_0+1}^{t_0+\Delta_t^2} \frac{\Theta(\kappa - |w_j|)}{\int dw_j^* e^{-\frac{w_j^* \sum_{j'\leq j}\Sigma(\mathbf{m})_{j',j}w_{j'}}{2}}\Theta(\kappa - |w_j^*|)} \right] \tag{D.7}
$$

$$
= \int dw_{t_0} \prod_{j=t_0+1}^{t_0+\Delta_t^2} \mathcal{D}u_j \left[ \prod_{j=t_0+1}^{t_0+t_1} \frac{\Theta\left(\kappa - |\underline{m}_1 w_{j-1} + \sqrt{V_1}u_j|\right)}{H(\kappa, \underline{m}_1 w_{j-1}, V_1)} \right]
$$

$$
\times \frac{\Theta\left(\kappa - |\underline{m}_1 w_{t_0+t_1} + \underline{m}_1(w_{t_0} - m_1^{t_1}w_{t_0+t_1}) + \sqrt{V_2}u_{t_0+\Delta_t^2}|\right)}{H[\kappa, \underline{m}_1 w_{t_0+t_1} + \underline{m}_2(w_{t_0} - m_1^{t_1}w_{t_0+t_1}), V_2]}
$$

$$
= \int \prod_{j=t_0}^{t_0+\Delta_t^2} dw_j \left[ \prod_{j=t_0+1}^{t_0+t_1} \frac{e^{-\frac{(w_j - \underline{m}_1 w_{j-1})^2}{2V_1}}\Theta(\kappa - |w_j|)}{\sqrt{2\pi V_1}\, H(\kappa, \underline{m}_1 w_{j-1}, V_1)} \right]
$$

$$
\times \frac{e^{-\frac{\left[w_{t_0+\Delta_t^2} - \underline{m}_1 w_{t_0+t_1} - \underline{m}_2\left(w_{t_0} - m_1^{t_1}w_{t_0+t_1}\right)\right]^2}{2V_2}}\Theta(\kappa - |w_{t_0+\Delta_t^2}|)}{\sqrt{2\pi V_2}H[\kappa, \underline{m}_1 w_{t_0+t_1} + \underline{m}_2(w_{t_0} - m_1^{t_1}w_{t_0+t_1}), V_2]}
$$

$$
= \int \prod_{j=t_0}^{t_0+\Delta_t^2} dw_j \left[ \prod_{j=t_0+1}^{t_0+t_1} T_{\kappa,1}(w_j, w_{j-1}, V_1) \right]
$$

$$
\times T_{\kappa,1}\left[ w_{t_0+\Delta_t^2}, w_{t_0+t_1} + \frac{\underline{m}_2}{\underline{m}_1}\left(w_{t_0} - m_1^{t_1}w_{t_0+t_1}\right), V_2 \right]
$$

$$= \int dw_{t_0} dw_{t_0+t_1} dw_{t_1+\Delta_t^2} T_{\kappa,1}{}^{t_1}(w_{t_0+t_1}, w_{t_0}, V_1)$$

$$\times T_{\kappa,1}\left[ w_{t_0+\Delta_t^2}, w_{t_0+t_1} + \frac{\underline{m_2}}{\underline{m_1}}\left( w_{t_0} - m_1^{t_1} w_{t_0+t_1} \right), V_2 \right]$$

$$= \int dw_{t_0} dw_{t_0+\Delta_t^2} T_{\kappa,2}\left( w_{t_1+1}, w_0, \{V_1, V_2\} \right),$$

with the definitions

$$T_{\kappa,1}(x, y, V) = \frac{e^{-\frac{(x-\underline{m_1}y)^2}{2V}} \Theta(\kappa - |x|)}{\sqrt{2\pi V} H(\kappa, \underline{m_1}y, V)}, \tag{D.8}$$

$$T_{\kappa,1}{}^t(x, y, V) = \int \prod_{j=1}^{t-1} dz_j T_{\kappa,1}(z_1, y, V) \left[ \prod_{j=1}^{t-2} T_{\kappa,1}(z_{j+1}, z_j, V) \right] T_{\kappa,1}(x, z_{t-1}, V), \tag{D.9}$$

and

$$T_{\kappa,2}(x, y, \{V_1, V_2\}) = \int dz\, T_{\kappa,1}{}^{t_1}(z, y, V_1) T_{\kappa,1}\left[ x, z + \frac{\underline{m_2}}{\underline{m_1}}(y - m_1^{t_1} z), V_2 \right]. \tag{D.10}$$

Having started with the memory-less Markov-chain generator $T_1(w_j, w_{j-1}, V_1)$, these computations steps simply highlight that $T_2(w_{t_0+\Delta_t^2}, w_{t_0}, \{V_k\}_{k\in[\![1,2]\!]})$ is the new Markov-chain generator. As we mentioned in Sec. 6, this construction can be further iterated with more nested Markov chains. Given that we have already integrated $k$ levels of nested Markov chains and that the system spends $t_k$ iterations in this level, the configuration $\mathbf{x}_{t_0+\Delta_t^{k+1}=t_0+(t_k+1)\Delta_t^k}$ verifies the overlaps

$$\mathbb{E}_\xi\left[ \mathbf{x}_{t_0+\Delta_t^{k+1}}.\mathbf{x}_{t_0+\Delta_t^{k+1}-\Delta_t^j} \right] = m_{\Delta_t^j}, \tag{D.11}$$

with the notation

$$\Delta_t^k = (t_{k-1}+1)\Delta_t^{k-1}, \quad \text{and} \quad \Delta_t^1 = 1. \tag{D.12}$$

More practically, $\Delta_t^k$ corresponds to the number of "links" in the chain that are passed when one iteration of the Markov-chain at level $k$ is performed. Again this profile for the overlaps allows to reduce the number of free fields variables as we have

$$\hat{m}_{t_0+\Delta_t^{k+1}, t'(\neq\{t_0+\Delta_t^{k+1}-\Delta_t^j\}_{j\in[\![1,k+1]\!]})} = 0. \tag{D.13}$$

The entropic term can be trivially simplified as

$$\sum_{x_{t_0},\dots,x_{t_0+\Delta_t^{k+1}}=\pm 1} \frac{1}{2} \prod_{j=t_0+1}^{t_0+\Delta_t^{k+1}} \frac{e^{\sum_{0\le j'<j}\hat{m}_{j,j'}x_j x_{j'}}}{2\cosh\left(\sum_{0\le j'<j}\hat{m}_{j,j'}x_{j'}\right)} = \sum_{x_{t_0},x_{t_0+\Delta_t^{k+1}}=\pm 1} \frac{e^{\tilde{m}_{\Delta_t^{k+1}}x_{t_0+\Delta_t^{k+1}}x_{t_0}}}{2\cosh(\tilde{m}_{\Delta_t^{k+1}})}, \tag{D.14}$$

with $\tilde{m}_{\Delta_t^{k+1}} = \text{arctanh}(m_{\Delta_t^{k+1}})$. Regarding the energetic term, the form of the correlations allows us to perform the change of variable

$$w_{t_0+\Delta_t^{k+1}} = \underline{m_1} w_{t_0+\Delta_t^{k+1}-\Delta_t^1} + \sum_{j=2}^{k+1} \underline{m_j}\left( w_{t_0+\Delta_{t_{k+1}}-\Delta_{t_j}} - m_{\Delta_{t_{j-1}}}^{t_{j-1}} w_{t_0+\Delta_{t_{k+1}}-\Delta_{t_{j-1}}} \right) \tag{D.15}$$

$$+ \sqrt{V_{k+1}} u_{t_0+\Delta_t^{k+1}},$$

with $u_l \sim \mathcal{N}(0,1)$ for $l \in [[t_0 + 1, t_0 + \Delta_t^{k+1}]]$. Then, it becomes simple to generalize the computation steps we performed for the first level of Markov-chain and to obtain

$$\int \prod_{j=0}^{(t_k+1)\Delta_t^k} dw_j e^{-\frac{\mathbf{w}\Sigma(\mathbf{m})\mathbf{w}}{2}} \left[ \prod_{j=1}^{(t_k+1)\Delta_t^k} \frac{\Theta(\kappa - |w_j|)}{\int dw_j^* e^{-\frac{w_j^* \sum_{j' \leq j} \Sigma(\mathbf{m})_{j',j} w_{j'}}{2}} \Theta(\kappa - |w_j^*|)} \right]$$

$$= \int dw_{t_0} dw_{t_0+\Delta_t^{k+1}} T_{\kappa,k+1} \left( w_{t_0+\Delta_t^{k+1}}, w_{t_0}, \{V_j\}_{j\in[[1,k+1]]} \right), \tag{D.16}$$

with the definitions

$$T_{\kappa,1}(x, y, V) = \frac{e^{-\frac{(x-\underline{m}_1 y)^2}{2V}} \Theta(\kappa - |x|)}{\sqrt{2\pi V} \, H(\kappa, \underline{m}_1 y, V)}, \tag{D.17}$$

$$T_{\kappa,k}^{\ t}\left(x, y, \{V_{k'}\}_{k'\in[[1,k]]}\right) = \int \prod_{j=1}^{t-1} dz_j \, T_{\kappa,k}\left(z_1, y, \{V_{k'}\}_{k'\in[[1,k]]}\right) \tag{D.18}$$

$$\times \left[ \prod_{j=1}^{t-2} T_{\kappa,k}\left(z_{j+1}, z_j, \{V_{k'}\}_{k'\in[[1,k]]}\right) \right] T_{\kappa,k}\left(x, z_{t-1}, \{V_{k'}\}_{k'\in[[1,k]]}\right),$$

and the recursion rule

$$T_{\kappa,k+1}\left(x, y, \{V_{k'}\}_{k'\in[[1,k+1]]}\right) \tag{D.19}$$

$$= \int dz \, T_{\kappa,k}^{\ t_k}\left(z, y, \{V_{k'}\}_{k'\in[[1,k]]}\right) T_{\kappa,k}\left[x, z + \frac{m_{j+1}}{\underline{m}_j}\left(y - m_j^{t_j} z\right), \{V_{k'}\}_{k'\in[[1,k-1]]\cup\{k+1\}}\right].$$

Detailing our notation, $\{V_{k'}\}_{k'\in[[1,k-1]]\cup\{k+1\}}$ means that, given the ordered sequence $\{V_1, V_2, \ldots, V_{k-1}, V_k\}$, we have replaced the last term as $\{V_1, V_2, \ldots, V_{k-1}, V_{k+1}\}$. For example, if we take the Markov chain at level $k = 3$ the generator $T_3(.,.,.)$ is

$$T_{\kappa,3}\left(x, y, \{V_{k'}\}_{k'\in[[1,3]]}\right) \tag{D.20}$$

$$= \int dz \, T_{\kappa,2}^{\ t_2}\left(z, y, \{V_1, V_2\}\right) T_{\kappa,2}\left[x, z + \frac{m_{j+1}}{\underline{m}_j}\left(y - m_j^{t_j} z\right), \{V_1, V_3\}\right]$$

$$= \int dz \, T_{\kappa,2}^{\ t_2}\left(z, y, \{V_1, V_2\}\right)$$

$$\times \left\{ \int dz' T_{\kappa,1}^{\ t_1}[z', z, V_1] T_{\kappa,1}\left[x, z' + \frac{m_2}{\underline{m}_1}\left(z - m_1^{t_1} z'\right) + \frac{m_3}{\underline{m}_1}\left(y - m_2^{t_2} z\right), V_3\right] \right\}.$$

## D.2 Computing the chain potential

In this section we will focus on computing the potential $V_t[.,.,.]$ for the very last step of the effective Markovian process presented in Sec. 6. We will consider that our connected solutions follow a memory pattern of $k_{tot}$ nested Markovian chains where it effectively jump from a solution $\mathbf{x}_{t_0}$ to $\mathbf{x}_{t_0+\Delta_t^{k_{tot}}}$. The aim is to compute the number of accessible solutions $\mathbf{x}_{t_0+\Delta_t^{k_{tot}}}$ when the chain is fixed from configuration $\mathbf{x}_{t_0}$ to $\mathbf{x}_{t_0+\Delta_t^{k_{tot}}-1}$. In this situation we recall that only a handful of overlaps are strictly fixed with our nested Markovian process, namely

$$\mathbb{E}_\xi\left[\mathbf{x}_{t_0+\Delta_t^{k_{tot}}} \cdot \mathbf{x}_{t_0+\Delta_t^{k_{tot}}-\Delta_t^j}\right] = m_{\Delta_t^j}, \tag{D.21}$$

where $\Delta_t^j$ corresponds to the number of "links" in the chain we pass when one iteration of the Markov-chain at level $j$ is performed. We recall that it is determined by the recursion

$$\Delta_t^k = (t_{k-1}+1)\Delta_t^{k-1}, \quad \text{and} \quad \Delta_t^1 = 1. \tag{D.22}$$

Moreover, we have to highlight that going from configuration $\mathbf{x}_{t_0+\Delta_t^{k_{tot}}-\Delta_t^{j+1}}$ to $\mathbf{x}_{t_0+\Delta_t^{k_{tot}}-\Delta_t^{j}}$ the system simply passes through the $j^{th}$ level of the Markovian process. All of these features allow simplifications in the potential $V^*_{t_0+\Delta_t^{k_{tot}}}[.,.,.]$. If we start for example with the entropic term, we have

$$\sum_{x_{t_0},\dots,x_{t_0+\Delta_t^{k_{tot}}-1}=\pm 1} \frac{1}{2} \prod_{j=t_0+1}^{t_0+\Delta_t^{k_{tot}}-1} \frac{e^{\sum_{t\le j'<j}\hat{m}_{j,j'}x_jx_{j'}}}{2\cosh\left(\sum_{t\le j'<j}\hat{m}_{j,j'}x_{j'}\right)} \tag{D.23}$$

$$\times \log\left[2\cosh\left(\sum_{t_0\le j'<t_0+\Delta_t^{k_{tot}}}\hat{m}_{t+\Delta_t^{k_{tot}},j'}x_{j'}\right)\right]$$

$$= \sum_{x_{t_0}=\pm 1}\frac{1}{2}\prod_{j=1}^{k_{tot}-1}\left[\sum_{x_{t+\Delta_t^{k_{tot}}-\Delta_t^j}=\pm 1}\frac{e^{\tilde{m}_{\Delta_t^j}x_{t_0+\Delta_t^{k_{tot}}-\Delta_t^j}x_{t_0+\Delta_t^{k_{tot}}-\Delta_t^{j+1}}}}{2\cosh(\tilde{m}_{\Delta_t^j})}\right]$$

$$\times \log\left[2\cosh\left(\sum_{j=0}^{k_{tot}}\hat{m}_{t_0+\Delta_t^{k_{tot}},t_0+\Delta_t^{k_{tot}}-\Delta_t^j}x_{t_0+\Delta_t^{k_{tot}}-\Delta_t^j}\right)\right],$$

with

$$\tilde{m}_{\Delta_t^j} = \operatorname{arctanh}\left(m_{\Delta_t^j}^{t_j}\right). \tag{D.24}$$

Regarding the energetic term, we can again use the Markov-chain generators $T_j(.,.,\{V_k\}_{k\in[\![1,j]\!]})$ that describe the jump from a configuration $\mathbf{x}_{t'}$ to $\mathbf{x}_{t'+\Delta_t^j}$. In particular, to go from $\mathbf{x}_{t_0+\Delta_t^{k_{tot}}-\Delta_t^{j+1}}$ to $\mathbf{x}_{t_0+\Delta_t^{k_{tot}}-\Delta_t^{j}}$ we have to apply $T_j^{t_j}(.,.,\{V_k\}_{k\in[\![1,j]\!]})$. With these generators we can rewrite the energetic term as

$$\alpha\int\prod_{j=0}^{t-1}dw_j P^{\kappa_0}[w_0]\prod_{j=1}^{t-1}\frac{e^{-\frac{\sum_{0\le j'\le j}\Sigma_{j,j'}(\mathbf{m})w_jw_{j'}}{2}}\Theta(\kappa^j-|w_j|)}{\int dw_j^* e^{-\frac{\sum_{0\le j'\le j}\Sigma_{j,j'}(\mathbf{m})w_j^*w_{j'}}{2}}\Theta(\kappa^j-|w_j^*|)} \tag{D.25}$$

$$\times \log\left[\int dw_t e^{-\frac{\sum_{0\le j'\le t}\Sigma_{t,j'}(\mathbf{m})w_tw_{j'}}{2}}\Theta(\kappa^t-|w_t|)\right]$$

$$= \alpha\int dw_{t_0}P^\kappa_{k_{tot}-\text{mem. state}}[w_{t_0}]\prod_{j=1}^{k_{tot}-1}\left[dw_{t_0+\Delta_t^{k_{tot}}-\Delta_t^j}T_j^{t_j}(w_{t_0+\Delta_t^{k_{tot}}-\Delta_t^j},w_{t_0+\Delta_t^{k_{tot}}-\Delta_t^{j+1}},\{V_k\}_{k\in[\![1,j]\!]})\right]$$

$$\times \log\left[H\left(\kappa,\underline{m}_1 w_{t_0+\Delta_t^{k_{tot}}-\Delta_t^1}+\sum_{j=1}^{k_{tot}}\underline{m}_j\left(w_{t_0+\Delta_t^{k_{tot}}-\Delta_t^j}-m_{\Delta_t^{j-1}}^{t_{j-1}}w_{t_0+\Delta_t^{k_{tot}}-\Delta_t^{j-1}}\right),V_{k_{tot}}\right)\right],$$

where we recall the notation

$$\underline{m}_1 = m_1, \quad \underline{m}_{k\ne 1} = \frac{m_{\Delta_t^k}-m_{\Delta_t^{k-1}}^{t_{k-1}+1}}{1-m_{\Delta_t^{k-1}}^{2t_{k-1}}}, \quad V_1 = 1-m_1^2, \quad V_{k\ne 1} = V_{k-1}+\frac{\left(m_{\Delta_t^k}-m_{\Delta_t^{k-1}}^{t_{k-1}+1}\right)^2}{1-m_{k-1}^{2t_{k-1}}}. \tag{D.26}$$

Combining together the entropic and energetic terms we obtain

$$
V^*_{t_0+\Delta_t^{k_{tot}}}\left[\mathbf{x}_0, \{m_{\Delta_t^j}\}_{j=[\![1,k_{tot}]\!]}, \{\hat{m}_{t_0+\Delta_t^{k_{tot}},t_0+\Delta_t^{k_{tot}}-\Delta_t^j}\}_{j=[\![1,k_{tot}]\!]}, \{\kappa_j=\kappa\}_{j\in[\![t_0,t_0+\Delta_t^{k_{tot}}]\!]}\right]
$$

$$
=-\sum_{j=0}^{k_{tot}}\hat{m}_{t_0+\Delta_t^{k_{tot}},t_0+\Delta_t^{k_{tot}}-\Delta_t^j}m_{\Delta_t^j} \tag{D.27}
$$

$$
+\sum_{x_{t_0}=\pm1}\frac{1}{2}\prod_{j=1}^{k_{tot}-1}\left[\sum_{x_{t+\Delta_t^{k_{tot}}-\Delta_t^j}=\pm1}\frac{e^{\tilde{m}_{\Delta_t^j}x_{t_0+\Delta_t^{k_{tot}}-\Delta_t^j}x_{t_0+\Delta_t^{k_{tot}}-\Delta_t^{j+1}}}}{2\cosh(\tilde{m}_{\Delta_t^j})}\right]
$$

$$
\times\log\left[2\cosh\left(\sum_{j=0}^{k_{tot}}\hat{m}_{t_0+\Delta_t^{k_{tot}},t_0+\Delta_t^{k_{tot}}-\Delta_t^j}x_{t_0+\Delta_t^{k_{tot}}-\Delta_t^j}\right)\right]
$$

$$
+\alpha\int dw_{t_0}P^\kappa_{k_{tot}-\text{mem. state}}[w_{t_0}]
$$

$$
\times\prod_{j=1}^{k_{tot}-1}\left[dw_{t_0+\Delta_t^{k_{tot}}-\Delta_t^j}T_j^{t_j}(w_{t_0+\Delta_t^{k_{tot}}-\Delta_t^j},w_{t_0+\Delta_t^{k_{tot}}-\Delta_t^{j+1}},\{V_k\}_{k\in[\![1,j]\!]})\right]
$$

$$
\times\log\left[H\left(\kappa,\underline{m}_1w_{t_0+\Delta_t^{k_{tot}}-\Delta_t^1}+\sum_{j=1}^{k_{tot}}\underline{m}_j\left(w_{t_0+\Delta_t^{k_{tot}}-\Delta_t^j}-m_{\Delta_t^{j-1}}^{t_{j-1}}w_{t_0+\Delta_t^{k_{tot}}-\Delta_t^{j-1}}\right),V_{k_{tot}}\right)\right].
$$

From then on, the potential $V^*_{t+\Delta_t^{k_{tot}}}[.,.,.,.]$ can be evaluated numerically by integrating sequentially in the entropic and energetic term over the variables $\{x_{t_0+\Delta_t^{k_{tot}}-\Delta_t^1},w_{t_0+\Delta_t^{k_{tot}}-\Delta_t^1}\}$, then $\{x_{t_0+\Delta_t^{k_{tot}}-\Delta_t^2},w_{t_0+\Delta_t^{k_{tot}}-\Delta_t^2}\}$ and so on and so forth. This sequence of integration yields similar recursion equations than the one obtain for the Markov-chain generators $T_j(.,.,.)$.

Finally comes the optimization scheme for the potential. We want either to determine the localization/delocalization critical point $\kappa^\alpha_{k_{tot}-\text{mem. state}}$ ($\alpha$ being fixed) or $\alpha^\kappa_{k_{tot}-\text{mem. state}}$ ($\kappa$ being fixed). We recall that the transition occurs when the potential becomes null. In the present case, it is less numerically complex to determine $\alpha^\kappa_{k_{tot}-\text{mem. state}}$. Indeed, after satisfying the saddle-points over the fields $\{\hat{m}_{t_0+\Delta_t^{k_{tot}},t_0+\Delta_t^{k_{tot}}-\Delta_t^j}\}_{j=[\![1,k_{tot}]\!]}$ we can identify that the potential is null for

$$
\alpha=-\frac{V_{\text{entropic}}}{V_{\text{energetic}}}, \tag{D.28}
$$

with

$$
V_{\text{entropic}}=-\sum_{j=0}^{k_{tot}}\hat{m}_{t_0+\Delta_t^{k_{tot}},t_0+\Delta_t^{k_{tot}}-\Delta_t^j}m_{\Delta_t^j} \tag{D.29}
$$

$$
+\sum_{x_{t_0}=\pm1}\frac{1}{2}\prod_{j=1}^{k_{tot}-1}\left[\sum_{x_{t+\Delta_t^{k_{tot}}-\Delta_t^j}=\pm1}\frac{e^{\tilde{m}_{\Delta_t^j}x_{t_0+\Delta_t^{k_{tot}}-\Delta_t^j}x_{t_0+\Delta_t^{k_{tot}}-\Delta_t^{j+1}}}}{2\cosh(\tilde{m}_{\Delta_t^j})}\right]
$$

$$
\times\log\left[2\cosh\left(\sum_{j=0}^{k_{tot}}\hat{m}_{t_0+\Delta_t^{k_{tot}},t_0+\Delta_t^{k_{tot}}-\Delta_t^j}x_{t_0+\Delta_t^{k_{tot}}-\Delta_t^j}\right)\right],
$$

$$V_{\text{energetic}} = \int dw_{t_0} P^\kappa_{k_{tot}-\text{mem. state}}[w_{t_0}] \tag{D.30}$$

$$\times \prod_{j=1}^{k_{tot}-1} \left[ dw_{t_0+\Delta_t^{k_{tot}}-\Delta_t^j} T_j^{t_j}(w_{t_0+\Delta_t^{k_{tot}}-\Delta_t^j}, w_{t_0+\Delta_t^{k_{tot}}-\Delta_t^{j+1}}, \{V_k\}_{k\in[\![1,j]\!]}) \right]$$

$$\times \log \left[ H\left( \kappa, \underline{m}_1 w_{t_0+\Delta_t^{k_{tot}}-\Delta_t^1} + \sum_{j=1}^{k_{tot}} \underline{m}_j \left( w_{t_0+\Delta_t^{k_{tot}}-\Delta_t^j} - m^{t_{j-1}}_{\Delta_t^{j-1}} w_{t_0+\Delta_t^{k_{tot}}-\Delta_t^{j-1}} \right), V_{k_{tot}} \right) \right] .$$

Consequently, having fixed $\kappa$, we can obtain $\alpha^\kappa_{k_{tot}-\text{mem. state}}$ by simply tuning the set of variables $\left\{ m_{\Delta_t^j}, \Delta_t^j \right\}_{j\in[\![1,k_{tot}]\!]}$ so as to maximize the value of $\alpha$ given by Eq. (D.28). In particular, while our construction implies that $t_j \in \mathbb{N}$ (and consequently $\Delta_t^j \in \mathbb{N}$) for all $j \in [\![2, k_{tot}]\!]$, we determine the critical point after extending the computation to $t_j \in \mathbb{R}$.

Again, we recall that the potential $V_{t'}^*[.,.,.,.]$ should be evaluate throughout the entire effective Markovian process to ensure that it remains positive (or null) for the period $t_0 \to t_0 + \Delta_t^{k_{tot}}$. However, we observe numerically that it is always the very last step, i.e. $t_0 + \Delta_t^{k_{tot}} - 1 \to t_0 + \Delta_t^{k_{tot}}$, that has the minimal potential and that is consequently the limiting "link" in the chain.

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
