# Peer review of "How to escape atypical regions in the symmetric binary perceptron: a journey through connected-solutions states"

_SciPost Physics, doi:SciPost Phys. 18, 115 (2025)_

## Round 2 · Referee Report · Anonymous (Referee 1) · 2024-11-26

Report

In this manuscript, the authors introduce a method to explore the solution set of the symmetric binary perceptron. Their formalism introduces a chain of solutions (anchored on a planted solution x_0), allowing them to explore atypical connected regions of solutions. Parts 1 to 4 introduce the chain formalism, explain the need for a simplification (no-memory anzatz) assuming that solutions at distance larger than 1 on the chain deccorellates. Part 5 confront theoretical results obtained in previous sections (within no-memory anzatz) with numerical simulations. Part 6 builds the next steps to go beyond the no-memory anzatz. The paper is well written and present strong results. It is solid scientifically and the conclusions are well supported. I recommend it for publication.

I have some comments/suggestions that I think could improve the comprehension of the manuscript:

  • eqs (32) and (33): maybe you should give more explanations on these inequalities, in particular explain why do we need to distinghish the cases K_t>K_{t-1} and K_t<=K_{t-1}. For eq (33), if I understand correctly, the local entropy for m_1 < m <= 1 is positive because x_t solutions are less constrained (K_t>K_{t-1})? But is there an intuitive explanation why there is still a range [m_0,m_1] for which V<0 ? Also, when restricted to one case (e.g. K_t>K_{t-1}), how do the values m_0, m_1 change when changing \kappa_t, \kappa_{t-1} ? If so, are there choices of K_t,K_{t-1} for which m_1=m_0 ? Do the values m_0,m_1 also depend on the previous choices of \kappa_j , j<t-1 ?

  • eq (29): From line 1 to line 2, the term with the integral over w_{t-1} is removed, how do you justify it?

  • also, I couldn't find in the main text a definition of the large size limit that is usually taken for making the replica computation. Sometimes, I found that it is a bit unclear what are the size-dependent parameters (overlap constraint $m$, margin $\kappa$, density of constraints $\alpha$, and what are the constant ones. Since in section 4.2 the author justifies its choice of a non-trivial scaling for the magnetization $m$, I think it would be useful to first explain in a few words what is the usual setting?

  • eq (17, 18) and below: I might have missed the point, but I don't really agree with the justification for the iterative procedure fixing the m's and mhat's. Even if the x_1,x_2,...x_t are built recursively, it doesn't mean that the optimization of V_{t} over all parameters simultaneously ({m_{j,j'}, mhat{j,j'}) is equivalent to optimizing them one by one looking at V_{t=1}, V_{t=2}, especially because I dont see a clear recursive relation between V^_{t} and V^{t-1}. In other words, the optimal {m 0<=j'<j<=t)} for the chain of length $t$ might be different from the optimal m_{3,1} for a chain of length 3, and the optimal m_{4,1},m_{4,2} for a chain of lenth 4... Are you describing here an aproximate method to get the optimized values ? If so, I think you should mention it.

  • still in the same paragraph, the sentence 'To put it more concretely, we set mhat_{1,0} and m_{1,0} by optimizing V^{t=1}. Then, we fix mhat by optimizing V^}, mhat_{2,1}, m_{2,0} and m_{2,1{t=2} (and so on and so forth).' But m are parameters of the potential V, so one shouldn't optimize them. Is it a typo ?} and m_{2,1

  • section 4.1: Since you have the full expression (17), did you checked numerically (maybe only for t=2, t=3) that the no-memory anzatz was justified, by computing the hat{m}_{j,j'}, |j-j'|>1 ?

  • section 5.1: for a given run, the mapping f: t_rescaled -> t might also change along the run ? In other words, I think that flipping f(N) spins might require a different time depending on whether we are at the beginning or at the end of the Monte-Carlo run, so how could you justify that you take a mapping that remains the same along the run ? On fig. 3,4,5, could you justify the choice made for the rescaled time function f ?

Other remarks:

  • Figure 2: I suggest using a larger font for writing the means.

  • In section 1.2, paragraph 'Solutions chain formalism', the sentence "However, when taking the limit m_{j,j−1}->1, it can be shown that an anneal ed evaluation of the entropy becomes exact." It is not clear at first sight that this is actually shown in appendix A. Could you make the link with the appendix, so the reader is not confused ?

  • eq (24): might be useful to remind to the reader that the definition of the function H is given in (10) ?

  • section 5.3: when describing fig. 3 (in the main text), you mention both times the right panel, I guess the second time (when talking about the plot with rescaled time) 'right' should be replaced by 'left'

  • section 5.1, typo: the matching function is called g : t_{rescaled} -> t in the main text, but it is called f in the figures.

Recommendation

Publish (surpasses expectations and criteria for this Journal; among top 10%)

  • validity: -
  • significance: -
  • originality: -
  • clarity: -
  • formatting: -
  • grammar: -

Author:  Damien Barbier  on 2025-01-31  [id 5169]

(in reply to Report 1 on 2024-11-26)
Category:
answer to question
correction

We would like to start by thanking the referee for their meaningful comments and questions. In the following we will answer him point by point.

"-Eqs (32) and (33): maybe you should give more explanations on these inequalities [...]."

To clarify the discussion about the presence of an overlap-gap close to $m'=1$, we added a schematic representation the potential $V_{t}[.]$ for three cases of $\kappa_t$. In particular, if $\kappa_t$ is set slightly higher than $\kappa_{t-1}$, the gap starts closing ($m_0$ increases while $m_1$ decreases) but it remains present. In other words, the potential gets smoothly deformed as we increased $\kappa_t$. As mentioned by the referee, we have forgotten to mention that if $\kappa_t$ is increased high enough the gap closes. However, we ruled out this case in our paper. Closing the gap for each solution in the chain causes $\kappa_t$ to diverge very quickly and seemed unreasonable physically speaking. We added in the main text a few comments mentioning this case. We also added that these curves are dependent on the whole past of the chain. In fact, it depends on the interactions distribution $P^{\kappa_{t-1}}[.]$, and this distribution depends on the whole history of the chain.

"- eq (29): From line 1 to line 2, the term with the integral over $w_{t-1}$ is removed, how do you justify it?"

When the local entropy is a non-trivial function, both terms (the entropic and the energetic one) are of the same magnitude in order to compete. In the scaling argument of Eq. (29), only the overall magnitude of the potential matters. Thus, keeping one of the two terms is usually enough.

"- also, I couldn't find in the main text a definition of the large size limit [...] I think it would be useful to first explain in a few words what is the usual setting?"

To make this part clearer we added before the discussion on the scaling of $m$, that this computation is well controlled for $N$ going to infinity first and then keeping all other parameters constants. Then, we discuss how the saddle-point approximation can actually be extended to a broader regime of parameter values.

"- eq (17, 18) and below: I might have missed the point, but I don't really agree with the justification for the iterative procedure fixing the m's and mhat's.[...] If so, I think you should mention it."

The comment is right, a global optimization of the parameters could give a different result compared to the time-ordered optimization. We added this comment in the main text. We also mentioned that to take into account time-ordered dynamics, the time-ordered optimization scheme seemed more adequate.

"- still in the same paragraph [...]. Is it a typo ?"

This is indeed a typo and we corrected it in the new version of the manuscript.

"- section 4.1: Since you have the full expression (17), did you checked numerically (maybe only for t=2, t=3) that the no-memory anzatz was justified, by computing the $\hat{m}_{j,j'}$, $|j-j'|>1$ ?"

Yes, we tested the time ordered optimization up to t=3 with good numerical agreements with the no-memory Ansatz. The paper being already quite long we decided not to present these results.

"- section 5.1: for a given run, the mapping f: $t_{rescaled}\rightarrow t$ might also change along the run ? [...]
On fig. 3,4,5, could you justify the choice made for the rescaled time function f ?"

The intuition is correct, the beginning of the Monte-Carlo has a higher rate of accepted spin-flips than later in the run. If the spin-flip rate was constant, the function $g$ would simply be a linear function: to go from $t_{\rm rescaled} f(N)$ from $2t_{\rm rescaled} f(N)$ spin-flips we would have to wait the same amount of time as going from $0$ to $t_{\rm rescaled} f(N)$ spin-flips. Therefore, the slowing down of the dynamics corresponds to $g$ being convex. We added a comment about that in the main text. Moreover, we have no choice on the rescaling function for each Monte-Carlo run. In practice we simply stored the correlation function for different times and how many spin-flips were accepted at these exact moments. Then, we plotted the correlation functions as function of how many spin-flips were actually accepted for each time.

Other remarks:

"- Figure 2: I suggest using a larger font for writing the means."

In the new version of the manuscript the font is now larger.

"- In section 1.2, paragraph 'Solutions chain formalism'. [...] Could you make the link with the appendix, so the reader is not confused ?"

Taking into account this comment, we renamed App A and App B.2. We specified that App A simply gives the proof for the simple planted setting, and App B.2 generalizes it for the chain formalism. In the main text we redirected the reader not to App A but to App B.2.

"- eq (24): might be useful to remind to the reader that the definition of the function H is given in (10) ?"

We added a reminder for the expression of H in the new version of the paper

"- section 5.3: when describing fig. 3 (in the main text), you mention both times the right panel, I guess the second time (when talking about the plot with rescaled time) 'right' should be replaced by 'left'"

We corrected this typo in the new version of the manuscript

"- section 5.1, typo: the matching function is called g : $t_{rescaled}\rightarrow t$ in the main text, but it is called f in the figures."

The referee pointed out a confusing notation. The label had to specify the correlation profile of the no-memory chain in the plot (dashed black curve). We dropped this confusing notation and made it clearer in the new manuscript.

---

## Round 2 · Referee Report · Anonymous (Referee 2) · 2024-11-29

Report

In the present paper, the author studies atypical solutions in the binary symmetric perceptron model to understand why some solutions to the problem are algorithmically accessible, even if typical solutions are isolated and unreachable. The question the author tries to answer is very relevant and still largely open. The work seems scientifically solid, but the presentation is challenging to follow and should be improved.

The method proposed by the author to count atypical solutions is rather complicated as it requires the optimization of a potential over a large number of parameters. So let me start with a naive question, which I believe is needed to understand why the author is doing such a challenging computation. Given that atypical solutions reached by the Monte Carlo (MC) algorithm implemented by the author have a very specific distribution of margins P(w), would it be possible to count such atypical solutions and check whether they are less isolated than typical ones or even connected? We know very well that biasing the measure towards atypical solutions can change the phase diagram of a constraint satisfaction problem and the corresponding algorithmic thresholds (see, e.g., Budzynski et al., J. Stat. Mech. (2019) 023302). The author should comment on this kind of computation both if this has already been done (and in that case, should report the results) and if it is still missing (and in that case, should explain why not doing it instead of building complicated chains of solutions).

Observing the data reported in Figure 4 (left), the author states that the MC dynamics require superlinear (in N) times to go below the correlation predicted within the no-memory ansatz (roughly 0.9). The author insists a lot on this point, repeating it many times throughout the paper. The problem is that the data shown in Figure 4 (left) do not support the author's claim: below the value 0.9, the decorrelation is becoming faster (and not slower) by increasing the system size. Indeed, the largest size (N=40000, yellow curve) shows the faster relaxation. The author should better explain his claim or modify the claims throughout the paper, making it in agreement with the data reported.

Moreover, I don't believe that considering superlinear times is really needed to describe that regime of correlations. The author shows that the no-memory ansatz is unable to enter that regime, but probably, one should just consider solutions with a different type of correlation, which can still be reached in linear times. An illuminating example could be the one discussed in Ref. [33]: in the xorsat model, the simplest ansatz describes well the relaxation dynamics until an energy value close to 0.1 (see Fig. 6 of Ref. [33]); below that value, a more complicated ansatz is needed, but the times to reach those energy values are still linear in the system size N. So, I think the author should decouple the breaking of the validity of the no-memory ansatz to the appearance of superlinear times (otherwise, he should provide a piece of evidence that the present version of the paper is lacking).

In Figure 6, when commenting on the distribution of margins P(w) in configurations reached by the MC, the author claims that the difference from the truncated Gaussian distribution extends over a distance $\sqrt{1-m^2}$ from the edge. It would be useful to show analytical evidence for such a claim.

One of the main results of the paper is presented in Figure 7, where the author shows that a nontrivial region of $\kappa$ values can be explored through delocalized no-memory chains. Given the importance of the result, I suggest the author to present it also for other values of $\alpha$ and other choices of m. Restricting to $m=1-2/N$ seems very limited. Once the threshold is computed for several values of m, one could, in principle, optimize over this parameter to maximize the delocalized phase. If this is not doable or not useful, the author should explain why.

If the no-memory computation can be done with different m values (e.g., with $m=1-c/N$ for several values of c), I would appreciate a comparison of the MC data with the solution predicted by the no-memory ansatz with different m values. For example, the MC data shown in Figure 8 look like a distribution of margins in the no-memory ansatz but with a smaller value of m (i.e., a larger value of c). Making the comparison between MC data and the no-memory ansatz only with $m=1-2/N$ is very limited.

Section 6 of the manuscript is very hard to follow. The reader will probably get lost in very long formulas, and the messages the author would like to convey are at risk of being missed. I suggest the author reorganize the section by moving technical details in the appendix and concentrating on explaining the main results.

In my opinion, section 6, reporting the results obtained with memory ansatz, should provide evidence that the analytical prediction is closer to the MC results. Given that the analytical computation with this new ansatz is much more demanding, the author must provide convincing evidence that it is worth making this extra effort. At present, the only evidence that the memory ansatz is doing better than the no-memory ansatz comes from Table 1. Moreover, in that table, the result for k=1 (no-memory) is obtained, if I understand correctly, with m=0.98 and not with $m=1-2/N$. Can the k=1 threshold be optimized by changing the value of m?

The suggestion of comparing the memory ansatz with the no-memory ansatz optimized over the m parameter is further supported by the results reported in Figures 10 and 11. In Figure 10, the decay of the correlation in the memory ansatz is slower than in the no-memory ansatz but looks still like an exponential decay with a larger m value. In Figure 11, the P(w) computed in configurations reached through the memory ansatz is very close to the one of the no-memory ansatz and probably very far from the P(w) obtained via the MC. Data from MC simulations should be added to Figure 11 (at least in the right panel) to provide evidence about how good the proposed ansatz is in describing the actual MC dynamics.

I understand that the optimization of the parameters in the memory ansatz is complicated, but the author should provide some information about the optimal parameters obtained at the end of this process. This would be important to understand the differences with respect to the no-memory ansatz.

Many figures contain fonts that are unreadable when printed. This must be avoided.

Minor points: - Between equations (4) and (5), the definition of margins lacks the right normalization. - On page 18 the comment "(conversely m->0)" is obscure to me. "Conversely" to what?

Recommendation

Ask for minor revision

  • validity: good
  • significance: high
  • originality: high
  • clarity: low
  • formatting: good
  • grammar: good

Author:  Damien Barbier  on 2025-01-31  [id 5170]

(in reply to Report 2 on 2024-11-29)
Category:
answer to question
reply to objection
correction
suggestion for further work

Before responding, we would like to thank the referee for the very interesting comments and suggestions they made on this manuscript. In the following, we will keep the order of the comments and respond to them one by one.

"The method proposed by the author to count atypical solutions is rather complicated as it requires the optimization of a potential over a large number of parameters. [...] The author should comment on this kind of computation both if this has already been done (and in that case, should report the results) and if it is still missing (and in that case, should explain why not doing it instead of building complicated chains of solutions)."

In the case of a no-memory chain, the local entropy around a solution is given by Eq.(27). We mention in Sec 4.3 that it has the same form as the local entropy around atypical solutions with the predicted distribution of margins P(w) (i.e. without any consideration about chaining). What I understand is that the referee would like to have a a full plot of $V_{t}[.]$ as a function of $m'$ and check if the overlap gap closes. However, the problem with SBP is that the local entropy gets more and more dominated by isolated solutions as $m'$ goes away from one. Therefore, any prediction for $1-m'\gg 1-m$ ($m$ being the chain overlap) cannot account for any dynamics as the potential describe inaccessible solutions. We added in Sec 5.3 a paragraph mentioning that the local entropy around a planted solution has already been studied -in D. Barbier, A. E. Alaoui, F. Krzakala and L. Zdeborová, Journal of Physics A: Mathematical and Theoretical 57(19), 195202 (2024)- but fails to predict correctly the Monte-Carlo dynamics has it lacks the connectivity condition we add in this paper.

"Observing the data reported in Figure 4 (left), the author states that the MC dynamics require superlinear (in N) times to go below the correlation predicted within the no-memory ansatz (roughly 0.9) [...]. The author should better explain his claim or modify the claims throughout the paper, making it in agreement with the data reported."

In Figure 4 (left), the time axis is rescaled to account only for accepted spin-flip. This scaling is interesting because it removes any notion of slowing down or accelerating the dynamics. The fact that increasing the system size makes the correlation function go lower highlights a simple fact. Each time the Monte-Carlo dynamics flip a spin, there is a finite probability that it performs a backward move. In the chain formalism, this means that instead of going from ${\bf x}_t$ to ${\bf x}_{t+1}$ we go from ${\bf x}_t$ to ${\bf x}_{t-1}$. As the system size is increased this move gets more and more negligible and the correlation function collapses on the prediction given by a no-memory chain that only accept forward moves. We added comments in the main text about this phenomenon.

"Moreover, I don't believe that considering superlinear times is really needed to describe that regime of correlations [...]. So, I think the author should decouple the breaking of the validity of the no-memory ansatz to the appearance of superlinear times (otherwise, he should provide a piece of evidence that the present version of the paper is lacking)."

It is most probably true that a more complex memory kernel could allow us to decorrelate more from initialization. This is actually the point of the section about nested Markovian chains. In terms of Monte-Carlo dynamics, this means that introducing memory feedback could improve the decorrelation. However, the naive Monte-Carlo dynamics appears to follow only the no-memory ansatz for $\mathcal{O}(N)$ time steps. There is indeed part of the dynamics for which the system deccorelates more than the no-memory chain prediction. Nevertheless, for a fixed normalized time of the dynamics (number of total spin-flip trials divided by the system size) this regime disappear as $N$ grows (see left panels of Fig. 3 and 4). This means that this regime cannot be entered in $\mathcal{O}(N)$ spin-flip trials when $N$ goes to infinity.

"In Figure 6, when commenting on the distribution of margins P(w) in configurations reached by the MC, the author claims that the difference from the truncated Gaussian distribution extends over a distance $\sqrt{1-m2}$ from the edge. It would be useful to show analytical evidence for such a claim."

We have added the analytical expression of the margins distribution ${P}_{\rm no-mem.\, state}^{\kappa}[w]$. It is a simple Gaussian multiplied by error functions. Therefore, it deviates from the Gaussian distribution if the error functions are not approximately equal to one. It is straightforward to observe that it happen only for $\kappa-\vert w\vert\sim \sqrt{1-m^2}$.

"One of the main results of the paper is presented in Figure 7, where the author shows that a nontrivial region of $\kappa$ values can be explored through delocalized no-memory chains [...]. If this is not doable or not useful, the author should explain why."

"If the no-memory computation can be done with different m values (e.g., with $m=1-c/N$ for several values of c), I would appreciate a comparison of the MC data with the solution predicted by the no-memory ansatz with different m values [...]. Making the comparison between MC data and the no-memory ansatz only with $m=1-2/N$ is very limited."

The referee proposes to study chains for which the overlap $m$ can tuned. This proposal is interesting but is a bit beyond the scope of this paper. We want to study connected states and for this it is imperative to send $m$ to one. Otherwise, we cannot assess if the solutions we probe are actually connected. Then, because of finite size effects (both in the simulations and the theoretical computation) we cannot set the distance between to neighbors in the chain to brutally to zero but to $2/N$. We clarified this point by rephrasing the discussion in Sec. 4.2. Moreover, changing $\alpha$ does provide any qualitative changes for the phase diagram (Fig. 7). We added a comment about this in the main text.

"Section 6 of the manuscript is very hard to follow. The reader will probably get lost in very long formulas, and the messages the author would like to convey are at risk of being missed. I suggest the author reorganize the section by moving technical details in the appendix and concentrating on explaining the main results."

Following the recommendation of the referee, we moved the detailed computation in an appendix and dedicated the main text to discussions about the results.

"In my opinion, section 6, reporting the results obtained with memory ansatz, should provide evidence that the analytical prediction is closer to the MC results [...] . Can the $k=1$ threshold be optimized by changing the value of m?"

In the regime where the MC decorrelates more than the no-memory solution, the dynamics starts probing a lot short paths with dead ends. It slows down because it has to explore those short paths, backtrack and then catch an atypical path that does not stop. The chain formalism cannot catch this behavior of the dynamics as it is tuned exactly to avoid dead ends. Thus, the matching between the MC simulations and the memory $Ansatz$ remains to be solved. One idea could be to introduce memory feedback in the MC dynamics. This could force it to probe paths with the right correlation profile and thus avoid these dead ends. Ultimately, we present a memory kernel computation in this paper to show the direction we should look to find connected paths at low $\alpha$, $\kappa$. This an important question as it has been proven that local algorithms (probing connected paths of solutions) work for $\alpha=\mathcal{O}(1)$ and $\kappa=\mathcal{O}(1)$. Finally, we choose $m=0.98$ to pick an chain overlap close to one, and for which the distribution of margins $P[w]$ can be observably compressed with a handful for levels in the memory $Ansatz$.

"The suggestion of comparing the memory ansatz with the no-memory ansatz optimized over the m parameter is further supported by the results reported in Figures 10 and 11 [...]. Data from MC simulations should be added to Figure 11 (at least in the right panel) to provide evidence about how good the proposed ansatz is in describing the actual MC dynamics."

We agree with the referee that checking the no-memory chain (with optimized $m$) appears to be a natural direction to investigate. However, it would not catch the correct behavior of the model. Without sending $m$ to one, we loose any control as whether the solutions are isolated or connected. In fact, the overlap gap that we highlight in Sec. 4.3 would span over extensive distances if $m$ is not close to one. The solutions we would start to sample would be truly isolated. One of the main interests with the nested Markovian chain is to built an effective no-memory chain with arbitrary $m$, given that each jump passes through connected solutions. Again, to make the point about sending $m$ to one clearer, we rewrite part of the discussion in Sec 4.2.

"I understand that the optimization of the parameters in the memory ansatz is complicated, but the author should provide some information about the optimal parameters obtained at the end of this process. This would be important to understand the differences with respect to the no-memory ansatz."

Regarding this comment, we added in the main text a table summarizing the order parameters for each optimized nested Markovian chain.

"Many figures contain fonts that are unreadable when printed. This must be avoided."

We increased the size of the font in the new version of the manuscript.

"Minor points: - Between equations (4) and (5), the definition of margins lacks the right normalization. - On page 18 the comment "(conversely m->0)" is obscure to me. "Conversely" to what?"

We corrected the mentioned typos. Regarding the sentence "(conversely $m\rightarrow0$)", it is a mistake. We were mentioning that $N\rightarrow+\infty$ implies $m=1-2/N$ goes to one. We rephrased this sentence.

---

## Round 3 · Referee Report · Anonymous (Referee 2) · 2025-2-4

Report

The author has made several changes to explain better the points I found unclear.
I believe the manuscript is worth publishing in its present form.

Recommendation

Publish (easily meets expectations and criteria for this Journal; among top 50%)

---

## Round 3 · Referee Report · Anonymous (Referee 1) · 2025-2-21

Report

The author has made significant efforts to present some part of the article more clearly, and answered my questions satisfactorily. I recommend this article for publication.

Recommendation

Publish (surpasses expectations and criteria for this Journal; among top 10%)

---

## Round 3 · List of Changes

-In Sec. 1.2, when mentioning the replica computation being annealed, we redirected explicitely the reader to App B.2.
-In Sec 3, we added a few comments about time-ordered versus non time-ordered optimization of the chain potential.
-In Sec 4.2, we rephrased part of the discussion about sending the chain overlap 'm' to one. In particular, we insisted on the fact that it cannot be an adjustable order parameter since we want to describe connected solutions.
-In Sec 4.3, we added details about the overlap-gap appearing in the chain formalism. We also added a sketch of the chain potential to make the discussion clearer.
-In Sec 5.1, we added comments about the function rescaling time. In particular, we insisted on the fact that the slowing down of the dynamics makes this function convex.
-In Sec 5.2, we added the analytical form of the stable interactions distribution for the no-memory chain.
-In Sec 5.3, we mentioned that the simple planted computation (without connectivity criteria) has already been studied. However, it fails to describe the Monte-Carlo dynamics. Again in this section, we added a comment explaining why the correlation drops in Fig. 5 as the system size is increased.
-In Sec 5.4, we mentioned that the phase diagram does not change qualitatively as the value of alpha is changed.
-In Sec 6, the computation steps for the nested Markovian chain have been moved to App D.1. We also added a table summarizing the order parameters obtained after optimization for the presented memory kernels.
-We increased the font size in most of the figures included in the manuscript.

---

## Editorial Decision

published